# Stochastic Halpern Iteration with Variance Reduction for Stochastic Monotone Inclusions

**Xufeng Cai**
Department of Computer Sciences
University of Wisconsin-Madison
`xcai74@wisc.edu`

**Chaobing Song**
Department of Computer Sciences
University of Wisconsin-Madison
`songcb16@gmail.com`

**Cristóbal Guzmán**
Institute for Mathematical and Computational Eng.
Facultad de Matemáticas and Escuela de Ingeniería
Pontificia Universidad Católica de Chile
`crguzmanp@mat.uc.cl`

**Jelena Diakonikolas**
Department of Computer Sciences
University of Wisconsin-Madison
`jelena@cs.wisc.edu`

## Abstract

We study stochastic monotone inclusion problems, which widely appear in machine learning applications, including robust regression and adversarial learning. We propose novel variants of stochastic Halpern iteration with recursive variance reduction. In the cocoercive—and more generally Lipschitz-monotone—setup, our algorithm attains $\epsilon$ norm of the operator with $\mathcal{O}(\frac{1}{\epsilon^3})$ stochastic operator evaluations, which significantly improves over state of the art $\mathcal{O}(\frac{1}{\epsilon^4})$ stochastic operator evaluations required for existing monotone inclusion solvers applied to the same problem classes. We further show how to couple one of the proposed variants of stochastic Halpern iteration with a scheduled restart scheme to solve stochastic monotone inclusion problems with $\mathcal{O}(\frac{\log(1/\epsilon)}{\epsilon^2})$ stochastic operator evaluations under additional sharpness or strong monotonicity assumptions.

## 1 Introduction

Recent trends in machine learning (ML) involve the study of models whose solutions do not reduce to optimization but rather to *equilibrium conditions*. Standard examples include generative adversarial networks, adversarially robust training of ML models, and training of ML models under notions of fairness. It turns out that several of these equilibrium conditions (including, but not limited to, first-order stationary points, saddle-points, and Nash equilibria of minimax games) can be cast as solutions to a *monotone inclusion* problem, which is defined as the problem of computing a zero of a (maximal) monotone operator $F : \mathbb{R}^d \to \mathbb{R}^d$ (see (MI) for a formal definition). In the context of min-max optimization problems, monotone inclusion reduces to a stationarity condition, which for unconstrained problems boils down to finding a point with small gradient norm.

Of particular interest to machine learning are stochastic versions of these problems, in which the operator $F$ is not readily available, but can only be accessed through a stochastic oracle $\widehat{F}$. Such are the settings mentioned above, where the definitions of equilibria involve expectations over continuous high-dimensional spaces. The corresponding problem, known as the *stochastic monotone inclusion*, has not been thoroughly studied, particularly in the context of its stochastic oracle complexity. Understanding stochastic oracle complexity of monotone inclusion in all standard settings with Lipschitz operators, from the algorithmic aspect, is the main motivation of this work.

36th Conference on Neural Information Processing Systems (NeurIPS 2022).

## 1.1 Contributions

We study three main classes of stochastic monotone inclusion problems with Lipschitz operators, defined by the assumptions made about the operator itself: (i) cocoercive class, which is the most restricted class, but nevertheless fundamental for understanding monotone inclusion, as it relates to the problem of finding a fixed point of a nonexpansive (1-Lipschitz) operator; (ii) Lipschitz monotone class, which is perhaps the most basic class arising in the study of smooth convex-concave min-max optimization problems; and (iii) Lipschitz monotone class with an additional sharpness property of the operator. Sharpness is a widely studied property of optimization problems, often referred to as the "local error bound" condition, which is weaker than strong convexity and roughly corresponds to the problem landscape being curved outside of the solution set (see [41] for a survey of classical results).

From an algorithmic standpoint, we consider variants of classical Halpern iteration [21], which was originally introduced for solving fixed point equations with nonexpansive operators. Variants of this iteration have recently been shown to lead to (near-)optimal first-order oracle complexity for all aforementioned standard problem classes in *deterministic* settings [12, 13, 53]. However, to the best of our knowledge, stochastic variants of these methods have received very limited attention prior to our work. The only results we are aware of are for a two-step extragradient-like variant of Halpern iteration in negative comonotone Lipschitz settings [29] and which show that when variance of operator estimates is bounded by order-$\frac{\epsilon^2}{k}$ in iteration $k$, the method attains operator norm $\epsilon$ after $\mathcal{O}(\frac{1}{\epsilon})$ iterations. However, [29] does not discuss how such variance control would be obtained. Simple mini-batching, as we show, only leads to $\mathcal{O}(\frac{1}{\epsilon^4})$ stochastic oracle complexity.

We show that existing variants of the Halpern iteration [12, 51] can be effectively combined with recursive variance reduction [32] to obtain $\mathcal{O}(\frac{1}{\epsilon^3})$ stochastic oracle complexity in the cocoercive and Lipschitz monotone setups. We then show that the complexity can be further reduced to $\mathcal{O}\left(\frac{1}{\epsilon^2}\log(\frac{1}{\epsilon})\right)$ under an additional sharpness assumption about the operator. The last bound is unimprovable in terms of the dependence on $\epsilon$, due to existing lower bounds, as we argue for completeness in Appendix A.

To the best of our knowledge, our work is the first to use variance reduction to reduce stochastic oracle complexity of monotone inclusion (small gradient norm in min-max optimization settings), and the attained bounds are the best achieved to date for direct methods.

## 1.2 Techniques

Inspired by the potential function originally used by [12] and later used either in the same or slightly modified form by [13, 29, 51, 53], we adapt this potential function-based argument to account for stochastic error terms arising due to the stochastic oracle access to the operator. We first show that in the cocoercive minibatch setting, this argument only leads to $\mathcal{O}(\frac{1}{\epsilon^4})$ stochastic oracle complexity, and it is unclear how to improve it directly, as the analysis appears tight. We then combine the cocoercive variant of Halpern iteration [12] with the PAGE estimator [32] to reduce the stochastic oracle complexity to $\mathcal{O}(\frac{1}{\epsilon^3})$. The same variance reduced estimator is also used in conjunction with the two-step extrapolated variant of Halpern iteration introduced by [51], as a direct application of Halpern iteration is not known to converge on the class of Lipschitz monotone operators.

While the basic ideas in our arguments are simple, their realization requires addressing major technical obstacles. First, the variance reduced estimator that we use [32] was originally devised for smooth nonconvex optimization problems, where it was coupled with a stochastic variant of gradient descent. This is significant, because the proof relies on a descent lemma, which allows cancelling the error arising from the variance of the estimator by the "descent" part. Such an argument is not possible in our setting, as there is no objective function to descend on. Instead, our analysis relies on an intricate inductive argument that ensures that the expected norm of the operator is bounded in each iteration, assuming a suitable bound on the variance of the estimator. To obtain our desired result for the variance, we propose a data-dependent batch allocation in PAGE estimator [32] (see Corollary 2.2), which scales proportionally to the squared distance between successive iterates, similar to [3]. We inductively argue that the squared distance between successive iterates arising in the batch size of the estimator reduces at rate $\frac{1}{k^2}$ *in expectation*. This allows us to further certify that the estimators do not only remain accurate, but their variance decreases as $\mathcal{O}(\epsilon^2/k)$, where $k$ is the iteration count.

In the context of the potential function argument, unlike in the deterministic settings, we *do not* establish that the potential function is non-increasing, even in expectation. The stochastic error terms

that arise due to the stochastic nature of the operator evaluations are controlled by taking slightly smaller step sizes than in the vanilla methods from [12, 51], which allows us to "leak" negative quadratic terms that are further used in controlling the stochastic error. The argument for controlling the value of the potential function is itself coupled with the inductive argument for ensuring that the expected operator norm remains bounded.

Finally, while applying a restarting strategy is standard under sharpness conditions [46], obtaining the claimed stochastic oracle complexity result of $\mathcal{O}\big(\frac{1}{\epsilon^2} \log(\frac{1}{\epsilon})\big)$ requires a rather technical argument to bound the total number of stochastic queries to the operator.

### 1.3 Related work

**Monotone inclusion and variational inequalities.** Variational inequality problems were originally devised to deal with approximating equilibria. Their systematic study was initiated by [50]. The relationship between variational inequalities and min-max optimization was observed soon after [44], while one of the earliest papers to study solving monotone inclusion as a generalization of variational inequalities, convex and min-max optimization, and complementarity problems is [45]. For a historical overview of this area and an extensive review of classical results, see [17].

In the case of monotone operators, standard variants of variational inequality problems (see Section 2) and monotone inclusion are equivalent—their solution sets coincide. This is a consequence of the celebrated Minty Theorem [35]. However, there is a major difference between these problems when it comes to solving them to a finite accuracy. In particular, on unbounded domains, approximating variational inequalities is meaningless, whereas monotone inclusion remains well-defined. This is most readily seen from the observation that mapping from min-max optimization, variational inequalities correspond to primal-dual gap guarantees, while monotone inclusion corresponds to a guarantee in gradient norm. For a simple bilinear function $f(x, y) = xy$ which has the unique min-max solution at $(x, y) = (0, 0)$, the primal-dual gap is infinite for any point other than $(0, 0)$, while the gradient remains finite and is a good proxy for measuring quality of a solution. Further, even on bounded domains or using restricted gap functions on unbounded domains as in e.g., [37], optimal oracle complexity guarantees for approximate monotone inclusion imply optimal complexity guarantees for approximately satisfied variational inequalities (see, e.g., [12]). The opposite does not hold in general. In particular, in deterministic settings, standard algorithms such as the celebrated extragradient [28, 36], dual extrapolation [37], or Popov's method [42] that have the optimal oracle complexity $O(\frac{1}{\epsilon})$ for approximating variational inequalities are suboptimal for monotone inclusion and attain oracle complexity of the order $O(\frac{1}{\epsilon^2})$ [13, 20].

**Halpern iteration.** Halpern iteration is a classical fixed point iteration originally introduced by [21], and studied extensively in terms of both its asymptotic and non-asymptotic convergence guarantees [8, 26, 27, 31, 33, 52]. The first tight nonasymptotic convergence rate guarantee of $1/t$ was obtained in [33, 47]. This rate was also matched by [25].

The usefulness of Halpern iteration for solving monotone inclusion problems was first observed by [12],[1] who showed that its variants can be used to obtain near-optimal oracle complexity results for all standard classes of monotone inclusion problems with Lipschitz operators also studied in this work. The near-tightness (up to poly-logarithmic factors) of the results from [12] was certified using lower bound reductions from min-max optimization lower bounds introduced by [39]. These lower bounds were made tight for the cocoercive setup in [13].

The generalization of Halpern iteration from the cocoercive to Lipschitz monotone setup in [12] utilized approximating what is known as the resolvent operator, which led to a double-loop algorithm and an additional $\log(1/\epsilon)$ in the resulting complexity. This log factor was shaved off in [53], who introduced a two-step variant of Halpern iteration, inspired by the extragradient method of [28]. The results of [12, 53] were further extended to other classes of Lipschitz operators by [29, 51]. Except for [29] which considered controlled variance as discussed above, all of the existing results only targeted deterministic settings.

---

[1] Interestingly, the algorithm proposed by [25] for cocoercive inclusion coincides with the Halpern iteration for a related nonexpansive operator (see [9, Proposition 4.3])

**Stochastic settings and variance reduction.** Vanilla stochastic gradient methods have constant variance of stochastic gradients, which creates a bottleneck in the convergence rate. To improve the convergence rate, in the past decade, powerful variance reduction techniques have been proposed.

For strongly convex finite-sum problems, SAG [48], which used a biased stochastic estimator of the full gradient, was the first stochastic gradient method with a linear convergence rate. [24] and [11] improved [48] by proposing unbiased estimators of SVRG-type and SAGA-type, respectively. Such unbiased estimators were further combined with Nesterov acceleration [2, 49], or applied to nonconvex finite-sum/infinite-sum problems [30, 43]. For nonconvex stochastic (infinite-sum) problems, SARAH [38] and SPIDER [18, 54, 55] estimators were proposed to attain the optimal oracle complexity of $\mathcal{O}(1/\epsilon^3)$ for finding an $\epsilon$-approximate stationary point. Both estimators are referred to as "recursive" variance reduction estimators, as they are biased when taking expectation w.r.t. current randomness but unbiased w.r.t. all the randomness in history. PAGE [32] and STORM [10] significantly simplified SARAH and SPIDER in terms of reducing the number of loops and avoiding large minibatches, respectively. [3] further extended this line of work by incorporating second-order information and dynamic batch sizes.

In the setting of min-max optimization and variational inequalities/monotone inclusion, variance reduction has primarily been used for approximating variational inequalities, corresponding to the primal-dual gap in min-max optimization; see, for example [1, 5, 6, 23, 34, 40]. Under strong monotonicity (or sharpness in the case of [34]), such results generalize to monotone inclusion; however, to the best of our knowledge, there have been no results that address monotone inclusion under the weaker assumptions considered in this work. In the context of monotone inclusion with Lipschitz operators, the tightest complexity result that we are aware of is $\mathcal{O}(\frac{1}{\epsilon^4})$, due to [14], and it applies to a more general class of structured non-monotone Lipschitz operators, for the best iterate. The same oracle complexity can be deduced for the last iterate of a two-step variant of Halpern from [29, Theorem 6.1], using mini-batching. All the results in our work are also for the last iterate.

## 2 Preliminaries

We consider a real $d$-dimensional normed space $(\mathbb{R}^d, \|\cdot\|)$, where $\|\cdot\|$ is induced by an inner product associated with the space, i.e., $\|\cdot\| = \sqrt{\langle \cdot, \cdot \rangle}$. Let $\mathcal{U} \subseteq \mathbb{R}^d$ be closed and convex; in the unconstrained case, $\mathcal{U} \equiv \mathbb{R}^d$. When $\mathcal{U}$ is bounded, $D = \max_{\mathbf{u}, \mathbf{v} \in \mathcal{U}} \|\mathbf{u} - \mathbf{v}\|$ denotes its diameter.

**Classes of monotone operators.** We say that an operator $F : \mathbb{R}^d \to \mathbb{R}^d$ is

1. monotone, if $\forall \mathbf{u}, \mathbf{v} \in \mathbb{R}^d, \langle F(\mathbf{u}) - F(\mathbf{v}), \mathbf{u} - \mathbf{v} \rangle \geq 0$.

2. $L$-Lipschitz continuous for some $L > 0$, if $\forall \mathbf{u}, \mathbf{v} \in \mathbb{R}^d, \|F(\mathbf{u}) - F(\mathbf{v})\| \leq L \|\mathbf{u} - \mathbf{v}\|$.

3. $\gamma$-cocoercive for some $\gamma > 0$, if $\forall \mathbf{u}, \mathbf{v} \in \mathbb{R}^d, \langle F(\mathbf{u}) - F(\mathbf{v}), \mathbf{u} - \mathbf{v} \rangle \geq \gamma \|F(\mathbf{u}) - F(\mathbf{v})\|^2$.

4. $\mu$-strongly monotone for some $\mu > 0$, if $\forall \mathbf{u}, \mathbf{v} \in \mathbb{R}^d, \langle F(\mathbf{u}) - F(\mathbf{v}), \mathbf{u} - \mathbf{v} \rangle \geq \mu \|\mathbf{u} - \mathbf{v}\|^2$.

Note that we can easily specialize these definitions to the set $\mathcal{U}$ by restricting $\mathbf{u}, \mathbf{v}$ to be from $\mathcal{U}$.

Throughout the paper, the minimum assumption that we make about an operator $F$ is that it is monotone and Lipschitz. Observe that any $\gamma$-cocoercive operator is monotone and $\frac{1}{\gamma}$-Lipschitz. The converse to this statement does not hold in general.

**Monotone inclusion and variational inequalities.** Monotone inclusion asks for $\mathbf{u}^*$ such that

$$\mathbf{0} \in F(\mathbf{u}^*) + \partial I_{\mathcal{U}}(\mathbf{u}^*), \tag{MI}$$

where $I_{\mathcal{U}}$ is the indicator function of the set $\mathcal{U}$ and $\partial I_{\mathcal{U}}(\cdot)$ denotes the subdifferential of $I_{\mathcal{U}}$.

If $F$ is continuous and monotone, the solution set to (MI) is the same as the solution set of the Stampacchia Variational Inequality (SVI) problem, which asks for $\mathbf{u}^* \in \mathcal{U}$ such that

$$(\forall \mathbf{u} \in \mathcal{U}) : \quad \langle F(\mathbf{u}^*), \mathbf{u} - \mathbf{u}^* \rangle \geq 0. \tag{SVI}$$

Further, when $F$ is monotone, the solution set of (SVI) is equivalent to the solution set of the Minty Variational Inequality (MVI) problem consisting in finding $\mathbf{u}^*$ such that

$$(\forall \mathbf{u} \in \mathcal{U}) : \quad \langle F(\mathbf{u}), \mathbf{u}^* - \mathbf{u} \rangle \leq 0. \tag{MVI}$$

We assume throughout the paper that a solution to monotone inclusion (MI) exists, which implies that solutions to both (SVI) and (MVI) exist as well. Existence of solutions follows from standard results and is guaranteed whenever e.g., $\mathcal{U}$ is compact, or, if there exists a compact set $\mathcal{U}'$ such that $\mathrm{Id} - \frac{1}{L} F$ maps $\mathcal{U}'$ to itself, where $\mathrm{Id}$ is the identity map [17]. As remarked in the introduction, in unbounded setups it is generally not possible to approximate (MVI) and (SVI), whereas approximating (MI) is quite natural: we only need to find $\mathbf{u}$ such that $\mathbf{0} \in F(\mathbf{u}) + \partial I_{\mathcal{U}}(\mathbf{u}) + \mathcal{B}(\epsilon)$, where $\mathbf{0}$ denotes the zero vector and $\mathcal{B}(\epsilon)$ denotes the centered ball of radius $\epsilon$.

**Stochastic access to the operator.** We consider the stochastic setting for monotone inclusion problems. More specifically, we make the following assumptions for stochastic queries to $F$. These assumptions are made throughout the paper, without being explicitly invoked.

**Assumption 1** (Unbiased samples with bounded variance). For each query point $\mathbf{x} \in \mathcal{U}$, we observe $\widehat{F}(\mathbf{x}, z)$ where $z \sim P_z$ is a random variable that satisfies the following assumptions:

$$\mathbb{E}_z\big[\widehat{F}(\mathbf{x}, z)\big] = F(\mathbf{x}) \quad \text{and} \quad \mathbb{E}_z\big[\|\widehat{F}(\mathbf{x}, z) - F(\mathbf{x})\|^2\big] \leq \sigma^2.$$

**Assumption 2** (Multi-point oracle). We can query a set of points $(\mathbf{x}_1, \ldots, \mathbf{x}_n)$ and receive

$$\widehat{F}(\mathbf{x}_1, z), \ldots, \widehat{F}(\mathbf{x}_n, z) \quad \text{where} \quad z \sim P_z.$$

**Assumption 3** (Lipschitz in expectation). $\mathbb{E}_z\big[\|\widehat{F}(\mathbf{u}, z) - \widehat{F}(\mathbf{v}, z)\|^2\big] \leq L^2 \|\mathbf{u} - \mathbf{v}\|^2, \forall \mathbf{u}, \mathbf{v} \in \mathcal{U}$.

We note that complexity results of the paper will bound the total number of queries made to this oracle. In particular, if multiple query points and/or multiple samples $z$ are used in a single iteration, our complexity is given by the sum of all those queries throughout all iterations of the method. Also, Assumption 3 is primary with parameter $L$, by which $F$ is also $L$-Lipschitz using Jensen's inequality.

**PAGE variance-reduced estimator.** We now summarize a variant of the PAGE estimator, originally developed for smooth nonconvex optimization by [32], adapted to our setting. In particular, given queries to $\widehat{F}$, we define the variance reduced estimator $\widetilde{F}(\mathbf{u}_k)$ for $k \geq 1$ by

$$\widetilde{F}(\mathbf{u}_k) = \begin{cases} \frac{1}{S_1^{(k)}} \sum_{i=1}^{S_1^{(k)}} \widehat{F}(\mathbf{u}_k, z_i^{(k)}) & \text{w. p. } p_k, \\ \widetilde{F}(\mathbf{u}_{k-1}) + \frac{1}{S_2^{(k)}} \sum_{i=1}^{S_2^{(k)}} \left( \widehat{F}(\mathbf{u}_k, z_i^{(k)}) - \widehat{F}(\mathbf{u}_{k-1}, z_i^{(k)}) \right) & \text{w. p. } 1 - p_k, \end{cases} \tag{2.1}$$

where $p_0 = 1$, $z_i^{(k)} \overset{\text{i.i.d.}}{\sim} P_z$, and $S_1^{(k)}$ and $S_2^{(k)}$ are the sample sizes at iteration $k$. Observe that Assumption 2 guarantees that we can query $\widehat{F}$ at $\mathbf{u}_k$ and $\mathbf{u}_{k-1}$ using the same random seed. Our analysis will make use of conditional expectations, and to that end, we define natural filtration $\mathcal{F}_k$ by $\mathcal{F}_k := \sigma(\{\widetilde{F}(\mathbf{u}_j)\}_{j \leq k})$; namely $\mathcal{F}_k$ contains all the randomness that arises in the definitions of $\widetilde{F}(\mathbf{u}_j)$ for $j \leq k$. Following a similar argument as in [32], we recursively bound the variance of the estimator $\widetilde{F}$, as summarized in the following lemma. The proof is provided in Appendix B.

**Lemma 2.1.** *Let $F$ be a monotone operator accessed via stochastic queries $\widehat{F}$, under Assumptions 1–3. Then, the variance of $\widetilde{F}$ defined by Eq. (2.1) satisfies the following recursive bound: for all $k \geq 1$,*

$$\mathbb{E}[\|\widetilde{F}(\mathbf{u}_k) - F(\mathbf{u}_k)\|^2] \leq \frac{p_k \sigma^2}{S_1^{(k)}} + (1 - p_k)\Big( \mathbb{E}[\|\widetilde{F}(\mathbf{u}_{k-1}) - F(\mathbf{u}_{k-1})\|^2] + \mathbb{E}\Big[\frac{L^2 \|\mathbf{u}_k - \mathbf{u}_{k-1}\|^2}{S_2^{(k)}}\Big]\Big).$$

With the choices of $p_k, S_1^{(k)}, S_2^{(k)}$ specified in the following corollary and using induction with the inequality from Lemma 2.1, we obtain the following bound on the variance.

**Corollary 2.2.** *Given a target error $\epsilon > 0$, if for all $k \geq 1$, $p_k = \frac{2}{k+1}, S_1^{(k)} \geq \lceil \frac{8\sigma^2}{p_k \epsilon^2} \rceil, S_2^{(k)} \geq \lceil \frac{8L^2 \|\mathbf{u}_k - \mathbf{u}_{k-1}\|^2}{p_k^2 \epsilon^2} \rceil$, then $\mathbb{E}\big[\|\widetilde{F}(\mathbf{u}_k) - F(\mathbf{u}_k)\|^2\big] \leq \frac{\epsilon^2}{k}$.*

## 3 Stochastic Halpern iteration for cocoercive operators

In this section, we consider the setting of $\frac{1}{L}$-cocoercive operators $F$. While cocoercivity is a strong assumption that implies that an operator is both Lipschitz and monotone (as discussed in Section 2),

it is nevertheless the most basic setup for studying the Halpern iteration. In particular, while Halpern iteration can be applied directly to the nonexpansive counterpart of a cocoercive operator $F$ (i.e., to the linear transformation $\text{Id} - \frac{2}{L}F$, where $\frac{1}{L}$ is an upper bound on the cocoercivity parameter of $F$), convergence does not seem possible to establish for the more general class of Lipschitz monotone operators. We begin this section by providing a generic proof of stochastic oracle complexity, which we then use to briefly illustrate how to obtain $\mathcal{O}(\frac{1}{\epsilon^4})$ oracle complexity with a simple minibatch stochastic estimator of $F$. We then show how to improve this bound to $\mathcal{O}(\frac{1}{\epsilon^3})$ by applying the proposed variant of the PAGE estimator from Eq. (2.1) to Halpern iteration.

The stochastic variant of Halpern iteration that we consider is defined by

$$\mathbf{u}_{k+1} = \lambda_{k+1}\mathbf{u}_0 + (1 - \lambda_{k+1})\Big(\mathbf{u}_k - \frac{2}{L_{k+1}}\widetilde{F}(\mathbf{u}_k)\Big), \tag{3.1}$$

where $\widetilde{F}$ is a stochastic (possibly biased) estimator of $F$, $\lambda_{k+1} = \Theta(\frac{1}{k})$ is the step size, and $L_{k+1} \geq L$ is a parameter of the algorithm. Compared to the classical iteration $\mathbf{u}_{k+1} = \lambda_{k+1}\mathbf{u}_0 + (1 - \lambda_{k+1})T(\mathbf{u}_k)$, where $T : \mathbb{R}^d \to \mathbb{R}^d$ is a nonexpansive (1-Lipschitz) map [21], $T$ is replaced by the mapping $\text{Id} - \frac{2}{L_{k+1}}\widetilde{F}$, which is stochastic and may not be nonexpansive (as the stochastic estimate $\widetilde{F}$ of $F$ is not guaranteed to be cocoercive even when $F$ is). Compared to the iteration variant considered by [12], the access to the monotone operator is stochastic and we also take slightly larger (by a factor of 2) values of $L_{k+1}$ to bound the stochastic error terms.

Our argument for bounding the total number of stochastic queries to $F$ is based on the use of the following potential function $\mathcal{C}_k = \frac{A_k}{L_k}\|F(\mathbf{u}_k)\|^2 + B_k\langle F(\mathbf{u}_k), \mathbf{u}_k - \mathbf{u}_0\rangle$, where $\{A_k\}_{k\geq 1}$ and $\{B_k\}_{k\geq 1}$ are positive and non-decreasing sequences of real numbers, while the step size $\lambda_k$ is defined by $\lambda_k := \frac{B_k}{A_k + B_k}$. Such potential function was previously used for the deterministic case of Halpern iteration in [12, 13]. Observe that even though we make oracle queries to $\widehat{F}$, the potential function $\mathcal{C}_k$ and the final bound we obtain are in terms of the true operator value $F$.

Compared to the analysis of Halpern iteration in the deterministic case [12, 13], our analysis for the stochastic case needs to account for the error terms caused by accessing $F$ via stochastic queries and is based on an intricate inductive argument. A generic bound on iteration complexity, under mild assumptions about the estimator $\widetilde{F}$, is summarized in Theorem 3.1. The proof is in Appendix C.

**Theorem 3.1.** *Given an arbitrary $\mathbf{u}_0 \in \mathbb{R}^d$, suppose that iterates $\mathbf{u}_k$ evolve according to Halpern iteration from Eq. (3.1) for $k \geq 1$, where $L_k = 2L$ and $\lambda_k = \frac{1}{k+1}$. Assume further that the stochastic estimate $\widetilde{F}(\mathbf{u})$ is unbiased for $\mathbf{u} = \mathbf{u}_0$ and $\mathbb{E}[\|F(\mathbf{u}_0) - \widetilde{F}(\mathbf{u}_0)\|^2] \leq \frac{\epsilon^2}{8}$. Given $\epsilon > 0$, if for all $k \geq 1$, we have that $\mathbb{E}\big[\big\|F(\mathbf{u}_k) - \widetilde{F}(\mathbf{u}_k)\big\|^2\big] \leq \frac{\epsilon^2}{k}$, then for all $k \geq 1$,*

$$\mathbb{E}[\|F(\mathbf{u}_k))\|] \leq \frac{\Lambda_0}{k} + \Lambda_1\epsilon, \tag{3.2}$$

*where $\Lambda_0 = 76L\|\mathbf{u}_0 - \mathbf{u}^*\|$ and $\Lambda_1 = 4\sqrt{\frac{2}{3}}$. As a result, stochastic Halpern iteration from Eq. (3.1) returns a point $\mathbf{u}_k$ such that $\mathbb{E}[\|F(\mathbf{u}_k)\|] \leq 4\epsilon$ after at most $N = \lceil\frac{2\Lambda_0}{\epsilon}\rceil = \mathcal{O}\big(\frac{L\|\mathbf{u}_0 - \mathbf{u}^*\|}{\epsilon}\big)$ iterations.*

We remark that the previous result states an iteration complexity bound under a rather high accuracy assumption for the operator estimators at each iteration. In order to attain these accuracy requirements, we could either use a minibatch at every iteration, or use variance reduction. In what follows we explore both approaches. We further remark that we made no effort to optimize the constants in the bound above, and thus the constants are likely improvable.

Finally, observe that due to the required low error for the estimates $\mathbb{E}[\|F(\mathbf{u}_k) - \widetilde{F}(\mathbf{u}_k)\|^2] \leq \frac{\epsilon^2}{k}$, we can certify by Chebyshev bound that $\mathbb{P}[\|F(\mathbf{u}_k) - \widetilde{F}(\mathbf{u}_k)\| \geq \epsilon] \leq \frac{1}{k}$. In particular, after $O(\frac{1}{\epsilon})$ iterations, if we have $\|\widetilde{F}(\mathbf{u}_k)\| \leq \epsilon$ (which holds in expectation), then $\|F(\mathbf{u}_k)\|$ is also $O(\epsilon)$ with probability at least $1 - \epsilon$. This is particularly important for practical implementations, where a stopping criterion can be based on the value of $\|\widetilde{F}(\mathbf{u}_k)\|$, which, unlike $\|F(\mathbf{u}_k)\|$, can be efficiently evaluated.

## 3.1 Stochastic oracle complexity with a simple mini-batch estimate

A direct consequence of Theorem 3.1 is that a simple estimator $\widetilde{F}(\mathbf{u}_k) = \frac{1}{S_k}\sum_{i=1}^{S_k}\widehat{F}(\mathbf{u}_k, z_i^{(k)})$ leads to the overall $\mathcal{O}(\frac{1}{\epsilon^4})$ oracle complexity, as stated below while the proof is deferred to Appendix C.

**Corollary 3.2.** *Under the assumptions of Theorem 3.1, if $\widetilde{F}(\mathbf{u}_k) = \frac{1}{S_k}\sum_{i=1}^{S_k}\widehat{F}(\mathbf{u}_k, z_i^{(k)})$, where $\widehat{F}(\mathbf{u}_k, z_i^{(k)})$ satisfies Assumption 1 and $z_i^{(k)} \overset{i.i.d.}{\sim} P_z$, then setting $S_k = \frac{\sigma^2(k+1)}{\epsilon^2}$ for all $k \geq 0$ guarantees that $\mathbb{E}[\|F(\mathbf{u}_k)\|] \leq 4\epsilon$ after at most $\mathcal{O}\left(\frac{\sigma^2 L^2 \|\mathbf{u}_0 - \mathbf{u}^*\|^2}{\epsilon^4}\right)$ queries to $\widehat{F}$.*

## 3.2 Improved oracle complexity via variance reduction

We now consider using the recursive variance reduction method from Eq. (2.1) to obtain the variance bound required in Theorem 3.1, as summarized in Algorithm 1. Of course, in practice, $\|\mathbf{u}_0 - \mathbf{u}^*\|$ is not known, and instead of running the algorithm for a fixed number of iterations $N$, one could run it, for example, until reaching a point with $\|\widetilde{F}(\mathbf{u}_k)\| \leq \epsilon$.

---

**Algorithm 1:** Stochastic Halpern-Cocoercive (Halpern)

**Input:** $\mathbf{u}_0 \in \mathbb{R}^d$, $\|\mathbf{u}_0 - \mathbf{u}^*\|$, $L$, $\epsilon > 0$, $\sigma$;

**Initialize:** $\Lambda_0 = \frac{76L\|\mathbf{u}_0 - \mathbf{u}^*\|}{\epsilon}$, $N = \lceil\frac{2\Lambda_0}{\epsilon}\rceil$, $S_1^{(0)} = \lceil\frac{8\sigma^2}{\epsilon^2}\rceil$, $\widetilde{F}(\mathbf{u}_0) = \frac{1}{S_1^{(0)}}\sum_{i=1}^{S_1^{(0)}}\widehat{F}(\mathbf{u}_0, z_i^{(0)})$;

**for** $k = 1:N$ **do**

    $\mathbf{u}_k = \frac{1}{k+1}\mathbf{u}_0 + \frac{k}{k+1}\left(\mathbf{u}_{k-1} - \frac{1}{L}\widetilde{F}(\mathbf{u}_{k-1})\right)$;

    $p_k = \frac{2}{k+1}$, $S_1^{(k)} = \lceil\frac{8\sigma^2}{p_k\epsilon^2}\rceil$, $S_2^{(k)} = \lceil\frac{8L^2\|\mathbf{u}_k - \mathbf{u}_{k-1}\|^2}{p_k{}^2\epsilon^2}\rceil$;

    Compute $\widetilde{F}(\mathbf{u}_k)$ based on Eq. (2.1)

**Return:** $\mathbf{u}_N$

---

Notice that convergence is guaranteed by Theorem 3.1; however it does not directly address the problem of the oracle complexity (as batch sizes depend on successive iterate distances). To resolve this issue, we first provide a bound on $\|\mathbf{u}_k - \mathbf{u}_{k-1}\|$ as in Lemma 3.3, while the proof is deferred to Appendix C, and make the appropriate parameter settings for the estimator from Eq. (2.1). It is now possible to apply Theorem 3.1 to obtain the improved $\mathcal{O}(\frac{1}{\epsilon^3})$ stochastic oracle complexity bound.

**Lemma 3.3.** *Given an arbitrary initial point $\mathbf{u}_0 \in \mathbb{R}^d$, let $\{\mathbf{u}_k\}_{k\geq 1}$ be the sequence of points produced by Algorithm 1. Assume further that $\lambda_k = \frac{1}{k+1}$, $L_k = 2L$ for all $k \geq 0$. Then,*

$$\|\mathbf{u}_k - \mathbf{u}_{k-1}\|^2 \leq \begin{cases} \frac{1}{4L^2}\|\widetilde{F}(\mathbf{u}_0)\|^2 & \text{if } k = 1, \\ \frac{2k^2}{L^2(k+1)^2}\|\widetilde{F}(\mathbf{u}_{k-1})\|^2 + \sum_{i=0}^{k-2}\frac{2(i+1)^2}{k(k+1)^2L^2}\|\widetilde{F}(\mathbf{u}_i)\|^2 & \text{if } k \geq 2. \end{cases} \tag{3.3}$$

*Moreover, if for $1 \leq i \leq k-1$, all of the following conditions hold (same as in Theorem 3.1): (i) $\mathbb{E}[\|F(\mathbf{u}_i)\|] \leq \frac{\Lambda_0}{i} + \Lambda_1\epsilon$, where $\Lambda_0 = 76L\|\mathbf{u}_0 - \mathbf{u}^*\|$ and $\Lambda_1 = 4\sqrt{\frac{2}{3}}$, (ii) $\mathbb{E}\left[\left\|F(\mathbf{u}_i) - \widetilde{F}(\mathbf{u}_i)\right\|^2\right] \leq \frac{\epsilon^2}{i}$, and (iii) $\epsilon \leq \frac{\Lambda_0}{k}$, then $\mathbb{E}[\|\mathbf{u}_k - \mathbf{u}_{k-1}\|^2] = \mathcal{O}\left(\frac{\|\mathbf{u}_0 - \mathbf{u}^*\|^2}{k^2}\right)$.*

**Corollary 3.4.** *Given arbitrary $\mathbf{u}_0 \in \mathbb{R}^d$ and $\epsilon > 0$, consider $\mathbf{u}_N$ returned by Algorithm 1. Then, $\mathbb{E}[\|F(\mathbf{u}_N)\|] \leq 4\epsilon$ with expected $\mathcal{O}\left(\frac{\sigma^2 L\|\mathbf{u}_0 - \mathbf{u}^*\| + L^3\|\mathbf{u}_0 - \mathbf{u}^*\|^3}{\epsilon^3}\right)$ oracle queries to $\widehat{F}$.*

*Proof.* Let $m_k$ be the number of stochastic queries made by the estimator from Eq. (2.1) in iteration $k$. Using Corollary 2.2, we have

$$\mathbb{E}[m_{k+1}|\mathcal{F}_{k-1}] = p_k S_1^{(k)} + 2(1-p_k)S_2^{(k)} = p_k\lceil\tfrac{8\sigma^2}{p_k\epsilon^2}\rceil + 2(1-p_k)\lceil\tfrac{8L^2\|\mathbf{u}_k - \mathbf{u}_{k-1}\|^2}{p_k^2\epsilon^2}\rceil,$$

where the first equality holds because $S_2^{(k)}$ is measurable w.r.t. $\mathcal{F}_{k-1}$ and the only random choice that remains is the selection of the estimator in Eq. (2.1) determined by probabilities $p_k$ and $1 - p_k$.

Taking expectation with respect to all randomness on both sides, rearranging the terms, and using the fact that $\lceil x \rceil \leq x + 1$ for any $x \in \mathbb{R}$, we obtain $\mathbb{E}[m_{k+1}] \leq \frac{8\sigma^2}{\epsilon^2} + \frac{16(1-p_k)L^2\mathbb{E}[\|\mathbf{u}_k - \mathbf{u}_{k-1}\|^2]}{p_k^2\epsilon^2} + 2$. Recalling that $p_k = \frac{2}{k+1} = \mathcal{O}(\frac{1}{k})$ and $\mathbb{E}[\|\mathbf{u}_k - \mathbf{u}_{k-1}\|^2] = \mathcal{O}(\frac{\|\mathbf{u}_0 - \mathbf{u}^*\|^2}{k^2})$ by Lemma 3.3, it follows that $\mathbb{E}[m_{k+1}] = \mathcal{O}(\frac{\sigma^2 + L^2\|\mathbf{u}_0 - \mathbf{u}^*\|^2}{\epsilon^2})$. As, by Theorem 3.1, the total number of iterations to attain $4\epsilon$ norm of the operator in expectation is $N = \lceil \frac{2\Lambda_0}{\epsilon} \rceil = \mathcal{O}(\frac{L\|\mathbf{u}_0 - \mathbf{u}^*\|}{\epsilon})$ and $m_0 = S_1^{(0)} = \mathcal{O}(\frac{\sigma^2}{\epsilon^2})$, the total number of queries to $\widehat{F}$ is $\mathbb{E}[M] = \mathbb{E}[\sum_{k=1}^{N} m_k] = \mathcal{O}(\frac{\sigma^2 L\|\mathbf{u}_0 - \mathbf{u}^*\| + L^3\|\mathbf{u}_0 - \mathbf{u}^*\|^3}{\epsilon^3})$. □

We note in passing that the running time guarantee of this algorithm is of Las Vegas-type: despite its iteration number being surely bounded by $\lceil \frac{2\Lambda_0}{\epsilon} \rceil = \mathcal{O}(\frac{L\|\mathbf{u}_0 - \mathbf{u}^*\|}{\epsilon})$, the batch sizes (in particular $S_2^{(k)}$) are random, and are not universally bounded.

We further argue that Algorithm 1 can be extended to **constrained settings** by defining the operator mapping as in [12] and modifying the variance-reduced stochastic estimator accordingly based on the projection of $\widetilde{F}$. We show that the newly defined operator mapping is also cocoercive while the variance of the modified estimator is bounded by the variance of $\widetilde{F}$, so arguments from Theorem 3.1 and Corollary 3.4 extend to this case. This modified estimator need not be unbiased (as neither is $\widetilde{F}$); however, this is irrelevant to our analysis as it does not require unbiasedness. For completeness, a detailed extension to the constrained case is provided in Appendix C.2.

## 4 Monotone and Lipschitz setup

Throughout this section, we assume that $F$ is monotone and $L$-Lipschitz. While the previous section addresses the cocoercive setup using the classical version of Halpern iteration adapted to cocoercive operators, it is unclear how to directly generalize this result to the setting with monotone Lipschitz operators. In the deterministic setting, generalization to monotone Lipschitz operators can be achieved through the use of a resolvent operator (see [12]). However, such an approach incurs an additional $\log(1/\epsilon)$ factor in the iteration complexity coming from approximating the resolvent and it is further unclear how to generalize it to stochastic settings, as the properties of the stochastic estimate $\widetilde{F}$ of $F$ do not readily translate into the same or similar properties for the resolvent of $\widetilde{F}$. Instead of taking the approach based on the resolvent, we consider a recently proposed two-step variant of Halpern iteration [51], adapted here to the stochastic setting. The variant uses extrapolation and is defined by

$$\begin{cases} \mathbf{v}_k & := \lambda_k \mathbf{u}_0 + (1 - \lambda_k)\mathbf{u}_k - \eta_k \widetilde{F}(\mathbf{v}_{k-1}), \\ \mathbf{u}_{k+1} := \lambda_k \mathbf{u}_0 + (1 - \lambda_k)\mathbf{u}_k - \eta_k \widetilde{F}(\mathbf{v}_k), \end{cases} \quad (4.1)$$

where $\lambda_k \in [0, 1)$, $\eta_k > 0$, and $\widetilde{F}$ is defined by (2.1). The resulting algorithm with a complete parameter setting is provided in Algorithm 2.

To analyze the convergence of the extrapolated Halpern variant from Eq. (4.1), we use the potential function $\mathcal{V}_k = A_k\|F(\mathbf{u}_k)\|^2 + B_k\langle F(\mathbf{u}_k), \mathbf{u}_k - \mathbf{u}_0\rangle + c_k L^2\|\mathbf{u}_k - \mathbf{v}_{k-1}\|^2$, previously used by [51], where $A_k$, $B_k$ and $c_k$ are positive parameters to be determined later. Observe that this is essentially the same potential function as $\mathcal{C}_k$, corrected by the quadratic term $c_k L^2\|\mathbf{u}_k - \mathbf{v}_{k-1}\|^2$ to account for error terms appearing in the analysis of the two-step variant from Eq. (4.1). Similarly as in the cocoercive setup, the potential function is not monotonically non-increasing, due to the error terms that arise due to the stochastic access to $F$. Bounding these error terms requires a careful technical argument, and is the main technical contribution of this section. Due to space constraints, the complete technical argument is deferred to Appendix D, while the main results are stated below.

**Theorem 4.1.** *Given an arbitrary initial point $\mathbf{u}_0 \in \mathbb{R}^d$ and target error $\epsilon > 0$, assume that the iterates $\mathbf{u}_k$ evolve according to Algorithm 2 for $k \geq 1$. Then, for all $k \geq 2$,*

$$\mathbb{E}\left[\|F(\mathbf{u}_k)\|^2 + 2L^2\|\mathbf{u}_k - \mathbf{v}_{k-1}\|^2\right] \leq \frac{\Lambda_0}{(k+1)(k+2)} + \Lambda_1\epsilon^2, \quad (4.2)$$

*where $\Lambda_0 = \frac{4(L^2\eta_0\underline{\eta} + 1)\|\mathbf{u}_0 - \mathbf{u}^*\|^2}{\underline{\eta}^2}$ and $\Lambda_1 = \frac{5(1 + M\eta_0)}{M\underline{\eta}^2}$. In particular, $\mathbb{E}\left[\|F(\mathbf{u}_N)\|^2 + 2L^2\|\mathbf{u}_N - \mathbf{v}_{N-1}\|^2\right] \leq 2\Lambda_1\epsilon^2 = \mathcal{O}(\epsilon^2)$ after at most $N = \lceil \frac{\sqrt{\Lambda_0}}{\sqrt{\Lambda_1}\epsilon} \rceil = \mathcal{O}(\frac{L\|\mathbf{u}_0 - \mathbf{u}^*\|}{\epsilon})$ iterations. The total number of oracle queries to $\widehat{F}$ is $\mathcal{O}(\frac{\sigma^2 L\|\mathbf{u}_0 - \mathbf{u}^*\| + L^3\|\mathbf{u}_0 - \mathbf{u}^*\|^3}{\epsilon^3})$ in expectation.*

**Algorithm 2:** Extrapolated Stochastic Halpern-Monotone (E-Halpern)

---

**Input:** $\mathbf{u}_0 \in \mathbb{R}^d$, $\|\mathbf{u}_0 - \mathbf{u}^*\|$, $0 < \eta_0 \leq \frac{1}{3\sqrt{3}L}$, $L, \epsilon > 0, \sigma$;

**Initialize:** $\mathbf{v}_{-1} = \mathbf{u}_0$, $S_1^{(-1)} = S_1^{(0)} = \lceil \frac{8\sigma^2}{\epsilon^2} \rceil$, $M = 9L^2$, $\underline{\eta} = \frac{\eta_0(1-2M\eta_0^2)}{1-M\eta_0^2}$;

Set $\Lambda_0 = \frac{4(L^2\eta_0\underline{\eta}+1)\|\mathbf{u}_0-\mathbf{u}^*\|^2}{\underline{\eta}^2}$, $\Lambda_1 = \frac{5(1+M\underline{\eta}\eta_0)}{M\underline{\eta}^2}$, $N = \lceil \frac{\sqrt{\Lambda_0}}{\sqrt{\Lambda_1}\epsilon} \rceil$;

$\widetilde{F}(\mathbf{v}_{-1}) = \frac{1}{S_1^{(-1)}} \sum_{i=1}^{S_1^{(-1)}} \widehat{F}(\mathbf{v}_{-1}, z_i^{(-1)})$, where $z_i^{(-1)} \stackrel{\text{i.i.d.}}{\sim} \mathcal{P}_z$;

**for** $k = 1 : N$ **do**

$\quad$ $\mathbf{v}_{k-1} = \frac{1}{k+1}\mathbf{u}_0 + \frac{k}{k+1}\mathbf{u}_{k-1} - \eta_{k-1}\widetilde{F}(\mathbf{v}_{k-2})$;

$\quad$ $p_{k-1} = \min(\frac{2}{k}, 1)$, $S_1^{(k-1)} = \lceil \frac{8\sigma^2}{p_{k-1}\epsilon^2} \rceil$, $S_2^{(k-1)} = \lceil \frac{8L^2\|\mathbf{v}_{k-1}-\mathbf{v}_{k-2}\|^2}{p_{k-1}^2\epsilon^2} \rceil$;

$\quad$ Compute $\widetilde{F}(\mathbf{v}_{k-1})$ based on Eq. (2.1);

$\quad$ $\mathbf{u}_k = \frac{1}{k+1}\mathbf{u}_0 + \frac{k}{k+1}\mathbf{u}_{k-1} - \eta_{k-1}\widetilde{F}(\mathbf{v}_{k-1})$;

$\quad$ $\eta_k = \frac{(1-\frac{1}{(k+1)^2}-M\eta_{k-1}^2)(k+1)^2}{(1-M\eta_{k-1}^2)k(k+2)}\eta_{k-1}$

**Return:** $\mathbf{u}_N$

---

## 5 Faster convergence under a sharpness condition

We now show that by restarting Algorithm 2, we can achieve the $\mathcal{O}\left(\frac{1}{\epsilon^2}\log\frac{1}{\epsilon}\right)$ oracle complexity under a milder than strong monotonicity $\mu$-*sharpness condition*: for all $\mathbf{u} \in \mathcal{U}$, $\langle F(\mathbf{u}) - F(\mathbf{u}^*), \mathbf{u} - \mathbf{u}^* \rangle \geq \mu\|\mathbf{u} - \mathbf{u}^*\|^2$. The scheme is summarized in Algorithm 3, and the proof is deferred to Appendix E.

---

**Algorithm 3:** Restarted Extrapolated Stochastic Halpern-Sharp (Restarted E-Halpern)

---

**Input:** $\mathbf{v}_{-1} = \mathbf{u}_0 \in \mathbb{R}^d$, $\|\mathbf{u}_0 - \mathbf{u}^*\|$, $0 < \eta_0 \leq \frac{1}{3\sqrt{3}L}$, $L, \mu, \epsilon > 0, \sigma$;

**Initialize:** $M = 9L^2$, $\underline{\eta} = \frac{\eta_0(1-2M\eta_0^2)}{1-M\eta_0^2}$, $N = \left\lceil \log\left(\frac{\sqrt{6}\|\mathbf{u}_0-\mathbf{u}^*\|}{2\epsilon}\right) \right\rceil$;

**for** $k = 1 : N$ **do**

$\quad$ Call Algorithm 2 with initialization $\mathbf{v}_{-1}^{(k)} = \mathbf{u}_0^{(k)} = \mathbf{u}_{k-1}$, $\epsilon_k = \frac{\mu\epsilon\sqrt{M\underline{\eta}^2}}{2\sqrt{5(1+M\underline{\eta}\eta_0)}}$, and

$\quad$ $S_1^{(-1)} = S_1^{(0)} = \lceil \frac{8\sigma^2}{\epsilon_k^2} \rceil$, for $K = \left\lceil \frac{4\sqrt{L^2\eta_0\underline{\eta}+1}}{\mu\underline{\eta}} \right\rceil$ iterations, and return $\mathbf{u}_k$;

**Return:** $\mathbf{u}_N$

---

**Theorem 5.1.** *Given $L$-Lipschitz and $\mu$-sharp $F$ and the precision parameter $\epsilon$, Algorithm 3 outputs $\mathbf{u}_N$ with $\mathbb{E}[\|\mathbf{u}_N - \mathbf{u}^*\|^2] \leq \epsilon^2$ as well as $\mathbb{E}[\|F(\mathbf{u}_N)\|^2] \leq L^2\epsilon^2$ after $N = \mathcal{O}\left(\frac{L}{\mu}\log\frac{\|\mathbf{u}_0-\mathbf{u}^*\|}{\epsilon}\right)$ iterations with at most $\mathcal{O}\left(\frac{\sigma^2(\mu+L)\log(\|\mathbf{u}_0-\mathbf{u}^*\|/\epsilon)+L^3\|\mathbf{u}_0-\mathbf{u}^*\|^2}{\mu^3\epsilon^2}\right)$ queries to $\widehat{F}$ in expectation.*

## 6 Numerical experiments and discussion

We now illustrate the empirical performance of stochastic Halpern iteration on robust least square problems. Specifically, given data matrix $\mathbf{A} \in \mathbb{R}^{n \times d}$ and noisy observation vector $\mathbf{b} \in \mathbb{R}^n$ subject to bounded deterministic perturbation $\delta$ with $\|\delta\| \leq \rho$, robust least square (RLS) minimizes the worst-case residue as $\min_{\mathbf{x}\in\mathbb{R}^d} \max_{\delta:\|\delta\|\leq\rho} \|\mathbf{A}\mathbf{x} - \mathbf{y}\|_2^2$ with $\mathbf{y} = \mathbf{b} + \delta$ [16]. We consider solving MI induced from RLS with Lagrangian relaxation where $\mathbf{u} = (\mathbf{x}, \mathbf{y})^T$ and $F(\mathbf{u}) = (\nabla_\mathbf{x}L_\lambda(\mathbf{x}, \mathbf{y}), -\nabla_\mathbf{y}L_\lambda(\mathbf{x}, \mathbf{y}))^T$ for $L_\lambda(\mathbf{x}, \mathbf{y}) = \frac{1}{2n}\|A\mathbf{x} - \mathbf{y}\|_2^2 - \frac{\lambda}{2n}\|\mathbf{y} - \mathbf{b}\|_2^2$. We use a real-world superconductivity dataset [22] from UCI Machine Learning Repository [15] for our experiment, which is of size $21263 \times 81$. To ensure the problem is concave in $\mathbf{y}$, we need that $\lambda > 1$; in the experiments, we set $\lambda = 1.5$. For the experiment, we compare Halpern, E-Halpern, and Restarted E-Halpern algorithms with gradient descent-ascent (GDA), extragradient (EG) [28], and Popov's method [42] in stochastic settings. Even though our theoretical results for Restarted

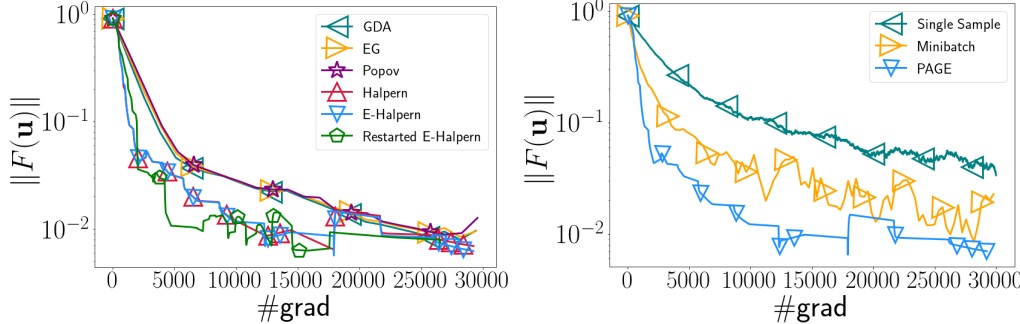

(a) Comparison on superconductivity dataset.    (b) E-Halpern with different stochastic estimators.

Figure 1: Empirical comparison of min-max algorithms on the robust least squares problem.

E-Halpern require scheduled restarts based on known problem parameters, in the implementation, to avoid complicated parameter tuning and illustrate empirical performance, we restart E-Halpern whenever the norm of stochastic estimator $\widetilde{F}$ used in E-Halpern halves. All Halpern variants are implemented with PAGE estimator considered in our paper; all other algorithms are implemented using minibatches. Additionally, we compare E-Halpern with the PAGE estimator against E-Halpern with single-sample and mini-batch estimators.

We plot the (empirical) operator norm $\|F(\mathbf{u})\|$ against the number of stochastic operator evaluations. Note that evaluations of $\|F(\mathbf{u})\|$ are only used for plotting but not for running any of the algorithms. We use the same random initialization and tune the batch sizes and step sizes (to the values achieving fastest convergence under noise) for each method by grid search. We use constant batch sizes and constant step sizes for GDA, EG, and Popov. We also choose the batch sizes of PAGE estimator to ensure $\mathbb{E}[\|F(\mathbf{u}_k) - \widetilde{F}(\mathbf{u}_k)\|^2] \leq \mathcal{O}(\frac{1}{k})$, which handles error accumulation [29] and early stagnation of stochastic Halpern iteration. We implement all the algorithms in Python and run each algorithm using one CPU core on a macOS machine with Intel 2.3GHz Dual Core i5 Processor and 8GB RAM.[2]

We observe that (i) in Figure 1(a) both Halpern and E-Halpern exhibit faster convergence to approximate stationary points (with much smaller gradient norm after same number of gradient evaluations) than other algorithms, and restarting E-Halpern provides additional speedup, validating our theoretical insights; (ii) in Figure 1(b), E-Halpern with PAGE estimator displays faster convergence compared to other two estimators, in agreement with our theoretical analysis.

## 7    Conclusion

We introduced stochastic variance reduced variants of Halpern iteration for addressing monotone inclusion problems. Our work addresses all standard classes of Lipschitz monotone problems and achieves improved stochastic oracle complexity guarantees, all for the last iterate. Subsequent to this work, [7] obtained near-optimal bounds for the cases considered in this work, by reducing the Lipschitz monotone case to the Lipschitz strongly monotone case, using regularization. It is an open question to obtain such near-optimal bounds with a direct method, without the use of regularization.

## Acknowledgements

XC and CS were supported in part by the NSF grant 2023239. CG's research was partially supported by INRIA Associate Teams project, FONDECYT 1210362 grant, ANID Anillo ACT210005 grant, and National Center for Artificial Intelligence CENIA FB210017, Basal ANID. Part of this work was done while CG was at the University of Twente. JD was supported by the NSF grant 2007757, by the Office of Naval Research under contract number N00014-22-1-2348, and by the Wisconsin Alumni Research Foundation. Part of this work was done while JD and CS were visiting Simons Institute for the Theory of Computing.

---

[2]Code is available at https://github.com/zephyr-cai/Halpern.

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
