## A  (Near) Tightness of stochastic oracle complexity bounds for the sharp case

In this section, we briefly discuss lower bound reductions which imply that our results for Lipschitz sharp setups are unimprovable in terms of the dependence on $\epsilon$. To keep the discussion simple, we only focus on the $\epsilon$ dependence here and unconstrained settings. The near-optimality of our bounds is implied by the known lower bound for the optimality gap in $L$-smooth $\mu$-strongly convex stochastic optimization, which is of the order $\Omega(\frac{\sigma^2}{\mu\epsilon})$ in the high noise $\sigma^2$ or low error $\epsilon$ regimes; see, for example, the discussion in [19] (the omitted part of the lower bound comes from the deterministic complexity of smooth strongly convex optimization and is less interesting in our context). The same lower bound implies a lower bound of $\Omega(\frac{\sigma^2}{\epsilon^2})$ for minimizing the gradient of a smooth strongly convex function $f$. Suppose not (for the purpose of contradiction); i.e., suppose that there were an algorithm that constructs a point $\mathbf{x}$ with $\mathbb{E}[\|\nabla f(\mathbf{x})\|^2] \leq \bar{\epsilon}^2$ in $o(\frac{\sigma^2}{\bar{\epsilon}^2})$ oracle queries to the stochastic gradient. By $\mu$-strong convexity of $f$, this would imply that we get $\mathbb{E}[f(\mathbf{x}) - \min_{\mathbf{u}} f(\mathbf{u})] \leq \frac{1}{2\mu}\mathbb{E}[\|\nabla f(\mathbf{x})\|^2] \leq \frac{\bar{\epsilon}^2}{2\mu}$ with $o(\frac{\sigma^2}{\bar{\epsilon}^2})$ oracle queries to the stochastic gradient. Setting $\bar{\epsilon} = \sqrt{\epsilon\mu}$, we get that this would imply oracle complexity $o(\frac{\sigma^2}{\mu\epsilon})$, and we reach a contradiction on the lower bound for the optimality gap.

Hence, $\Omega(\frac{\sigma^2}{\epsilon^2})$ lower bound applies to the minimization of the gradient of smooth strongly convex functions in stochastic regimes. Observe that the gradients of smooth strongly convex functions are Lipschitz and strongly monotone (thus also sharp), so a lower bound for this problem class implies a lower bound for the class of sharp Lipschitz monotone inclusion problems. Thus, we can conclude that our result from Section 5 for sharp Lipschitz monotone inclusion problems that gives $\mathcal{O}\left(\frac{\sigma^2(\mu+L)\log(\|\mathbf{u}_0-\mathbf{u}^*\|/\epsilon)+L^3\|\mathbf{u}_0-\mathbf{u}^*\|^2}{\mu^3\epsilon^2}\right)$ stochastic oracle complexity is near-optimal in terms of the dependence on $\sigma$ and $\epsilon$ (but suboptimal in terms of the dependence on the remaining problem parameters, due to [7, 34]).

## B  Omitted proofs from Section 2

**Lemma 2.1.** *Let $F$ be a monotone operator accessed via stochastic queries $\widehat{F}$, under Assumptions 1–3. Then, the variance of $\widetilde{F}$ defined by Eq. (2.1) satisfies the following recursive bound: for all $k \geq 1$,*

$$\mathbb{E}[\|\widetilde{F}(\mathbf{u}_k) - F(\mathbf{u}_k)\|^2] \leq \frac{p_k\sigma^2}{S_1^{(k)}} + (1-p_k)\left(\mathbb{E}[\|\widetilde{F}(\mathbf{u}_{k-1}) - F(\mathbf{u}_{k-1})\|^2] + \mathbb{E}\left[\frac{L^2\|\mathbf{u}_k - \mathbf{u}_{k-1}\|^2}{S_2^{(k)}}\right]\right).$$

*Proof.* Using the definition of $\widetilde{F}$, conditional on $\mathcal{F}_{k-1}$, we have for all $k \geq 1$

$$\mathbb{E}\left[\left\|\widetilde{F}(\mathbf{u}_k) - F(\mathbf{u}_k)\right\|^2 \Big| \mathcal{F}_{k-1}\right]$$

$$= p_k\mathbb{E}\left[\left\|\frac{1}{S_1^{(k)}}\sum_{i=1}^{S_1^{(k)}} \widehat{F}(\mathbf{u}_k, z_i^{(k)}) - F(\mathbf{u}_k)\right\|^2 \Big| \mathcal{F}_{k-1}\right]$$

$$+ (1-p_k)\mathbb{E}\left[\left\|\widetilde{F}(\mathbf{u}_{k-1}) + \frac{1}{S_2^{(k)}}\sum_{i=1}^{S_2^{(k)}} \left(\widehat{F}(\mathbf{u}_k, z_i^{(k)}) - \widehat{F}(\mathbf{u}_{k-1}, z_i^{(k)})\right) - F(\mathbf{u}_k)\right\|^2 \Big| \mathcal{F}_{k-1}\right],$$

where $\mathcal{F}_{k-1} = \sigma(\{\widetilde{F}(\mathbf{u}_j)\}_{j \le k-1})$ is the natural filtration, as defined in Section 2. Note that both $\mathbf{u}_{k-1} \in \mathcal{F}_{k-1}$ and $\mathbf{u}_k \in \mathcal{F}_{k-1}$ by the updating scheme considered in this paper, so we have

$$
\mathbb{E}\left[\left\|\widetilde{F}(\mathbf{u}_k) - F(\mathbf{u}_k)\right\|^2 \Big| \mathcal{F}_{k-1}\right]
$$

$$
= p_k \underbrace{\mathbb{E}_{z^{(k)}}\left[\left\|\frac{1}{S_1^{(k)}} \sum_{i=1}^{S_1^{(k)}} \widehat{F}(\mathbf{u}_k, z_i^{(k)}) - F(\mathbf{u}_k)\right\|^2\right]}_{\mathcal{T}_1}
$$

$$
+ (1-p_k) \underbrace{\mathbb{E}_{z^{(k)}}\left[\left\|\widetilde{F}(\mathbf{u}_{k-1}) + \frac{1}{S_2^{(k)}} \sum_{i=1}^{S_2^{(k)}} \left(\widehat{F}(\mathbf{u}_k, z_i^{(k)}) - \widehat{F}(\mathbf{u}_{k-1}, z_i^{(k)})\right) - F(\mathbf{u}_k)\right\|^2\right]}_{\mathcal{T}_2}.
$$

(B.1)

Here we use $\mathbb{E}_{z^{(k)}}$ to denote taking expectation with respect to the randomness of random seeds $z_i^{(k)} \overset{\text{i.i.d.}}{\sim} P_z$ sampled at iteration $k$.

For the term $\mathcal{T}_1$, we have

$$
\mathbb{E}_{z^{(k)}}\left[\left\|\frac{1}{S_1^{(k)}} \sum_{i=1}^{S_1^{(k)}} \widehat{F}(\mathbf{u}_k, z_i^{(k)}) - F(\mathbf{u}_k)\right\|^2\right]
$$

(B.2)

$$
\overset{(i)}{=} \mathbb{E}_{z^{(k)}}\left[\frac{1}{\left(S_1^{(k)}\right)^2} \sum_{i=1}^{S_1^{(k)}} \left\|\widehat{F}(\mathbf{u}_k, z_i^{(k)}) - F(\mathbf{u}_k)\right\|^2\right] \le \frac{\sigma^2}{S_1^{(k)}},
$$

where $(i)$ is due to $z_i^{(k)} \overset{\text{i.i.d.}}{\sim} P_z$ and $\mathbb{E}\left[\widehat{F}(\mathbf{u}_k, z_i^{(k)})\right] = F(\mathbf{u}_k)$.

For the term $\mathcal{T}_2$, we have

$$
\mathbb{E}_{z^{(k)}}\left[\left\|\widetilde{F}(\mathbf{u}_{k-1}) + \frac{1}{S_2^{(k)}} \sum_{i=1}^{S_2^{(k)}} \left(\widehat{F}(\mathbf{u}_k, z_i^{(k)}) - \widehat{F}(\mathbf{u}_{k-1}, z_i^{(k)})\right) - F(\mathbf{u}_k)\right\|^2\right]
$$

$$
\overset{(i)}{=} \mathbb{E}_{z^{(k)}}\left[\frac{1}{\left(S_2^{(k)}\right)^2} \left\|\sum_{i=1}^{S_2^{(k)}} \left[\left(\widehat{F}(\mathbf{u}_k, z_i^{(k)}) - \widehat{F}(\mathbf{u}_{k-1}, z_i^{(k)})\right) - (F(\mathbf{u}_k) - F(\mathbf{u}_{k-1}))\right]\right\|^2\right]
$$

$$
+ \mathbb{E}_{z^{(k)}}\left[\left\|\widetilde{F}(\mathbf{u}_{k-1}) - F(\mathbf{u}_{k-1})\right\|^2\right]
$$

$$
\overset{(ii)}{=} \mathbb{E}_{z^{(k)}}\left[\frac{1}{\left(S_2^{(k)}\right)^2} \sum_{i=1}^{S_2^{(k)}} \left\|\widehat{F}(\mathbf{u}_k, z_i^{(k)}) - \widehat{F}(\mathbf{u}_{k-1}, z_i^{(k)}) - (F(\mathbf{u}_k) - F(\mathbf{u}_{k-1}))\right\|^2\right]
$$

$$
+ \mathbb{E}_{z^{(k)}}\left[\left\|\widetilde{F}(\mathbf{u}_{k-1}) - F(\mathbf{u}_{k-1})\right\|^2\right],
$$

where $(i)$ and $(ii)$ can be verified by expanding the square norm and using the assumption that all $z_i^{(k)}$ are i.i.d. and $\widehat{F}(\mathbf{x}, z_i^{(k)})$ is unbiased. Since $\mathbb{E}[\|X - \mathbb{E}X\|^2] \le \mathbb{E}[\|X\|^2]$ for any random variable $X$, and using Assumption 3 for the stochastic queries, we have

$$
\mathbb{E}_{z^{(k)}}\left[\frac{1}{\left(S_2^{(k)}\right)^2} \sum_{i=1}^{S_2^{(k)}} \left\|\widehat{F}(\mathbf{u}_k, z_i^{(k)}) - \widehat{F}(\mathbf{u}_{k-1}, z_i^{(k)}) - (F(\mathbf{u}_k) - F(\mathbf{u}_{k-1}))\right\|^2\right]
$$

$$
\le \frac{1}{\left(S_2^{(k)}\right)^2} \sum_{i=1}^{S_2^{(k)}} \mathbb{E}_{z^{(k)}}\left[\left\|\widehat{F}(\mathbf{u}_k, z_i^{(k)}) - \widehat{F}(\mathbf{u}_{k-1}, z_i^{(k)})\right\|^2\right] \le \frac{L^2 \|\mathbf{u}_k - \mathbf{u}_{k-1}\|^2}{S_2^{(k)}}.
$$

So we obtain

$$
\mathbb{E}_{z^{(k)}}\left[\left\|\widetilde{F}(\mathbf{u}_{k-1}) + \frac{1}{S_2^{(k)}}\sum_{i=1}^{S_2^{(k)}}\left(\widehat{F}(\mathbf{u}_k, z_i^{(k)}) - \widehat{F}(\mathbf{u}_{k-1}, z_i^{(k)})\right) - F(\mathbf{u}_k)\right\|^2\right]
$$

$$
\leq \left\|\widetilde{F}(\mathbf{u}_{k-1}) - F(\mathbf{u}_{k-1})\right\|^2 + \frac{L^2\left\|\mathbf{u}_k - \mathbf{u}_{k-1}\right\|^2}{S_2^{(k)}}. \tag{B.3}
$$

Plugging Inequalities (B.2) and (B.3) into Eq. (B.1), we have

$$
\mathbb{E}\left[\left\|\widetilde{F}(\mathbf{u}_k) - F(\mathbf{u}_k)\right\|^2 \Big| \mathcal{F}_{k-1}\right]
$$

$$
\leq \frac{p_k\sigma^2}{S_1^{(k)}} + (1-p_k)\left\|\widetilde{F}(\mathbf{u}_{k-1}) - F(\mathbf{u}_{k-1})\right\|^2 + \frac{(1-p_k)L^2\left\|\mathbf{u}_k - \mathbf{u}_{k-1}\right\|^2}{S_2^{(k)}}.
$$

Taking expectation with respect to all the randomness on both sides, and by the tower property of conditional expectations, we now obtain

$$
\mathbb{E}\left[\left\|\widetilde{F}(\mathbf{u}_k) - F(\mathbf{u}_k)\right\|^2\right] \leq p_k\sigma^2 \mathbb{E}\left[\frac{1}{S_1^{(k)}}\right] + (1-p_k)\mathbb{E}\left[\left\|\widetilde{F}(\mathbf{u}_{k-1}) - F(\mathbf{u}_{k-1})\right\|^2\right]
$$

$$
+ (1-p_k)L^2\mathbb{E}\left[\frac{\left\|\mathbf{u}_k - \mathbf{u}_{k-1}\right\|^2}{S_2^{(k)}}\right],
$$

which leads to the inequality in the lemma when $S_1^{(k)}$ are deterministic, thus completing the proof. $\square$

**Corollary 2.2.** *Given a target error $\epsilon > 0$, if for all $k \geq 1$, $p_k = \frac{2}{k+1}$, $S_1^{(k)} \geq \lceil \frac{8\sigma^2}{p_k\epsilon^2}\rceil$, $S_2^{(k)} \geq \lceil \frac{8L^2\|\mathbf{u}_k - \mathbf{u}_{k-1}\|^2}{p_k^2\epsilon^2}\rceil$, then $\mathbb{E}\left[\|\widetilde{F}(\mathbf{u}_k) - F(\mathbf{u}_k)\|^2\right] \leq \frac{\epsilon^2}{k}$.*

*Proof.* We prove it by induction whose base step is

$$
\mathbb{E}\left[\left\|\widetilde{F}(\mathbf{u}_1) - F(\mathbf{u}_1)\right\|^2\right] \leq \frac{p_1\sigma^2}{S_1^{(1)}} \leq \frac{\epsilon^2}{8} \leq \epsilon^2,
$$

where we use that $p_1 = 1$.

Assume that the result holds for all $j < k$; then by Lemma 2.1, we have that at iteration $k$

$$
\mathbb{E}\left[\left\|\widetilde{F}(\mathbf{u}_k) - F(\mathbf{u}_k)\right\|^2\right]
$$

$$
\leq \frac{p_k\sigma^2}{S_1^{(k)}} + (1-p_k)\mathbb{E}\left[\left\|\widetilde{F}(\mathbf{u}_{k-1}) - F(\mathbf{u}_{k-1})\right\|^2\right] + (1-p_k)L^2\mathbb{E}\left[\frac{\left\|\mathbf{u}_k - \mathbf{u}_{k-1}\right\|^2}{S_2^{(k)}}\right].
$$

Plugging in our choice of $p_k$, $S_1^{(k)}$ and $S_2^{(k)}$, we have

$$
\mathbb{E}\left[\left\|\widetilde{F}(\mathbf{u}_k) - F(\mathbf{u}_k)\right\|^2\right] \leq \frac{p_k^2\epsilon^2}{8} + \frac{(1-p_k)\epsilon^2}{k-1} + \frac{p_k^2(1-p_k)\epsilon^2}{8}
$$

$$
\overset{(i)}{\leq} \frac{p_k^2\epsilon^2}{4} + \frac{(1-p_k)\epsilon^2}{k-1} = \left(\frac{1}{(k+1)^2} + \frac{1}{k+1}\right)\epsilon^2 \overset{(ii)}{\leq} \frac{\epsilon^2}{k},
$$

where $(i)$ is due to $\frac{p_k^2(1-p_k)\epsilon^2}{8} \leq \frac{p_k^2\epsilon^2}{8}$, and $(ii)$ is because $k(k+2) \leq (k+1)^2$. Hence, by induction, we can conclude that the result holds for all $k \geq 1$. $\square$

## C  Omitted proofs from Section 3

### C.1  Unconstrained settings

Our argument for bounding the total number of stochastic queries to $F$ is based on the use of the following potential function, which was previously used for the deterministic case of Halpern iteration in [12, 13],

$$
\mathcal{C}_k = \frac{A_k}{L_k}\|F(\mathbf{u}_k)\|^2 + B_k\langle F(\mathbf{u}_k), \mathbf{u}_k - \mathbf{u}_0\rangle, \tag{C.1}
$$

where $\{A_k\}_{k \geq 1}$ and $\{B_k\}_{k \geq 1}$ are positive and non-decreasing sequences of real numbers, while the step size $\lambda_k$ is defined by $\lambda_k := \frac{B_k}{A_k + B_k}$. We start the proof by first justifying that a bound on the chosen potential function $\mathcal{C}_k$ leads to a bound on $\|F(\mathbf{u}_k)\|$ in expectation. The proof is a simple extension of [12, Lemma 4] and is provided for completeness.

**Lemma C.1.** *Given $k \geq 1$, let $\mathcal{C}_k$ be defined as in Eq. (C.1) and let $\mathbf{u}^*$ be a solution to the monotone inclusion problem corresponding to $F$. If $\mathbb{E}[\mathcal{C}_k] \leq \mathbb{E}[\mathcal{E}_k]$ for some error term $\mathcal{E}_k$, then*

$$\mathbb{E}\left[\|F(\mathbf{u}_k)\|^2\right] \leq \frac{B_k L_k}{A_k} \|\mathbf{u}_0 - \mathbf{u}^*\| \, \mathbb{E}[\|F(\mathbf{u}_k)\|] + \frac{L_k}{A_k} \mathbb{E}[\mathcal{E}_k], \qquad (\text{C.2})$$

*where the expectation is taken with respect to all random queries to $F$.*

*Proof.* By the definition of $\mathcal{C}_k$, we have

$$\begin{aligned}
\mathbb{E}\left[\|F(\mathbf{u}_k)\|^2\right] &\leq \frac{B_k L_k}{A_k} \mathbb{E}[\langle F(\mathbf{u}_k), \mathbf{u}_0 - \mathbf{u}_k \rangle] + \frac{L_k}{A_k} \mathbb{E}[\mathcal{E}_k] \\
&= \frac{B_k L_k}{A_k} \mathbb{E}[\langle F(\mathbf{u}_k), \mathbf{u}_0 - \mathbf{u}^* + \mathbf{u}^* - \mathbf{u}_k \rangle] + \frac{L_k}{A_k} \mathbb{E}[\mathcal{E}_k] \\
&= \frac{B_k L_k}{A_k} \mathbb{E}[\langle F(\mathbf{u}_k), \mathbf{u}_0 - \mathbf{u}^* \rangle] + \frac{B_k L_k}{A_k} \mathbb{E}[\langle F(\mathbf{u}_k), \mathbf{u}^* - \mathbf{u}_k \rangle] + \frac{L_k}{A_k} \mathbb{E}[\mathcal{E}_k].
\end{aligned}$$

Since $\mathbf{u}^*$ is a solution to the monotone inclusion problem, as discussed in Section 2, it is also a weak VI (or MVI) solution, and thus

$$(\forall k \geq 0) \quad \langle F(\mathbf{u}_k), \mathbf{u}^* - \mathbf{u}_k \rangle \leq 0.$$

As a result,

$$\begin{aligned}
\mathbb{E}\left[\|F(\mathbf{u}_k)\|^2\right] &\leq \frac{B_k L_k}{A_k} \mathbb{E}[\langle F(\mathbf{u}_k), \mathbf{u}_0 - \mathbf{u}^* \rangle] + \frac{L_k}{A_k} \mathbb{E}[\mathcal{E}_k] \\
&\overset{(i)}{\leq} \frac{B_k L_k}{A_k} \mathbb{E}[\|F(\mathbf{u}_k)\| \, \|\mathbf{u}_0 - \mathbf{u}^*\|] + \frac{L_k}{A_k} \mathbb{E}[\mathcal{E}_k] \\
&\overset{(ii)}{=} \frac{B_k L_k}{A_k} \|\mathbf{u}_0 - \mathbf{u}^*\| \, \mathbb{E}[\|F(\mathbf{u}_k)\|] + \frac{L_k}{A_k} \mathbb{E}[\mathcal{E}_k],
\end{aligned}$$

where we use Cauchy-Schwarz inequality for $(i)$, while $(ii)$ holds because $\|\mathbf{u}_0 - \mathbf{u}^*\|$ involves no randomness. $\qquad \square$

Using Lemma C.1, our goal now is to show that we can provide a bound on $\mathbb{E}[\mathcal{C}_k]$ by appropriately choosing the algorithm parameters. In the deterministic setup, it is sufficient to choose $L_k = \mathcal{O}(L)$ and $\lambda_k = \mathcal{O}(\frac{1}{k})$ to ensure that $\{A_k \mathcal{C}_k\}_{k \geq 1}$ is monotonically non-increasing, which immediately leads to $\mathcal{C}_k \leq \frac{A_1}{A_k} \mathcal{C}_1$. In the stochastic setup considered here, we follow the same motivation, but need to deal with additional error terms caused by the stochastic access to $F$.

We assume throughout that $L$ is known, and make the following assumption on the choice of $\{A_k\}_{k \geq 1}$, $\{B_k\}_{k \geq 1}$, and $\{L_k\}_{k \geq 1}$, and provide a corresponding bound on the change of $\mathcal{C}_k$ in Lemma C.2.

**Assumption 4.** $\{L_k\}_{k \geq 1}$ is a sequence of positive reals such that $L_k \geq L$ for all $k \in \mathbb{N}$. Sequences $\{A_k\}_{k \geq 1}$ and $\{B_k\}_{k \geq 1}$ are positive and non-decreasing, satisfying the following for all $k \geq 2$:

$$\frac{B_{k-1}}{A_k} = \frac{B_k}{A_k + B_k}, \qquad \frac{1}{L_k}\left(1 - \frac{2B_k}{A_k + B_k}\right) = \frac{A_{k-1}}{A_k L_{k-1}}.$$

**Lemma C.2.** *Let $\mathcal{C}_k$ be defined as in Eq. (C.1), where $\{A_k\}_{k \geq 1}$ and $\{B_k\}_{k \geq 1}$ satisfy Assumption 4. Let $L_k = 2L$ for all $k \geq 1$. Then, for any $k \geq 2$, we have*

$$\mathcal{C}_k - \mathcal{C}_{k-1} \leq \frac{A_k}{2L} \left\|F(\mathbf{u}_{k-1}) - \widetilde{F}(\mathbf{u}_{k-1})\right\|^2 + \frac{A_k - A_{k-1}}{2L} \left\langle F(\mathbf{u}_{k-1}), F(\mathbf{u}_{k-1}) - \widetilde{F}(\mathbf{u}_{k-1}) \right\rangle.$$

*Proof.* By the definition of $\mathcal{C}_k$, we have

$$\begin{aligned}
\mathcal{C}_k - \mathcal{C}_{k-1} = {} & \frac{A_k}{L_k} \|F(\mathbf{u}_k)\|^2 + B_k \langle F(\mathbf{u}_k), \mathbf{u}_k - \mathbf{u}_0 \rangle \\
& - \frac{A_{k-1}}{L_{k-1}} \|F(\mathbf{u}_{k-1})\|^2 - B_{k-1} \langle F(\mathbf{u}_{k-1}), \mathbf{u}_{k-1} - \mathbf{u}_0 \rangle.
\end{aligned}$$

Since the operator $F$ is cocoercive with parameter $\frac{1}{L}$, we have

$$\langle F(\mathbf{u}_k) - F(\mathbf{u}_{k-1}), \mathbf{u}_k - \mathbf{u}_{k-1} \rangle$$

$$\geq \frac{1}{L} \|F(\mathbf{u}_k) - F(\mathbf{u}_{k-1})\|^2$$

$$= \frac{1}{L_k} \|F(\mathbf{u}_k) - F(\mathbf{u}_{k-1})\|^2 + \left(\frac{1}{L} - \frac{1}{L_k}\right) \|F(\mathbf{u}_k) - F(\mathbf{u}_{k-1})\|^2$$

$$= \frac{1}{L_k} \|F(\mathbf{u}_k)\|^2 - \frac{2}{L_k} \langle F(\mathbf{u}_k), F(\mathbf{u}_{k-1}) \rangle + \frac{1}{L_k} \|F(\mathbf{u}_{k-1})\|^2$$

$$+ \left(\frac{1}{L} - \frac{1}{L_k}\right) \|F(\mathbf{u}_k) - F(\mathbf{u}_{k-1})\|^2 .$$

By rearranging, we obtain

$$\frac{1}{L_k} \|F(\mathbf{u}_k)\|^2 \leq \left\langle F(\mathbf{u}_k), \mathbf{u}_k - \mathbf{u}_{k-1} + \frac{2}{L_k} F(\mathbf{u}_{k-1}) \right\rangle - \langle F(\mathbf{u}_{k-1}), \mathbf{u}_k - \mathbf{u}_{k-1} \rangle$$

$$- \frac{1}{L_k} \|F(\mathbf{u}_{k-1})\|^2 - \left(\frac{1}{L} - \frac{1}{L_k}\right) \|F(\mathbf{u}_k) - F(\mathbf{u}_{k-1})\|^2 .$$

Multiplying $A_k$ on both sides and plugging into $\mathcal{C}_k - \mathcal{C}_{k-1}$, we have

$$\mathcal{C}_k - \mathcal{C}_{k-1} \leq \left\langle F(\mathbf{u}_k), A_k(\mathbf{u}_k - \mathbf{u}_{k-1}) + \frac{2A_k}{L_k} F(\mathbf{u}_{k-1}) + B_k(\mathbf{u}_k - \mathbf{u}_0) \right\rangle$$

$$- \langle F(\mathbf{u}_{k-1}), A_k(\mathbf{u}_k - \mathbf{u}_{k-1}) + B_{k-1}(\mathbf{u}_{k-1} - \mathbf{u}_0) \rangle$$

$$- \left(\frac{A_k}{L_k} + \frac{A_{k-1}}{L_{k-1}}\right) \|F(\mathbf{u}_{k-1})\|^2 - A_k \left(\frac{1}{L} - \frac{1}{L_k}\right) \|F(\mathbf{u}_k) - F(\mathbf{u}_{k-1})\|^2 .$$

Since $\lambda_k = \frac{B_k}{A_k + B_k}$, we have

$$\mathbf{u}_k = \frac{B_k}{A_k + B_k} \mathbf{u}_0 + \frac{A_k}{A_k + B_k} \left(\mathbf{u}_{k-1} - \frac{2}{L_k} \widetilde{F}(\mathbf{u}_{k-1})\right),$$

which leads to $A_k(\mathbf{u}_k - \mathbf{u}_{k-1}) + \frac{2A_k}{L_k} F(\mathbf{u}_{k-1}) + B_k(\mathbf{u}_k - \mathbf{u}_0) = \frac{2A_k}{L_k}\left(F(\mathbf{u}_{k-1}) - \widetilde{F}(\mathbf{u}_{k-1})\right)$.

Further, as $\frac{B_{k-1}}{A_k} = \frac{B_k}{A_k + B_k}$ by Assumption 4, we have

$$\langle F(\mathbf{u}_{k-1}), A_k(\mathbf{u}_k - \mathbf{u}_{k-1}) + B_{k-1}(\mathbf{u}_{k-1} - \mathbf{u}_0) \rangle$$

$$= A_k \left\langle F(\mathbf{u}_{k-1}), \mathbf{u}_k - \frac{B_{k-1}}{A_k} \mathbf{u}_0 - \frac{A_k - B_{k-1}}{A_k} \mathbf{u}_{k-1} \right\rangle$$

$$= A_k \left\langle F(\mathbf{u}_{k-1}), \mathbf{u}_k - \frac{B_k}{A_k + B_k} \mathbf{u}_0 - \frac{A_k}{A_k + B_k} \mathbf{u}_{k-1} \right\rangle$$

$$= -A_k \left\langle F(\mathbf{u}_{k-1}), \frac{2A_k}{L_k(A_k + B_k)} \widetilde{F}(\mathbf{u}_{k-1}) \right\rangle .$$

Moreover, by Assumption 4, we have $\frac{1}{L_k}\left(1 - \frac{2B_k}{A_k + B_k}\right) = \frac{A_{k-1}}{A_k L_{k-1}}$, so we obtain

$$\langle F(\mathbf{u}_{k-1}), A_k(\mathbf{u}_k - \mathbf{u}_{k-1}) + B_{k-1}(\mathbf{u}_{k-1} - \mathbf{u}_0) \rangle$$

$$= -A_k \left\langle F(\mathbf{u}_{k-1}), \frac{2A_k}{L_k(A_k + B_k)} \widetilde{F}(\mathbf{u}_{k-1}) \right\rangle$$

$$= -\left\langle F(\mathbf{u}_{k-1}), \left(\frac{A_k}{L_k} + \frac{A_{k-1}}{L_{k-1}}\right) \widetilde{F}(\mathbf{u}_{k-1}) \right\rangle .$$

Since by hypothesis $L_k = 2L$ for all $k \geq 1$, we have

$$\mathcal{C}_k - \mathcal{C}_{k-1} \leq \left\langle F(\mathbf{u}_k), \frac{A_k}{L}(F(\mathbf{u}_{k-1}) - \widetilde{F}(\mathbf{u}_{k-1})) \right\rangle + \left\langle F(\mathbf{u}_{k-1}), \frac{A_k + A_{k-1}}{2L} \widetilde{F}(\mathbf{u}_{k-1}) \right\rangle$$

$$- \frac{A_k + A_{k-1}}{2L} \|F(\mathbf{u}_{k-1})\|^2 - \frac{A_k}{2L} \|F(\mathbf{u}_k) - F(\mathbf{u}_{k-1})\|^2$$

$$\overset{(i)}{=} \frac{A_k}{L} \left\langle F(\mathbf{u}_k) - F(\mathbf{u}_{k-1}), F(\mathbf{u}_{k-1}) - \widetilde{F}(\mathbf{u}_{k-1}) \right\rangle - \frac{A_k}{2L} \|F(\mathbf{u}_k) - F(\mathbf{u}_{k-1})\|^2$$

$$+ \left\langle F(\mathbf{u}_{k-1}), \frac{A_k - A_{k-1}}{2L} \left(F(\mathbf{u}_{k-1}) - \widetilde{F}(\mathbf{u}_{k-1})\right) \right\rangle,$$

where $(i)$ is derived by rearranging and grouping terms. Using that $2\langle p, q\rangle - \|p\|^2 \le \|q\|^2$ holds for any $p, q \in \mathbb{R}^d$, we finally obtain

$$\mathcal{C}_k - \mathcal{C}_{k-1} \le \frac{A_k}{2L}\left\|F(\mathbf{u}_{k-1}) - \widetilde{F}(\mathbf{u}_{k-1})\right\|^2 + \frac{A_k - A_{k-1}}{2L}\left\langle F(\mathbf{u}_{k-1}), F(\mathbf{u}_{k-1}) - \widetilde{F}(\mathbf{u}_{k-1})\right\rangle,$$

thus completing the proof. $\qquad\square$

By Lemma C.2, if we choose $A_k = \mathcal{O}(k^2)$ and $B_k = \mathcal{O}(k)$ satisfying Assumption 4, and take sufficiently large size of samples queried to ensure that $\mathbb{E}\big[\big\|F(\mathbf{u}_k) - \widetilde{F}(\mathbf{u}_k)\big\|^2\big] \le \frac{\epsilon^2}{k}$ for $k \ge 0$, then we can obtain $\mathcal{O}(1/k)$ expected convergence rate in the norm of the operator by induction. Observe that we do not need an assumption that $\widetilde{F}$ is an unbiased estimator of $F$ for any point except for the initial one; all that is needed is that the second moment of the estimation error, $\|F(\mathbf{u}_k) - \widetilde{F}(\mathbf{u}_k)\|_2^2$, is bounded.

**Theorem 3.1.** *Given an arbitrary $\mathbf{u}_0 \in \mathbb{R}^d$, suppose that iterates $\mathbf{u}_k$ evolve according to Halpern iteration from Eq. (3.1) for $k \ge 1$, where $L_k = 2L$ and $\lambda_k = \frac{1}{k+1}$. Assume further that the stochastic estimate $\widetilde{F}(\mathbf{u})$ is unbiased for $\mathbf{u} = \mathbf{u}_0$ and $\mathbb{E}[\|F(\mathbf{u}_0) - \widetilde{F}(\mathbf{u}_0)\|^2] \le \frac{\epsilon^2}{8}$. Given $\epsilon > 0$, if for all $k \ge 1$, we have that $\mathbb{E}\big[\big\|F(\mathbf{u}_k) - \widetilde{F}(\mathbf{u}_k)\big\|^2\big] \le \frac{\epsilon^2}{k}$, then for all $k \ge 1$,*

$$\mathbb{E}[\|F(\mathbf{u}_k))\|] \le \frac{\Lambda_0}{k} + \Lambda_1\epsilon, \tag{3.2}$$

*where $\Lambda_0 = 76L\|\mathbf{u}_0 - \mathbf{u}^*\|$ and $\Lambda_1 = 4\sqrt{\frac{2}{3}}$. As a result, stochastic Halpern iteration from Eq. (3.1) returns a point $\mathbf{u}_k$ such that $\mathbb{E}[\|F(\mathbf{u}_k)\|] \le 4\epsilon$ after at most $N = \lceil\frac{2\Lambda_0}{\epsilon}\rceil = \mathcal{O}\big(\frac{L\|\mathbf{u}_0 - \mathbf{u}^*\|}{\epsilon}\big)$ iterations.*

*Proof.* Observe first that the chosen sequence of numbers $A_k, B_k$ satisfies Assumption 4, and thus Lemma C.2 applies. Observe further that, by Jensen's Inequality,

$$\mathbb{E}[\|F(\mathbf{u}_k))\|] \le \Big(\mathbb{E}[\|F(\mathbf{u}_k)\|^2]\Big)^{\frac{1}{2}}.$$

and, thus, to prove the theorem, it suffices to show that there exists $\Lambda_0$ and $\Lambda_1$ such that for all $k \ge 1$

$$\Big(\mathbb{E}[\|F(\mathbf{u}_k)\|^2]\Big)^{\frac{1}{2}} \le \frac{\Lambda_0}{k} + \Lambda_1\epsilon.$$

We prove this claim by induction on $k$. For the base case $k = 1$, in which $\mathbf{u}_1 = \mathbf{u}_0 - \frac{1}{2L}\widetilde{F}(\mathbf{u}_0)$, we have

$$\mathcal{C}_1 = \frac{1}{L}\|F(\mathbf{u}_1)\|^2 + 2\langle F(\mathbf{u}_1), \mathbf{u}_1 - \mathbf{u}_0\rangle = \frac{1}{L}\Big(\|F(\mathbf{u}_1)\|^2 - \big\langle F(\mathbf{u}_1), \widetilde{F}(\mathbf{u}_0)\big\rangle\Big). \tag{C.3}$$

Further, since the operator $F$ is cocoercive with parameter $\frac{1}{L}$, it is also cocoercive with parameter $\frac{1}{2L}$, and thus we have

$$\|F(\mathbf{u}_1) - F(\mathbf{u}_0)\|^2 \le 2L\langle F(\mathbf{u}_1) - F(\mathbf{u}_0), \mathbf{u}_1 - \mathbf{u}_0\rangle = \big\langle F(\mathbf{u}_1) - F(\mathbf{u}_0), -\widetilde{F}(\mathbf{u}_0)\big\rangle.$$

Expanding and rearranging the terms, we have

$$\|F(\mathbf{u}_1)\|^2 \le \big\langle F(\mathbf{u}_0), \widetilde{F}(\mathbf{u}_0) - F(\mathbf{u}_0)\big\rangle + 2\langle F(\mathbf{u}_1), F(\mathbf{u}_0)\rangle - \big\langle F(\mathbf{u}_1), \widetilde{F}(\mathbf{u}_0)\big\rangle.$$

Recall that, by assumption, $\mathbb{E}[\widetilde{F}(\mathbf{u}_0)] = F(\mathbf{u}_0)$. Subtracting $\big\langle F(\mathbf{u}_1), \widetilde{F}(\mathbf{u}_0)\big\rangle$ from both sides in the last inequality and taking expectation with respect to all the randomness on both sides, we have

$$\mathbb{E}\Big[\|F(\mathbf{u}_1)\|^2 - \big\langle F(\mathbf{u}_1), \widetilde{F}(\mathbf{u}_0)\big\rangle\Big]$$
$$\le \mathbb{E}\Big[\big\langle F(\mathbf{u}_0), \widetilde{F}(\mathbf{u}_0) - F(\mathbf{u}_0)\big\rangle + 2\langle F(\mathbf{u}_1), F(\mathbf{u}_0)\rangle - 2\big\langle F(\mathbf{u}_1), \widetilde{F}(\mathbf{u}_0)\big\rangle\Big]$$
$$= 2\mathbb{E}\Big[\big\langle F(\mathbf{u}_1), F(\mathbf{u}_0) - \widetilde{F}(\mathbf{u}_0)\big\rangle\Big]$$
$$\overset{(i)}{\le} \mathbb{E}\Big[\frac{1}{2}\|F(\mathbf{u}_1)\|^2 + 2\|F(\mathbf{u}_0) - \widetilde{F}(\mathbf{u}_0)\|^2\Big],$$

where for $(i)$ we use Young's inequality. Plugging into Eq. (C.3), we obtain that

$$\mathbb{E}[\mathcal{C}_1] \leq \frac{1}{L} \mathbb{E}\left[\frac{1}{2}\|F(\mathbf{u}_1)\|^2 + 2\|F(\mathbf{u}_0) - \widetilde{F}(\mathbf{u}_0)\|^2\right].$$

Note that $A_1 = B_1 = 2$ and $L_1 = 2L$, by Lemma C.1 we have

$$\mathbb{E}[\|F(\mathbf{u}_1)\|^2] \leq \frac{B_1 L_1}{A_1}\|\mathbf{u}_0 - \mathbf{u}^*\|\mathbb{E}[\|F(\mathbf{u}_1)\|] + \frac{L_1}{A_1}\frac{1}{L}\mathbb{E}\left[\frac{1}{2}\|F(\mathbf{u}_1)\|^2 + 2\|F(\mathbf{u}_0) - \widetilde{F}(\mathbf{u}_0)\|^2\right]$$

$$= 2L\|\mathbf{u}_0 - \mathbf{u}^*\|\mathbb{E}[\|F(\mathbf{u}_1)\|] + \mathbb{E}\left[\frac{1}{2}\|F(\mathbf{u}_1)\|^2 + 2\|F(\mathbf{u}_0) - \widetilde{F}(\mathbf{u}_0)\|^2\right].$$

Subtracting $\mathbb{E}[\frac{1}{2}\|F(\mathbf{u}_1)\|^2]$ on both sides and using that (by Jensen's inequality) $\mathbb{E}[\|F(\mathbf{u}_1)\|] \leq \left(\mathbb{E}[\|F(\mathbf{u}_1)\|^2]\right)^{\frac{1}{2}}$ and (by assumption) $\mathbb{E}[\|F(\mathbf{u}_0) - \widetilde{F}(\mathbf{u}_0)\|^2] \leq \frac{\epsilon^2}{8}$, we have

$$\mathbb{E}[[\|F(\mathbf{u}_1)\|^2] \leq 4L\|\mathbf{u}_0 - \mathbf{u}^*\|\left(\mathbb{E}[\|F(\mathbf{u}_1)\|^2]\right)^{\frac{1}{2}} + \frac{\epsilon^2}{2},$$

which is a quadratic inequality in $(\mathbb{E}[\|F(\mathbf{u}_1)\|^2])^{\frac{1}{2}}$. Bounding the solution to this quadratic inequality by its larger root, we have

$$(\mathbb{E}[\|F(\mathbf{u}_1)\|^2])^{\frac{1}{2}} \leq 2L\|\mathbf{u}_0 - \mathbf{u}^*\| + \frac{1}{2}\sqrt{16L^2\|\mathbf{u}_0 - \mathbf{u}^*\|^2 + 2\epsilon^2}$$

$$\leq 2L\|\mathbf{u}_0 - \mathbf{u}^*\| + \frac{1}{2}(4L\|\mathbf{u}_0 - \mathbf{u}^*\| + \sqrt{2}\epsilon)$$

$$\leq 4L\|\mathbf{u}_0 - \mathbf{u}^*\| + \epsilon$$

$$\leq \Lambda_0 + \Lambda_1\epsilon.$$

This completes the proof for the base case. Moreover, we can get a bound for $\mathbb{E}[\mathcal{C}_1]$ as follows

$$\mathbb{E}[\mathcal{C}_1] \leq \frac{1}{L}\mathbb{E}\left[\frac{1}{2}\|F(\mathbf{u}_1)\|^2 + 2\left\|F(\mathbf{u}_0) - \widetilde{F}(\mathbf{u}_0)\right\|^2\right]$$

$$\overset{(i)}{\leq} \frac{1}{2L}\left(24L^2\|\mathbf{u}_0 - \mathbf{u}^*\|^2 + \frac{3}{2}\epsilon^2\right) + \frac{2}{L}\frac{\epsilon^2}{8}$$

$$= 12L\|\mathbf{u}_0 - \mathbf{u}^*\|^2 + \frac{\epsilon^2}{L},$$

where $(i)$ can be verified by the bound we get above for $\mathbb{E}[\|F(\mathbf{u}_1)\|^2]$ and by applying Young's inequality and that, by assumption, $\mathbb{E}[\|F(\mathbf{u}_0) - \widetilde{F}(\mathbf{u}_0)\|^2] \leq \frac{\epsilon^2}{8}$.

For the inductive hypothesis, assume that the result holds for all $1 \leq i \leq k-1$, and consider iteration $k$. By Lemma C.2, we have for $\forall i \geq 2$

$$\mathcal{C}_i - \mathcal{C}_{i-1} \leq \frac{A_i}{2L}\left\|F(\mathbf{u}_{i-1}) - \widetilde{F}(\mathbf{u}_{i-1})\right\|^2 + \frac{A_i - A_{i-1}}{2L}\left\langle F(\mathbf{u}_{i-1}), F(\mathbf{u}_{i-1}) - \widetilde{F}(\mathbf{u}_{i-1})\right\rangle$$

$$\overset{(i)}{\leq} \frac{5i(i+1)}{2L}\left\|F(\mathbf{u}_{i-1}) - \widetilde{F}(\mathbf{u}_{i-1})\right\|^2 + \frac{i}{8L(i+1)}\|F(\mathbf{u}_{i-1})\|^2,$$

where we use Young's inequality and $A_i = i(i+1)$ for $(i)$. Taking expectation with respect to all randomness on both sides and telescoping from $i = 2$ to $k$, we obtain

$$\mathbb{E}[\mathcal{C}_k] \leq \mathbb{E}\left[\mathcal{C}_1 + \sum_{i=2}^{k}\left(\frac{5i(i+1)}{2L}\|F(\mathbf{u}_{i-1}) - \widetilde{F}(\mathbf{u}_{i-1})\|^2 + \frac{i}{8L(i+1)}\|F(\mathbf{u}_{i-1})\|^2\right)\right]$$

$$\leq \mathbb{E}\left[\sum_{i=2}^{k}\left(\frac{5i(i+1)}{2L}\left\|F(\mathbf{u}_{i-1}) - \widetilde{F}(\mathbf{u}_{i-1})\right\|^2 + \frac{i}{8L(i+1)}\|F(\mathbf{u}_{i-1})\|^2\right)\right] \quad (C.4)$$

$$+ 12L\|\mathbf{u}_0 - \mathbf{u}^*\|^2 + \frac{\epsilon^2}{L}.$$

Using that, by assumption, for $k \geq 1$, $\mathbb{E}[\|F(\mathbf{u}_k) - \widetilde{F}(\mathbf{u}_k)\|^2] \leq \frac{\epsilon^2}{k}$, we further have

$$
\begin{aligned}
\mathbb{E}\Big[\sum_{i=2}^{k} \frac{5i(i+1)}{2L} \|F(\mathbf{u}_{i-1}) - \widetilde{F}(\mathbf{u}_{i-1})\|^2\Big] &\leq \sum_{i=2}^{k} \frac{5i(i+1)}{2L} \frac{\epsilon^2}{i-1} \\
&\stackrel{(i)}{\leq} \sum_{i=2}^{k} \frac{5(i+1)\epsilon^2}{L} \\
&= \frac{5(k+4)(k-1)\epsilon^2}{2L},
\end{aligned}
\tag{C.5}
$$

where $(i)$ is because $\frac{i}{i-1} \leq 2$ for all $i \geq 2$. By induction, we have

$$
\begin{aligned}
\mathbb{E}\Big[\sum_{i=2}^{k} \frac{i}{8L(i+1)} \|F(\mathbf{u}_{i-1})\|^2\Big] &\stackrel{(i)}{\leq} \sum_{i=2}^{k} \frac{1}{8L}\Big(\frac{2\Lambda_0^2}{(i-1)^2} + 2\Lambda_1^2\epsilon^2\Big) \\
&\stackrel{(ii)}{\leq} \frac{1}{4L}\Big(\Lambda_0^2 \frac{\pi^2}{6} + (k-1)\Lambda_1^2\epsilon^2\Big) \\
&= \frac{1}{L}\Big(\frac{\Lambda_0^2\pi^2}{24} + \frac{(k-1)\Lambda_1^2\epsilon^2}{4}\Big),
\end{aligned}
\tag{C.6}
$$

where $(i)$ follows from induction and $\frac{i}{i+1} \leq 1$, and $(ii)$ is due to $\sum_{i=2}^{k} \frac{1}{(i-1)^2} \leq \sum_{i=1}^{\infty} \frac{1}{i^2} = \frac{\pi^2}{6}$.
Combining Eqs. (C.4)–(C.6), we get

$$
\mathbb{E}[\mathcal{C}_k] \leq 12L \|\mathbf{u}_0 - \mathbf{u}^*\|^2 + \frac{\epsilon^2}{L} + \frac{5(k+4)(k-1)\epsilon^2}{2L} + \frac{1}{L}\Big(\frac{\Lambda_0^2\pi^2}{24} + \frac{(k-1)\Lambda_1^2\epsilon^2}{4}\Big).
$$

Applying Lemma C.1 to the bound on $\mathcal{C}_k$ from the last inequality, we have

$$
\begin{aligned}
&\mathbb{E}\big[\|F(\mathbf{u}_k)\|^2\big] \\
&\leq \frac{B_k L_k}{A_k} \|\mathbf{u}_0 - \mathbf{u}^*\| \mathbb{E}[\|F(\mathbf{u}_k)\|] \\
&\quad + \frac{L_k}{A_k}\Big(12L\|\mathbf{u}_0 - \mathbf{u}^*\|^2 + \frac{\epsilon^2}{L} + \frac{5(k+4)(k-1)\epsilon^2}{2L} + \frac{\Lambda_0^2\pi^2}{24L} + \frac{(k-1)\Lambda_1^2\epsilon^2}{4L}\Big) \\
&= \frac{2L}{k} \|\mathbf{u}_0 - \mathbf{u}^*\| \mathbb{E}[\|F(\mathbf{u}_k)\|] \\
&\quad + \frac{1}{k(k+1)}\Big(24L^2\|\mathbf{u}_0 - \mathbf{u}^*\|^2 + 2\epsilon^2 + 5(k+4)(k-1)\epsilon^2 + \frac{\Lambda_0^2\pi^2}{12} + \frac{(k-1)\Lambda_1^2\epsilon^2}{2}\Big) \\
&\stackrel{(i)}{\leq} \frac{2L}{k} \|\mathbf{u}_0 - \mathbf{u}^*\| \mathbb{E}[\|F(\mathbf{u}_k)\|] + \Big(\frac{24L^2\|\mathbf{u}_0 - \mathbf{u}^*\|^2}{k^2} + \Big(8 + \frac{\Lambda_1^2}{2(k+1)}\Big)\epsilon^2 + \frac{\Lambda_0^2\pi^2}{12k^2}\Big),
\end{aligned}
$$

where $(i)$ is due to $\frac{1}{k(k+1)} \leq \frac{1}{k^2}$, $\frac{5(k+1)(k-1)}{k(k+1)} \leq 6$ and $\frac{k-1}{k(k+1)} \leq \frac{1}{k+1}$. Since $\mathbb{E}[\|F(\mathbf{u}_k)\|] \leq (\mathbb{E}[\|F(\mathbf{u}_k)\|^2])^{\frac{1}{2}}$ by Jensen's inequality, we have

$$
\begin{aligned}
&\mathbb{E}\big[\|F(\mathbf{u}_k)\|^2\big] \\
&\leq \frac{2L}{k} \|\mathbf{u}_0 - \mathbf{u}^*\| \Big(\mathbb{E}\big[\|F(\mathbf{u}_k)\|^2\big]\Big)^{\frac{1}{2}} + \Big(\frac{24L^2\|\mathbf{u}_0 - \mathbf{u}^*\|^2}{k^2} + \Big(8 + \frac{\Lambda_1^2}{2(k+1)}\Big)\epsilon^2 + \frac{\Lambda_0^2\pi^2}{12k^2}\Big),
\end{aligned}
$$

which is a quadratic inequality with respect to $(\mathbb{E}[\|F(\mathbf{u}_k)\|^2])^{\frac{1}{2}}$. Similarly as for $k = 1$, bounding its solution by its larger root, we obtain

$$
\left(\mathbb{E}\big[\,\|F(\mathbf{u}_k)\|^2\,\big]\right)^{\frac{1}{2}}
$$

$$
\leq \frac{L}{k}\|\mathbf{u}_0 - \mathbf{u}^*\| + \frac{1}{2}\sqrt{\frac{4L^2}{k^2}\|\mathbf{u}_0 - \mathbf{u}^*\|^2 + 4\Big(\frac{24L^2\|\mathbf{u}_0 - \mathbf{u}^*\|^2}{k^2} + \Big(8 + \frac{\Lambda_1^2}{2(k+1)}\Big)\epsilon^2 + \frac{\Lambda_0^2\pi^2}{12k^2}\Big)}
$$

$$
\overset{(i)}{\leq} \frac{2L}{k}\|\mathbf{u}_0 - \mathbf{u}^*\| + \Big(\frac{5L\|\mathbf{u}_0 - \mathbf{u}^*\|}{k} + \sqrt{8 + \frac{\Lambda_1^2}{2(k+1)}}\,\epsilon + \frac{\Lambda_0\pi}{2\sqrt{3}k}\Big)
$$

$$
= \frac{7L\|\mathbf{u}_0 - \mathbf{u}^*\| + \frac{\Lambda_0\pi}{2\sqrt{3}}}{k} + \sqrt{8 + \frac{\Lambda_1^2}{2(k+1)}}\,\epsilon
$$

$$
\overset{(ii)}{\leq} \frac{\Lambda_0}{k} + \Lambda_1\epsilon,
$$

where $(i)$ is due to the fact that $\sqrt{\sum_{i=1}^{n}X_i^2} \leq \sum_{i=1}^{n}|X_i|$, and $(ii)$ is because of our choice of $\Lambda_0, \Lambda_1$. Hence, the result also holds for the case $k$. Then by induction we know that the result holds for all $k \geq 1$.

Finally, when $k \geq \frac{2\Lambda_0}{\epsilon}$, we have $\frac{\Lambda_0}{k} \leq \epsilon/2$. Also, since we have $\Lambda_1 = 4\sqrt{\frac{2}{3}} < 3.5$, we obtain

$$
\mathbb{E}[\|F(\mathbf{u}_k)\|] \leq \Big(\frac{1}{2} + 4\sqrt{\frac{2}{3}}\Big)\epsilon \leq 4\epsilon.
$$

Hence, the total number of iterations needed to attain $4\epsilon$ norm of the operator is

$$
N = \left\lceil\frac{2\Lambda_0}{\epsilon}\right\rceil = \mathcal{O}\Big(\frac{L\|\mathbf{u} - \mathbf{u}^*\|}{\epsilon}\Big),
$$

thus completing the proof. □

Then we provide the deferred proof for Corollary 3.2 on the oracle complexity using a simple mini-batch estimator.

**Corollary 3.2.** *Under the assumptions of Theorem 3.1, if $\widetilde{F}(\mathbf{u}_k) = \frac{1}{S_k}\sum_{i=1}^{S_k}\widehat{F}(\mathbf{u}_k, z_i^{(k)})$, where $\widehat{F}(\mathbf{u}_k, z_i^{(k)})$ satisfies Assumption 1 and $z_i^{(k)} \overset{i.i.d.}{\sim} P_z$, then setting $S_k = \frac{\sigma^2(k+1)}{\epsilon^2}$ for all $k \geq 0$ guarantees that $\mathbb{E}[\|F(\mathbf{u}_k)\|] \leq 4\epsilon$ after at most $\mathcal{O}\big(\frac{\sigma^2 L^2\|\mathbf{u}_0 - \mathbf{u}^*\|^2}{\epsilon^4}\big)$ queries to $\widehat{F}$.*

*Proof.* The averaged operator from the theorem statement is unbiased, by Assumption 1. Further, as by Assumption 1, $\|F(\mathbf{u}_k) - \widehat{F}(\mathbf{u}_k, z_i^{(k)})\|^2 \leq \sigma^2$, it immediately follows that $\|F(\mathbf{u}_k) - \widetilde{F}(\mathbf{u}_k)\|^2 \leq \frac{\sigma^2}{S_k} = \frac{\epsilon^2}{k+1}$. Applying Theorem 3.1, the total number of iterations $N$ of Halpern iteration until $\mathbb{E}[\|F(\mathbf{u}_N)\|] \leq 4\epsilon$ is $N = \mathcal{O}(\frac{L\|\mathbf{u}_0 - \mathbf{u}^*\|}{\epsilon})$. To complete the proof, it remains to bound the total number of oracle queries $\widehat{F}$ to $F$, which is simply $\sum_{k=0}^{N}S_k = \mathcal{O}\big(\frac{N^2\sigma^2}{\epsilon^2}\big) = \mathcal{O}\big(\frac{\sigma^2 L^2\|\mathbf{u}_0 - \mathbf{u}^*\|^2}{\epsilon^4}\big)$. □

**Lemma 3.3.** *Given an arbitrary initial point $\mathbf{u}_0 \in \mathbb{R}^d$, let $\{\mathbf{u}_k\}_{k\geq 1}$ be the sequence of points produced by Algorithm 1. Assume further that $\lambda_k = \frac{1}{k+1}$, $L_k = 2L$ for all $k \geq 0$. Then,*

$$
\|\mathbf{u}_k - \mathbf{u}_{k-1}\|^2 \leq \begin{cases} \frac{1}{4L^2}\|\widetilde{F}(\mathbf{u}_0)\|^2 & \text{if } k = 1, \\ \frac{2k^2}{L^2(k+1)^2}\|\widetilde{F}(\mathbf{u}_{k-1})\|^2 + \sum_{i=0}^{k-2}\frac{2(i+1)^2}{k(k+1)^2L^2}\|\widetilde{F}(\mathbf{u}_i)\|^2 & \text{if } k \geq 2. \end{cases} \tag{3.3}
$$

*Moreover, if for $1 \leq i \leq k - 1$, all of the following conditions hold (same as in Theorem 3.1): (i) $\mathbb{E}[\|F(\mathbf{u}_i)\|] \leq \frac{\Lambda_0}{i} + \Lambda_1\epsilon$, where $\Lambda_0 = 76L\|\mathbf{u}_0 - \mathbf{u}^*\|$ and $\Lambda_1 = 4\sqrt{\frac{2}{3}}$, (ii) $\mathbb{E}\big[\|F(\mathbf{u}_i) - \widetilde{F}(\mathbf{u}_i)\|^2\big] \leq \frac{\epsilon^2}{i}$, and (iii) $\epsilon \leq \frac{\Lambda_0}{k}$, then $\mathbb{E}[\|\mathbf{u}_k - \mathbf{u}_{k-1}\|^2] = \mathcal{O}\big(\frac{\|\mathbf{u}_0 - \mathbf{u}^*\|^2}{k^2}\big)$.*

*Proof.* For $k = 1$, $\mathbf{u}_1 = \frac{1}{2}\mathbf{u}_0 + \frac{1}{2}\big(\mathbf{u}_0 - \frac{1}{L}\widetilde{F}(\mathbf{u}_0)\big)$, which leads to $\|\mathbf{u}_1 - \mathbf{u}_0\|^2 = \big\|-\frac{1}{2L}\widetilde{F}(\mathbf{u}_0)\big\|^2 = \frac{1}{4L^2}\big\|\widetilde{F}(\mathbf{u}_0)\big\|^2$. For $k \geq 2$, recursively applying Eq. (3.1), we have $\mathbf{u}_k - \mathbf{u}_{k-1} = \lambda_k(\mathbf{u}_0 - \mathbf{u}_{k-1}) - \frac{1-\lambda_k}{L}\widetilde{F}(\mathbf{u}_{k-1}) = \lambda_k(1 - \lambda_{k-1})(\mathbf{u}_0 - \mathbf{u}_{k-2}) + \frac{\lambda_k(1-\lambda_{k-1})}{L}\widetilde{F}(\mathbf{u}_{k-2}) - \frac{1-\lambda_k}{L}\widetilde{F}(\mathbf{u}_{k-1})$, leading to

$$\mathbf{u}_k - \mathbf{u}_{k-1} = -\frac{1-\lambda_k}{L}\widetilde{F}(\mathbf{u}_{k-1}) + \sum_{i=0}^{k-2}\frac{\lambda_k}{L}\Big(\prod_{j=i+1}^{k-1}(1-\lambda_j)\Big)\widetilde{F}(\mathbf{u}_i).$$

Recalling that $\lambda_k = \frac{1}{k+1}$, we have $\|\mathbf{u}_k - \mathbf{u}_{k-1}\|^2 = \big\|-\frac{k}{L(k+1)}\widetilde{F}(\mathbf{u}_{k-1}) + \sum_{i=0}^{k-2}\frac{i+1}{k(k+1)L}\widetilde{F}(\mathbf{u}_i)\big\|^2$, which gives us Inequality (3.3) by applying a generalized variant of Young's inequality $\big\|\sum_{i=1}^{K}X_i\big\|^2 \leq \sum_{i=1}^{K}K\|X_i\|^2$ twice (first to the sum of $-\frac{k}{L(k+1)}\widetilde{F}(\mathbf{u}_{k-1})$ and the summation term, then to the summation term, while noticing that $\frac{k-1}{k} \leq 1$).

For the second claim, by the lemma assumptions and the analysis in the proof for Theorem 3.1, we have $\mathbb{E}[\|F(\mathbf{u}_i)\|^2] = \mathcal{O}\big(\frac{L^2\|\mathbf{u}_0 - \mathbf{u}^*\|^2}{i^2}\big)$ for $i \leq k-1 \leq \mathcal{O}\big(\frac{1}{\epsilon}\big)$, thus $\mathbb{E}[\|\widetilde{F}(\mathbf{u}_i)\|^2] \leq 2\mathbb{E}[\|F(\mathbf{u}_i)\|]^2 + 2\mathbb{E}[\|F(\mathbf{u}_i) - \widetilde{F}(\mathbf{u}_i)\|^2] = \mathcal{O}\big(\frac{L^2\|\mathbf{u}_0 - \mathbf{u}^*\|^2}{i^2}\big)$. Plugging this bound into Inequality (3.3), we get $\mathbb{E}[\|\mathbf{u}_k - \mathbf{u}_{k-1}\|^2] = \mathcal{O}\big(\frac{\|\mathbf{u}_0 - \mathbf{u}^*\|^2}{k^2}\big)$. $\qquad\square$

### C.2 Constrained setting with a cocoercive operator

To extend the results to possibly constrained settings, similar to [12], we make use of the operator mapping defined by

$$G_\eta(\mathbf{u}) = \eta\Big(\mathbf{u} - \Pi_{\mathcal{U}}\big(\mathbf{u} - \frac{1}{\eta}F(\mathbf{u})\big)\Big), \tag{C.7}$$

where $\mathcal{U} \subseteq \mathbb{R}^d$ is the closed convex constraint set and $\Pi_{\mathcal{U}}(\mathbf{u})$ is the projection operator. Operator $G_\eta$ is a valid proxy for approximating (MI); see [12] for further details.

The extension of our results to constrained stochastic settings is not immediate; the reason is that the stochastic query assumptions (Assumptions 1 and 2) are made for the operator $F$, not $G_\eta$. Nevertheless, as we show in this subsection, it is not hard to match the stochastic oracle complexity of the unconstrained setups by proving an additional auxiliary result that bounds the variance of an operator mapping corresponding to $\widetilde{F}$ (Lemma C.4).

We begin by recalling that whenever $F$ is $\frac{1}{L}$-cocoercive and $\eta \geq L$, the operator mapping $G_\eta$ is $\frac{3}{4\eta}$-cocoercive (see, e.g., [12, Proposition 7] and [4, Lemma 10.11]).

**Proposition C.3.** *Let $F$ be $\frac{1}{L}$-cocoercive and let $G_\eta$ be defined as in Eq. (C.7), where $\eta \geq L$. Then $G_\eta$ is $\frac{3}{4\eta}$-cocoercive.*

To state the variant of stochastic Halpern iteration for constrained settings, we also define the operator mapping corresponding to the stochastic estimate $\widetilde{F}$ by

$$\widetilde{G}_\eta(\mathbf{u}) = \eta\Big(\mathbf{u} - \Pi_{\mathcal{U}}\big(\mathbf{u} - \frac{1}{\eta}\widetilde{F}(\mathbf{u})\big)\Big). \tag{C.8}$$

In the following lemma, we bound the error between the stochastic operator mapping and true operator mapping by the variance of stochastic queries.

**Lemma C.4.** *Let $G_\eta(\cdot)$ and $\widetilde{G}_\eta(\cdot)$ be defined as in Eq. (C.7) and Eq. (C.8), respectively. Then, for any $\mathbf{u} \in \mathcal{U}$ and any $\eta > 0$, we have*

$$\|G_\eta(\mathbf{u}) - \widetilde{G}_\eta(\mathbf{u})\|^2 \leq \|F(\mathbf{u}) - \widetilde{F}(\mathbf{u})\|^2. \tag{C.9}$$

*Proof.* By the definition of gradient mapping, we have

$$\Big\|G_\eta(\mathbf{u}) - \widetilde{G}_\eta(\mathbf{u})\Big\|^2 = \eta^2\Big\|\Pi_{\mathcal{U}}\big(\mathbf{u} - \frac{1}{\eta}F(\mathbf{u})\big) - \Pi_{\mathcal{U}}\big(\mathbf{u} - \frac{1}{\eta}\widetilde{F}(\mathbf{u})\big)\Big\|^2.$$

Since the projection operator is non-expansive, we obtain

$$\left\| G_\eta(\mathbf{u}) - \widetilde{G}_\eta(\mathbf{u}) \right\|^2 \leq \eta^2 \left\| \left(\mathbf{u} - \frac{1}{\eta}F(\mathbf{u})\right) - \left(\mathbf{u} - \frac{1}{\eta}\widetilde{F}(\mathbf{u})\right) \right\|^2 = \left\| F(\mathbf{u}) - \widetilde{F}(\mathbf{u}) \right\|^2,$$

thus completing the proof. $\qquad\qquad\qquad\qquad\qquad\qquad\qquad\qquad\qquad\qquad\qquad\qquad\square$

Similar to the unconstrained setup, we define the following stochastic Halpern iteration for the constrained setup:

$$\mathbf{u}_{k+1} = \lambda_{k+1}\mathbf{u}_0 + (1 - \lambda_{k+1})\big(\mathbf{u}_k - \widetilde{G}_{L_k}(\mathbf{u}_k)/L_{k+1}\big), \tag{C.10}$$

where $L_k \geq L$, $\forall k \geq 0$. By the cocoercivity of the operator mapping and the error bound in Lemma C.4, we can immediately obtain the results for the iteration complexity and stochastic oracle complexity as in the unconstrained case, by applying Theorem 3.1 and Corollary 3.4 to $G_L$ and $\widetilde{G}_L$. This is summarized in the following Theorem C.6 and Corollary C.7. To prove these, we make use of the potential function as in the unconstrained settings

$$\mathcal{C}_k = \frac{A_k}{2L_k}\|G_{L_k}(\mathbf{u}_k)\|^2 + B_k \langle G_{L_k}(\mathbf{u}_k), \mathbf{u}_k - \mathbf{u}_0 \rangle, \tag{C.11}$$

and first bound the change of $\mathcal{C}_k$ in the following Lemma C.5. For short, we denote $G_L$ as $G$ below.

**Lemma C.5.** *Let $\mathcal{C}_k$ be defined as in Eq. (C.11), where $A_k$ and $B_k$ satisfy Assumption 4. Assume that $L$ is already known and we set $L_k = L$ for any $k \geq 1$. Then for any $k \geq 2$, we have*

$$\mathcal{C}_k - \mathcal{C}_{k-1} \leq \frac{A_k}{L}\left\| G(\mathbf{u}_{k-1}) - \widetilde{G}(\mathbf{u}_{k-1}) \right\|^2 + \frac{A_k - A_{k-1}}{2L}\left\langle G(\mathbf{u}_{k-1}), G(\mathbf{u}_{k-1}) - \widetilde{G}(\mathbf{u}_{k-1}) \right\rangle.$$

*Proof.* By the definition of $\mathcal{C}_k$, we have

$$\begin{aligned}
\mathcal{C}_k - \mathcal{C}_{k-1} = {}& \frac{A_k}{2L_k}\|G_{L_k}(\mathbf{u}_k)\|^2 + B_k \langle G_{L_k}(\mathbf{u}_k), \mathbf{u}_k - \mathbf{u}_0 \rangle \\
& - \frac{A_{k-1}}{2L_{k-1}}\|G_{L_{k-1}}(\mathbf{u}_{k-1})\|^2 - B_{k-1} \langle G_{L_{k-1}}(\mathbf{u}_{k-1}), \mathbf{u}_{k-1} - \mathbf{u}_0 \rangle.
\end{aligned}$$

Since $G_{L_k}$ is cocoercive with parameter $\frac{3}{4L_k}$ when $L_k \geq L$, we have

$$\begin{aligned}
& \langle G_{L_k}(\mathbf{u}_k) - G_{L_k}(\mathbf{u}_{k-1}), \mathbf{u}_k - \mathbf{u}_{k-1} \rangle \\
\geq {}& \frac{3}{4L_k}\|G_{L_k}(\mathbf{u}_k) - G_{L_k}(\mathbf{u}_{k-1})\|^2 \\
= {}& \frac{1}{2L_k}\Big( \|G_{L_k}(\mathbf{u}_k)\|^2 - 2\langle G_{L_k}(\mathbf{u}_k), G_{L_k}(\mathbf{u}_{k-1}) \rangle + \|G_{L_k}(\mathbf{u}_{k-1})\|^2 \Big) \\
& + \frac{1}{4L_k}\|G_{L_k}(\mathbf{u}_k) - G_{L_k}(\mathbf{u}_{k-1})\|^2.
\end{aligned}$$

Multiplying $A_k$ on both sides and rearranging the terms, we obtain

$$\begin{aligned}
\frac{A_k}{2L_k}\|G_{L_k}(\mathbf{u}_k)\|^2 \leq {}& \left\langle G_{L_k}(\mathbf{u}_k), A_k(\mathbf{u}_k - \mathbf{u}_{k-1}) + \frac{A_k}{L_k}G_{L_k}(\mathbf{u}_{k-1}) \right\rangle \\
& - \langle G_{L_k}(\mathbf{u}_{k-1}), A_k(\mathbf{u}_k - \mathbf{u}_{k-1}) \rangle \\
& - \frac{A_k}{2L_k}\|G_{L_k}(\mathbf{u}_{k-1})\|^2 - \frac{A_k}{4L_k}\|G_{L_k}(\mathbf{u}_k) - G_{L_k}(\mathbf{u}_{k-1})\|^2.
\end{aligned}$$

Plugging this into $\mathcal{C}_k - \mathcal{C}_{k-1}$, we have

$$\begin{aligned}
\mathcal{C}_k - \mathcal{C}_{k-1} \leq {}& \left\langle G_{L_k}(\mathbf{u}_k), A_k(\mathbf{u}_k - \mathbf{u}_{k-1}) + \frac{A_k}{L_k}G_{L_k}(\mathbf{u}_{k-1}) + B_k(\mathbf{u}_k - \mathbf{u}_0) \right\rangle \\
& - \langle G_{L_k}(\mathbf{u}_{k-1}), A_k(\mathbf{u}_k - \mathbf{u}_{k-1}) \rangle + \langle G_{L_{k-1}}(\mathbf{u}_{k-1}), B_{k-1}(\mathbf{u}_{k-1} - \mathbf{u}_0) \rangle \\
& - \left( \frac{A_k}{2L_k}\|G_{L_k}(\mathbf{u}_{k-1})\|^2 + \frac{A_{k-1}}{2L_{k-1}}\|G_{L_{k-1}}(\mathbf{u}_{k-1})\|^2 \right) \\
& - \frac{A_k}{4L_k}\|G_{L_k}(\mathbf{u}_k) - G_{L_k}(\mathbf{u}_{k-1})\|^2.
\end{aligned}$$

Since $\lambda_k = \frac{B_k}{A_k + B_k}$, we have

$$\mathbf{u}_k = \frac{B_k}{A_k + B_k}\mathbf{u}_0 + \frac{A_k}{A_k + B_k}\Big(\mathbf{u}_{k-1} - \widetilde{G}_{L_{k-1}}(\mathbf{u}_{k-1})/L_k\Big),$$

which leads to $A_k(\mathbf{u}_k - \mathbf{u}_{k-1}) + \frac{A_k}{L_k}G_{L_k}(\mathbf{u}_{k-1}) + B_k(\mathbf{u}_k - \mathbf{u}_0) = \frac{A_k}{L_k}\Big(G_{L_{k-1}}(\mathbf{u}_{k-1}) - \widetilde{G}_{L_{k-1}}(\mathbf{u}_{k-1})\Big)$.

Further, as $\frac{B_{k-1}}{A_k} = \frac{B_k}{A_k + B_k}$ by https://www.overleaf.com/project/5fe36b9ad2991b26777b720dAssumption 4, we have

$$\langle G_{L_k}(\mathbf{u}_{k-1}), A_k(\mathbf{u}_k - \mathbf{u}_{k-1}) + B_{k-1}(\mathbf{u}_{k-1} - \mathbf{u}_0)\rangle$$
$$= A_k\left\langle G_{L_k}(\mathbf{u}_{k-1}), \mathbf{u}_k - \mathbf{u}_{k-1} + \frac{B_{k-1}}{A_k}(\mathbf{u}_{k-1} - \mathbf{u}_0)\right\rangle$$
$$= A_k\left\langle G_{L_k}(\mathbf{u}_{k-1}), \mathbf{u}_k - \frac{A_k}{A_k + B_k}\mathbf{u}_{k-1} - \frac{B_k}{A_k + B_k}\mathbf{u}_0\right\rangle$$
$$= A_k\left\langle G_{L_k}(\mathbf{u}_{k-1}), -\frac{A_k}{A_k + B_k}\widetilde{G}_{L_{k-1}}(\mathbf{u}_{k-1})/L_k\right\rangle.$$

Moreover, by Assumption 4, we have $\frac{1}{L_k}\big(1 - \frac{2B_k}{A_k + B_k}\big) = \frac{A_{k-1}}{A_k L_{k-1}}$, so we obtain

$$\langle G_{L_k}(\mathbf{u}_{k-1}), A_k(\mathbf{u}_k - \mathbf{u}_{k-1}) + B_{k-1}(\mathbf{u}_{k-1} - \mathbf{u}_0)\rangle$$
$$= A_k\left\langle G_{L_k}(\mathbf{u}_{k-1}), -\frac{A_k}{A_k + B_k}\widetilde{G}_{L_{k-1}}(\mathbf{u}_{k-1})/L_k\right\rangle$$
$$= -\frac{1}{2}\left\langle G_{L_k}(\mathbf{u}_{k-1}), \Big(\frac{A_k}{L_k} + \frac{A_{k-1}}{L_{k-1}}\Big)\widetilde{G}_{L_{k-1}}(\mathbf{u}_{k-1})\right\rangle.$$

Having $L_k = L$ and denoting $G_L = G$ for short, we have

$$\mathcal{C}_k - \mathcal{C}_{k-1} \le \left\langle G(\mathbf{u}_k), \frac{A_k}{L}\Big(G(\mathbf{u}_{k-1}) - \widetilde{G}(\mathbf{u}_{k-1})\Big)\right\rangle + \left\langle G(\mathbf{u}_{k-1}), \frac{A_k + A_{k-1}}{2L}\widetilde{G}(\mathbf{u}_{k-1})\right\rangle$$
$$\quad - \frac{A_k + A_{k-1}}{2L}\|G(\mathbf{u}_{k-1})\|^2 - \frac{A_k}{4L}\|G(\mathbf{u}_k) - G(\mathbf{u}_{k-1})\|^2$$
$$= \frac{A_k}{L}\left\langle G(\mathbf{u}_k), G(\mathbf{u}_{k-1}) - \widetilde{G}(\mathbf{u}_{k-1})\right\rangle - \frac{A_k}{4L}\|G(\mathbf{u}_k) - G(\mathbf{u}_{k-1})\|^2$$
$$\quad - \frac{A_k + A_{k-1}}{2L}\left\langle G(\mathbf{u}_{k-1}), G(\mathbf{u}_{k-1}) - \widetilde{G}(\mathbf{u}_{k-1})\right\rangle$$
$$= \frac{A_k}{L}\left\langle G(\mathbf{u}_k) - G(\mathbf{u}_{k-1}), G(\mathbf{u}_{k-1}) - \widetilde{G}(\mathbf{u}_{k-1})\right\rangle - \frac{A_k}{4L}\|G(\mathbf{u}_k) - G(\mathbf{u}_{k-1})\|^2$$
$$\quad + \frac{A_k - A_{k-1}}{2L}\left\langle G(\mathbf{u}_{k-1}), G(\mathbf{u}_{k-1}) - \widetilde{G}(\mathbf{u}_{k-1})\right\rangle.$$

Since $2\langle p, q\rangle + \|p\|^2 \le \|q\|^2$ for any $p, q \in \mathbb{R}^d$, we have

$$\mathcal{C}_k - \mathcal{C}_{k-1} \le \frac{A_k}{L}\left\|G(\mathbf{u}_{k-1}) - \widetilde{G}(\mathbf{u}_{k-1})\right\|^2 + \frac{A_k - A_{k-1}}{2L}\left\langle G(\mathbf{u}_{k-1}), G(\mathbf{u}_{k-1}) - \widetilde{G}(\mathbf{u}_{k-1})\right\rangle,$$

thus completing the proof. □

**Theorem C.6.** *Given an arbitrary $\mathbf{u}_0 \in \mathbb{R}^d$, suppose that iterates $\mathbf{u}_k$ evolve according to Halpern iteration for the constrained setup from Eq. (C.10) for $k \ge 1$, where $L_k = L$ and $\lambda_k = \frac{1}{k+1}$. Given $\epsilon > 0$, if we have that $\mathbb{E}[\|F(\mathbf{u}_0) - \widetilde{F}(\mathbf{u}_0)\|^2] \le \frac{\epsilon^2}{8}$ and $\mathbb{E}\big[\big\|F(\mathbf{u}_k) - \widetilde{F}(\mathbf{u}_k)\big\|^2\big] \le \frac{\epsilon^2}{k}$ for all $k \ge 1$, then for all $k \ge 1$,*

$$\mathbb{E}[\|G(\mathbf{u}_k))\|] \le \frac{\Lambda_0}{k} + \Lambda_1\epsilon, \tag{C.12}$$

*where $\Lambda_0 = 20L\|\mathbf{u}_0 - \mathbf{u}^*\|$ and $\Lambda_1 = \sqrt{13}$. As a result, stochastic Halpern iteration from Eq. (3.1) returns a point $\mathbf{u}_k$ such that $\mathbb{E}[\|G(\mathbf{u}_k)\|] \le 5\epsilon$ after at most $N = \lceil\frac{2\Lambda_0}{\epsilon}\rceil = \mathcal{O}\big(\frac{L\|\mathbf{u}_0 - \mathbf{u}^*\|}{\epsilon}\big)$ iterations.*

*Proof.* First note that since $\mathcal{U}$ is convex and closed, and $\mathbf{u}_k - \widetilde{G}(\mathbf{u}_k)/L = \Pi_{\mathcal{U}}\left(\mathbf{u}_k - \frac{1}{L}\widetilde{F}(\mathbf{u}_k)\right)$, then we have for $\forall k > 0$.

$$\mathbf{u}_{k+1} = \lambda_{k+1}\mathbf{u}_0 + (1 - \lambda_{k+1})\left(\mathbf{u}_k - \widetilde{G}(\mathbf{u}_k)/L\right)$$

$$= \lambda_{k+1}\mathbf{u}_0 + (1 - \lambda_{k+1})\Pi_{\mathcal{U}}\left(\mathbf{u}_k - \frac{1}{L}\widetilde{F}(\mathbf{u}_k)\right) \in \mathcal{U}.$$

Then we come to prove the convergence. By Jensen's Inequality, we have for $k \geq 1$

$$\mathbb{E}[\|G(\mathbf{u}_k))\|] \leq \left(\mathbb{E}[\|G(\mathbf{u}_k)\|^2]\right)^{\frac{1}{2}}.$$

So it suffices to show that there exists $\Lambda_0$ and $\Lambda_1$ such that for all $k \geq 1$

$$\left(\mathbb{E}[\|G(\mathbf{u}_k)\|^2]\right)^{\frac{1}{2}} \leq \frac{\Lambda_0}{k} + \Lambda_1\epsilon.$$

We prove it by induction. First, we consider the basis case $k = 1$ in which $\mathbf{u}_1 = \mathbf{u}_0 - \frac{1}{2L}\widetilde{G}(\mathbf{u}_0)$, so we have $\mathcal{C}_1 = \frac{1}{L}\|G(\mathbf{u}_1)\|^2 + 2\langle G(\mathbf{u}_1), \mathbf{u}_1 - \mathbf{u}_0\rangle = \frac{1}{L}\left(\|G(\mathbf{u}_1)\|^2 - \langle G(\mathbf{u}_1), \widetilde{G}(\mathbf{u}_0)\rangle\right)$. Also, since the operator $G$ is cocoercive with parameter $\frac{3}{4L}$, thus cocoercive with $\frac{1}{2L}$, we have

$$\|G(\mathbf{u}_1) - G(\mathbf{u}_0)\|^2 \leq 2L\langle G(\mathbf{u}_1) - G(\mathbf{u}_0), \mathbf{u}_1 - \mathbf{u}_0\rangle = \langle G(\mathbf{u}_1) - G(\mathbf{u}_0), -\widetilde{G}(\mathbf{u}_0)\rangle.$$

Expanding and rearranging the terms, we have

$$\|G(\mathbf{u}_1)\|^2 \leq \langle G(\mathbf{u}_0), \widetilde{G}(\mathbf{u}_0) - G(\mathbf{u}_0)\rangle + 2\langle G(\mathbf{u}_1), G(\mathbf{u}_0)\rangle - \langle G(\mathbf{u}_1), \widetilde{G}(\mathbf{u}_0)\rangle.$$

Subtracting $\langle G(\mathbf{u}_1), \widetilde{G}(\mathbf{u}_0)\rangle$ and taking expectation with respect to all randomness on both sides, we have

$$\mathbb{E}\left[\|G(\mathbf{u}_1)\|^2 - \langle G(\mathbf{u}_1), \widetilde{G}(\mathbf{u}_0)\rangle\right]$$

$$\leq \mathbb{E}\left[\langle G(\mathbf{u}_0), \widetilde{G}(\mathbf{u}_0) - G(\mathbf{u}_0)\rangle + 2\langle G(\mathbf{u}_1), G(\mathbf{u}_0)\rangle - 2\langle G(\mathbf{u}_1), \widetilde{G}(\mathbf{u}_0)\rangle\right]$$

$$\overset{(i)}{\leq} \mathbb{E}\left[\frac{1}{2}\|G(\mathbf{u}_0)\|^2 + \frac{1}{2}\|G(\mathbf{u}_1)\|^2 + \frac{5}{2}\left\|G(\mathbf{u}_0) - \widetilde{G}(\mathbf{u}_0)\right\|^2\right],$$

where for $(i)$ we use Young's Inequality. Since $\mathbf{u}^*$ is the solution of monotone inclusion, then we have $G(\mathbf{u}^*) = 0$. So we have

$$\|G(\mathbf{u}_0)\|^2 = \|G(\mathbf{u}_0) - G(\mathbf{u}^*)\|^2 \overset{(i)}{\leq} 10L^2\|\mathbf{u}_0 - \mathbf{u}^*\|^2,$$

where $(i)$ can be verified by Young's Inequality and using the fact that the projection operator is non-expansive. Also using the results in Lemma C.4, we obtain that

$$\mathbb{E}[\mathcal{C}_1] \leq \frac{1}{L}\mathbb{E}\left[5L^2\|\mathbf{u}_0 - \mathbf{u}^*\|^2 + \frac{1}{2}\|G(\mathbf{u}_1)\|^2 + \frac{5}{2}\left\|F(\mathbf{u}_0) - \widetilde{F}(\mathbf{u}_0)\right\|^2\right].$$

Proceeding similar to Lemma C.1, we have

$$\mathbb{E}[\|G(\mathbf{u}_1)\|^2] \leq \frac{2B_1 L_1}{A_1}\|\mathbf{u}_0 - \mathbf{u}^*\|\mathbb{E}[\|G(\mathbf{u}_1)\|]$$

$$+ \frac{2L_1}{A_1 L}\mathbb{E}\left[5L^2\|\mathbf{u}_0 - \mathbf{u}^*\|^2 + \frac{1}{2}\|G(\mathbf{u}_1)\|^2 + \frac{5}{2}\left\|F(\mathbf{u}_0) - \widetilde{F}(\mathbf{u}_0)\right\|^2\right]$$

$$= 2L\|\mathbf{u}_0 - \mathbf{u}^*\|\mathbb{E}[\|G(\mathbf{u}_1)\|]$$

$$+ \mathbb{E}\left[5L^2\|\mathbf{u}_0 - \mathbf{u}^*\|^2 + \frac{1}{2}\|G(\mathbf{u}_1)\|^2 + \frac{5}{2}\left\|F(\mathbf{u}_0) - \widetilde{F}(\mathbf{u}_0)\right\|^2\right].$$

Subtracting $\mathbb{E}[\frac{1}{2}\|G(\mathbf{u}_1)\|^2]$ on both sides and using the fact that $\mathbb{E}[\|G(\mathbf{u}_1)\|] \leq \left(\mathbb{E}[\|G(\mathbf{u}_1)\|^2]\right)^{\frac{1}{2}}$ and $\mathbb{E}[\left\|F(\mathbf{u}_0) - \widetilde{F}(\mathbf{u}_0)\right\|^2] \leq \frac{\epsilon^2}{8}$, we have

$$\mathbb{E}[\|G(\mathbf{u}_1)\|^2] \leq 4L\|\mathbf{u}_0 - \mathbf{u}^*\|\left(\mathbb{E}[\|G(\mathbf{u}_1)\|^2]\right)^{\frac{1}{2}} + 10L^2\|\mathbf{u}_0 - \mathbf{u}^*\|^2 + \frac{5\epsilon^2}{8},$$

which is a quadratic function with respect to $(\mathbb{E}[\|G(\mathbf{u}_1)\|^2])^{\frac{1}{2}}$. So by its larger root we have

$$(\mathbb{E}[\|G(\mathbf{u}_1)\|^2])^{\frac{1}{2}} \leq 2L\|\mathbf{u}_0 - \mathbf{u}^*\| + \frac{1}{2}\sqrt{56L^2\|\mathbf{u}_0 - \mathbf{u}^*\|^2 + \frac{5}{2}\epsilon^2}$$

$$\leq 2L\|\mathbf{u}_0 - \mathbf{u}^*\| + \frac{1}{2}(2\sqrt{14}L\|\mathbf{u}_0 - \mathbf{u}^*\| + \frac{\sqrt{10}}{2}\epsilon)$$

$$\leq 6L\|\mathbf{u}_0 - \mathbf{u}^*\| + \epsilon \leq \Lambda_0 + \Lambda_1\epsilon.$$

So the result holds for the basis case. Moreover, we can get a bound for $\mathbb{E}[\mathcal{C}_1]$ as follows

$$\mathbb{E}[\mathcal{C}_1] \leq \frac{1}{L}\mathbb{E}\Big[5L^2\|\mathbf{u}_0 - \mathbf{u}^*\|^2 + \frac{1}{2}\|G(\mathbf{u}_1)\|^2 + \frac{5}{2}\left\|F(\mathbf{u}_0) - \widetilde{F}(\mathbf{u}_0)\right\|^2\Big]$$

$$\overset{(i)}{\leq} 5L\|\mathbf{u}_0 - \mathbf{u}^*\|^2 + \frac{1}{2L}\Big(50L^2\|\mathbf{u}_0 - \mathbf{u}^*\|^2 + \frac{5}{4}\epsilon^2\Big) + \frac{5}{2L}\frac{\epsilon^2}{8}$$

$$\leq 30L\|\mathbf{u}_0 - \mathbf{u}^*\|^2 + \frac{\epsilon^2}{L},$$

where $(i)$ can be verified by using the bound we get above for $\mathbb{E}[\|G(\mathbf{u}_1)\|^2]$ and applying Young's Inequaltiy, and the fact that $\mathbb{E}[\left\|F(\mathbf{u}_0) - \widetilde{F}(\mathbf{u}_0)\right\|^2] \leq \frac{\epsilon^2}{8}$.

Assume that the result holds for all $1 \leq i \leq k-1$, then we come to prove the case $k$. By Lemma C.5 we have for $\forall i \geq 2$

$$\mathcal{C}_i - \mathcal{C}_{i-1} \leq \frac{A_i}{L}\left\|G(\mathbf{u}_{i-1}) - \widetilde{G}(\mathbf{u}_{i-1})\right\|^2 + \frac{A_i - A_{i-1}}{2L}\left\langle G(\mathbf{u}_{i-1}), G(\mathbf{u}_{i-1}) - \widetilde{G}(\mathbf{u}_{i-1})\right\rangle$$

$$\overset{(i)}{\leq} \frac{3i(i+1)}{L}\left\|G(\mathbf{u}_{i-1}) - \widetilde{G}(\mathbf{u}_{i-1})\right\|^2 + \frac{i}{8L(i+1)}\|G(\mathbf{u}_{i-1})\|^2$$

$$\overset{(ii)}{\leq} \frac{3i(i+1)}{L}\left\|F(\mathbf{u}_{i-1}) - \widetilde{F}(\mathbf{u}_{i-1})\right\|^2 + \frac{i}{8L(i+1)}\|G(\mathbf{u}_{i-1})\|^2,$$

where we use Young's Inequality and $A_i = i(i+1)$ for $(i)$, and $(ii)$ is due to Lemma C.4. Taking expectation with respect to all randomness on both sides and telescoping from $i = 2$ to $k$, we obtain

$$\mathbb{E}[\mathcal{C}_k] \leq \mathbb{E}\Big[\mathcal{C}_1 + \sum_{i=2}^{k}\Big(\frac{3i(i+1)}{L}\left\|F(\mathbf{u}_{i-1}) - \widetilde{F}(\mathbf{u}_{i-1})\right\|^2 + \frac{i}{8L(i+1)}\|G(\mathbf{u}_{i-1})\|^2\Big)\Big]$$

$$\leq 30L\|\mathbf{u}_0 - \mathbf{u}^*\|^2 + \frac{\epsilon^2}{L} + \mathbb{E}\Big[\sum_{i=2}^{k}\frac{i}{8L(i+1)}\|G(\mathbf{u}_{i-1})\|^2\Big]$$

$$+ \mathbb{E}\Big[\sum_{i=2}^{k}\frac{3i(i+1)}{L}\left\|F(\mathbf{u}_{i-1}) - \widetilde{F}(\mathbf{u}_{i-1})\right\|^2\Big].$$

By Corollary 2.2, we have

$$\mathbb{E}\Big[\sum_{i=2}^{k}\frac{3i(i+1)}{L}\left\|F(\mathbf{u}_{i-1}) - \widetilde{F}(\mathbf{u}_{i-1})\right\|^2\Big]$$

$$\leq \sum_{i=2}^{k}\frac{3i(i+1)}{L}\frac{\epsilon^2}{i-1} \overset{(i)}{\leq} \sum_{i=2}^{k}\frac{6(i+1)\epsilon^2}{L} = \frac{3(k+4)(k-1)\epsilon^2}{L},$$

where $(i)$ is because $\frac{i}{i-1} \leq 2$ for all $i \geq 2$. By induction, we have

$$\mathbb{E}\Big[\sum_{i=2}^{k}\frac{i}{8L(i+1)}\|G(\mathbf{u}_{i-1})\|^2\Big]$$

$$\overset{(i)}{\leq} \sum_{i=2}^{k}\frac{1}{8L}\Big(2\frac{\Lambda_0^2}{(i-1)^2} + 2\Lambda_1^2\epsilon^2\Big)$$

$$\overset{(ii)}{\leq} \frac{1}{4L}\Big(\Lambda_0^2\frac{\pi^2}{6} + (k-1)\Lambda_1^2\epsilon^2\Big) = \frac{1}{L}\Big(\frac{\Lambda_0^2\pi^2}{24} + \frac{(k-1)\Lambda_1^2\epsilon^2}{4}\Big),$$

where $(i)$ follows from induction and $\frac{i}{i+1} \leq 1$, and $(ii)$ is due to $\sum_{i=2}^{k} \frac{1}{(i-1)^2} \leq \sum_{i=1}^{\infty} \frac{1}{i^2} = \frac{\pi^2}{6}$.
We now obtain

$$\mathbb{E}[\mathcal{C}_k] \leq 30L \left\| \mathbf{u}_0 - \mathbf{u}^* \right\|^2 + \frac{\epsilon^2}{L} + \frac{3(k+4)(k-1)\epsilon^2}{L} + \frac{1}{L}\left( \frac{\Lambda_0^2\pi^2}{24} + \frac{(k-1)\Lambda_1^2\epsilon^2}{4} \right).$$

By the same derivation of Lemma C.1, we have

$$
\begin{aligned}
\mathbb{E}\big[ \left\| G(\mathbf{u}_k) \right\|^2 \big] &\leq \frac{B_k L_k}{A_k} \left\| \mathbf{u}_0 - \mathbf{u}^* \right\| \mathbb{E}[\left\| G(\mathbf{u}_k) \right\|] \\
&\quad + \frac{L_k}{A_k}\left( 30L \left\| \mathbf{u}_0 - \mathbf{u}^* \right\|^2 + \frac{\epsilon^2}{L} + \frac{3(k+4)(k-1)\epsilon^2}{L} + \frac{\Lambda_0^2\pi^2}{24L} + \frac{(k-1)\Lambda_1^2\epsilon^2}{4L} \right) \\
&= \frac{L}{k} \left\| \mathbf{u}_0 - \mathbf{u}^* \right\| \mathbb{E}[\left\| G(\mathbf{u}_k) \right\|] + \frac{30L^2 \left\| \mathbf{u}_0 - \mathbf{u}^* \right\|^2}{k(k+1)} \\
&\quad + \frac{1}{k(k+1)}\left( \epsilon^2 + 3(k+4)(k-1)\epsilon^2 + \frac{\Lambda_0^2\pi^2}{24} + \frac{(k-1)\Lambda_1^2\epsilon^2}{4} \right) \\
&\overset{(i)}{\leq} \frac{L}{k} \left\| \mathbf{u}_0 - \mathbf{u}^* \right\| \mathbb{E}[\left\| G(\mathbf{u}_k) \right\|] + \left( \frac{30L^2 \left\| \mathbf{u}_0 - \mathbf{u}^* \right\|^2}{k^2} + (11 + \frac{\Lambda_1^2}{4k})\epsilon^2 + \frac{\Lambda_0^2\pi^2}{24k^2} \right),
\end{aligned}
$$

where $(i)$ is due to $\frac{1}{k(k+1)} \leq \frac{1}{k^2}$, $\frac{3(k+1)(k-1)}{k(k+1)} \leq 10$ and $\frac{k-1}{k(k+1)} \leq \frac{1}{k}$. Since $\mathbb{E}[\left\| G(\mathbf{u}_k) \right\|] \leq (\mathbb{E}[\left\| G(\mathbf{u}_k) \right\|^2])^{\frac{1}{2}}$ by Jensen's Inequality, we have

$$
\begin{aligned}
\mathbb{E}\big[ \left\| G(\mathbf{u}_k) \right\|^2 \big] &\leq \frac{L}{k} \left\| \mathbf{u}_0 - \mathbf{u}^* \right\| \left( \mathbb{E}\big[ \left\| G(\mathbf{u}_k) \right\|^2 \big] \right)^{\frac{1}{2}} \\
&\quad + \left( \frac{30L^2 \left\| \mathbf{u}_0 - \mathbf{u}^* \right\|^2}{k^2} + (11 + \frac{\Lambda_1^2}{4k})\epsilon^2 + \frac{\Lambda_0^2\pi^2}{24k^2} \right),
\end{aligned}
$$

which is a quadratic function with respect to $(\mathbb{E}[\left\| G(\mathbf{u}_k) \right\|^2])^{\frac{1}{2}}$. So by its larger root we obtain

$$
\begin{aligned}
\left( \mathbb{E}\big[ \left\| G(\mathbf{u}_k) \right\|^2 \big] \right)^{\frac{1}{2}} &\\
&\hspace{-2cm}\leq \frac{L}{2k} \left\| \mathbf{u}_0 - \mathbf{u}^* \right\| + \frac{1}{2}\sqrt{ \frac{L^2}{k^2} \left\| \mathbf{u}_0 - \mathbf{u}^* \right\|^2 + 4\left( \frac{30L^2 \left\| \mathbf{u}_0 - \mathbf{u}^* \right\|^2}{k^2} + (11 + \frac{\Lambda_1^2}{4k})\epsilon^2 + \frac{\Lambda_0^2\pi^2}{24k^2} \right) } \\
&\hspace{-2cm}\overset{(i)}{\leq} \frac{L}{k} \left\| \mathbf{u}_0 - \mathbf{u}^* \right\| + \left( \frac{\sqrt{30}L \left\| \mathbf{u}_0 - \mathbf{u}^* \right\|}{k} + \sqrt{11 + \frac{\Lambda_1^2}{4k}}\epsilon + \frac{\Lambda_0\pi}{2\sqrt{6}k} \right) \\
&\hspace{-2cm}= \frac{(1+\sqrt{30})L \left\| \mathbf{u}_0 - \mathbf{u}^* \right\| + \frac{\Lambda_0\pi}{2\sqrt{6}}}{k} + \sqrt{11 + \frac{\Lambda_1^2}{4k}}\epsilon \\
&\hspace{-2cm}\overset{(ii)}{\leq} \frac{\Lambda_0}{k} + \Lambda_1\epsilon,
\end{aligned}
$$

where $(i)$ is due to the fact that $\sqrt{\sum_{i=1}^{n} X_i^2} \leq \sum_{i=1}^{n} |X_i|$, and $(ii)$ is because of our choice of $\Lambda_0, \Lambda_1$. Hence, the result also holds for the case $k$. Then by induction we know that the result holds for all $k \geq 1$.

Finally, when $k \geq \frac{2\Lambda_0}{\epsilon}$, we have $\frac{\Lambda_0}{k} \leq \epsilon/2$. Also, since we have $\Lambda_1 = \sqrt{13}$, we obtain

$$\mathbb{E}[\left\| G(\mathbf{u}_k) \right\|] \leq \left( \frac{1}{2} + \sqrt{13} \right)\epsilon \leq 5\epsilon.$$

Hence, the total number of iterations needed to attain $5\epsilon$ norm of the operator is

$$N = \left\lceil \frac{2\Lambda_0}{\epsilon} \right\rceil \leq \frac{2\Lambda_0}{\epsilon} + 1 = \frac{\Delta}{\epsilon},$$

thus completing the proof. $\qquad \square$

**Corollary C.7.** *Given an arbitrary $\mathbf{u}_0 \in \mathbb{R}^d$, suppose that iterates $\mathbf{u}_k$ evolve according to Halpern iteration from Eq. (C.10) for $k \geq 1$, where $L_k = L$, and $\lambda_k = \frac{1}{k+1}$. Assume further that the*

*stochastic estimate $\widetilde{F}(\mathbf{u})$ is defined according to Eq. (2.1), with its parameters set according to Corollary 2.2. Then, given any $\epsilon > 0$, stochastic Halpern iteration from Eq. (C.10) returns a point $\mathbf{u}_k$ such that $\mathbb{E}[\|G(\mathbf{u}_k)\|] \leq 5\epsilon$ with at most $\mathcal{O}(\frac{\sigma^2 L\|\mathbf{u}_0 - \mathbf{u}^*\| + L^3\|\mathbf{u}_0 - \mathbf{u}^*\|^3}{\epsilon^3})$ oracle queries to $\widehat{F}$ in expectation.*

*Proof.* Let $m_k$ be the number of stochastic queries made by the estimator from Eq. (2.1) at iteration $k$. Conditional on $\mathcal{F}_k$ and using Corollary 2.2, since each stochastic gradient mapping $\widetilde{G}(\mathbf{u}_k)$ only involves one PAGE invariant stochastic estimate $\widetilde{F}(\mathbf{u}_k)$, we have

$$\mathbb{E}[m_{k+1}|\mathcal{F}_{k-1}] = p_k \left\lceil \frac{8\sigma^2}{p_k \epsilon^2} \right\rceil + 2(1 - p_k)\left\lceil \frac{8L^2 \|\mathbf{u}_k - \mathbf{u}_{k-1}\|^2}{p_k^2 \epsilon^2} \right\rceil$$

$$\overset{(i)}{\leq} p_k \left( \frac{8\sigma^2}{p_k \epsilon^2} + 1 \right) + 2(1 - p_k)\left( \frac{8L^2 \|\mathbf{u}_k - \mathbf{u}_{k-1}\|^2}{p_k^2 \epsilon^2} + 1 \right),$$

where $(i)$ is due to the fact that $\lceil x \rceil \leq x + 1$ for any $x \in \mathbb{R}$. Taking expectation with respect to all randomness on both sides, and rearranging the terms, we obtain

$$\mathbb{E}[m_{k+1}] \leq \frac{8\sigma^2}{\epsilon^2} + \frac{16(1 - p_k)L^2 \mathbb{E}[\|\mathbf{u}_k - \mathbf{u}_{k-1}\|^2]}{p_k^2 \epsilon^2} + 2.$$

By the same derivation as Lemma 3.3, we have

$$\|\mathbf{u}_k - \mathbf{u}_{k-1}\|^2 \leq \begin{cases} \frac{1}{4L^2} \left\| \widetilde{G}(\mathbf{u}_0) \right\|^2 & \text{if } k = 1, \\ \frac{2k^2}{L^2(k+1)^2} \left\| \widetilde{G}(\mathbf{u}_{k-1}) \right\|^2 + \sum_{i=0}^{k-2} \frac{2(i+1)^2}{k(k+1)^2 L^2} \left\| \widetilde{G}(\mathbf{u}_i) \right\|^2 & \text{if } k \geq 2. \end{cases} \tag{C.13}$$

By the corollary assumptions, we have $\mathbb{E}[\|G(\mathbf{u}_i)\|^2] \leq \mathcal{O}(\frac{L^2\|\mathbf{u}_0 - \mathbf{u}^*\|^2}{i^2})$ for $i \leq k-1$ by Theorem C.6. Then we obtain

$$\mathbb{E}\left[ \left\| \widetilde{G}(\mathbf{u}_i) \right\|^2 \right] \leq 2\mathbb{E}\left[ \|G(\mathbf{u}_i)\|^2 \right] + 2\mathbb{E}\left[ \left\| \widetilde{G}(\mathbf{u}_i) - G(\mathbf{u}_i) \right\|^2 \right]$$

$$\overset{(i)}{\leq} 2\mathbb{E}\left[ \|G(\mathbf{u}_i)\|^2 \right] + 2\mathbb{E}\left[ \left\| \widetilde{F}(\mathbf{u}_i) - F(\mathbf{u}_i) \right\|^2 \right]$$

$$\leq \mathcal{O}\left( \frac{L^2 \|\mathbf{u}_0 - \mathbf{u}^*\|^2}{i^2} \right),$$

where $(i)$ is due to Lemma C.4.

Plugging it into Inequality (C.13), we have $\mathbb{E}[\|\mathbf{u}_k - \mathbf{u}_{k-1}\|^2] = \mathcal{O}(\frac{\|\mathbf{u}_0 - \mathbf{u}^*\|^2}{k^2})$, which leads to

$$\mathbb{E}[m_{k+1}] = \mathcal{O}\left( \frac{\sigma^2 + L^2 \|\mathbf{u}_0 - \mathbf{u}^*\|^2}{\epsilon^2} \right)$$

using $p_k = \frac{2}{k+1} = \mathcal{O}(1/k)$.

Further, by Theorem C.6, the total number of iterations to attain $5\epsilon$ norm of the operator in expectation is $N = \mathcal{O}(\frac{L\|\mathbf{u}_0 - \mathbf{u}^*\|}{\epsilon})$ and $m_1 = S_1^{(0)} = \mathcal{O}(\frac{\sigma^2}{\epsilon^2})$, we conclude that the total number of stochastic queries to $F$ is

$$\mathbb{E}[M] = \mathbb{E}\left[ \sum_{k=1}^N m_k \right] = \mathcal{O}\left( \frac{\sigma^2 L \|\mathbf{u}_0 - \mathbf{u}^*\| + L^3 \|\mathbf{u}_0 - \mathbf{u}^*\|^3}{\epsilon^3} \right),$$

thus completing the proof. $\qquad\square$

## D  Omitted proofs from Section 4

We use the potential function, previously used by [51],

$$\mathcal{V}_k := A_k \|F(\mathbf{u}_k)\|^2 + B_k \langle F(\mathbf{u}_k), \mathbf{u}_k - \mathbf{u}_0 \rangle + c_k L^2 \|\mathbf{u}_k - \mathbf{v}_{k-1}\|^2, \tag{D.1}$$

prove Theorem 4.1. Here $A_k$, $B_k$ and $c_k$ are positive parameters to be determined later. We start by bounding the change of $\mathcal{V}_k$ under the following assumption on the parameters.

**Assumption 5.** $\lambda_k \in [0, 1)$, $\eta_k > 0$, and $A_k$, $B_k$ and $c_k$ are positive parameters satisfying $B_{k+1} = \frac{B_k}{1-\lambda_k}$, $A_k = \frac{B_k \eta_k}{2\lambda_k}$,

$$0 < \eta_{k+1} = \frac{(1 - \lambda_k^2 - M\eta_k^2)\lambda_{k+1}\eta_k}{(1 - M\eta_k^2)(1 - \lambda_k)\lambda_k}, \quad M\eta_k^2 + \lambda_k^2 < 1, \quad \text{and} \quad \eta_{k+1} \le \frac{\lambda_{k+1}(1 - \lambda_k)}{M\lambda_k\eta_k}, \quad \text{(D.2)}$$

where $M = 3L^2(2 + \theta)$ and $\theta > 0$ is some parameter that can be determined later.

The following lemma gives a bound on the difference between the potential function values at two consecutive iterations with the control of the parameters above.

**Lemma D.1.** *Let $\mathcal{V}_k$ be defined as in Eq. (D.1), where the parameters satisfy Assumption 5. Then the difference of potential function between two consecutive iterations can be bounded by*

$$\mathcal{V}_{k+1} - \mathcal{V}_k \le -L^2\left(\frac{\theta A_k}{M\eta_k^2} - c_{k+1}\right)\|\mathbf{u}_{k+1} - \mathbf{v}_k\|^2 - L^2(c_k - A_k)\|\mathbf{u}_k - \mathbf{v}_{k-1}\|^2$$
$$+ \frac{2A_k}{M\eta_k^2}\left\|F(\mathbf{v}_k) - \widetilde{F}(\mathbf{v}_k)\right\|^2 + A_k\left\|F(\mathbf{v}_{k-1}) - \widetilde{F}(\mathbf{v}_{k-1})\right\|^2. \tag{D.3}$$

*Proof.* By the iteration scheme in Eq. (4.1), we can deduce the following identities:

$$\begin{cases} \mathbf{u}_{k+1} - \mathbf{u}_k = \lambda_k(\mathbf{u}_0 - \mathbf{u}_k) - \eta_k\widetilde{F}(\mathbf{v}_k) \\ \mathbf{u}_{k+1} - \mathbf{u}_k = \dfrac{\lambda_k}{1 - \lambda_k}(\mathbf{u}_0 - \mathbf{u}_{k+1}) - \dfrac{\eta_k}{1 - \lambda_k}\widetilde{F}(\mathbf{v}_k) \\ \mathbf{u}_{k+1} - \mathbf{v}_k = -\eta_k\left(\widetilde{F}(\mathbf{v}_k) - \widetilde{F}(\mathbf{v}_{k-1})\right) \end{cases} \tag{D.4}$$

Further, by the definition of the potential function $\mathcal{V}_k$, we can write

$$\mathcal{V}_k - \mathcal{V}_{k+1} = \underbrace{A_k\|F(\mathbf{u}_k)\|^2 - A_{k+1}\|F(\mathbf{u}_{k+1})\|^2}_{\mathcal{T}_{[1]}}$$
$$+ \underbrace{B_k\langle F(\mathbf{u}_k), \mathbf{u}_k - \mathbf{u}_0\rangle - B_{k+1}\langle F(\mathbf{u}_{k+1}), \mathbf{u}_{k+1} - \mathbf{u}_0\rangle}_{\mathcal{T}_{[2]}} \tag{D.5}$$
$$+ c_k L^2\|\mathbf{u}_k - \mathbf{v}_{k-1}\|^2 - c_{k+1}L^2\|\mathbf{u}_{k+1} - \mathbf{v}_k\|^2.$$

To obtain the claimed bound, in the rest of the proof we focus on bounding $\mathcal{T}_{[1]}$ and $\mathcal{T}_{[2]}$.

To bound $\mathcal{T}_{[1]}$, by the Lipschitz continuity of $F$, we have

$$\|F(\mathbf{u}_{k+1}) - F(\mathbf{v}_k)\|^2 \le L^2\|\mathbf{u}_{k+1} - \mathbf{v}_k\|^2 = L^2\eta_k^2\left\|\widetilde{F}(\mathbf{v}_k) - \widetilde{F}(\mathbf{v}_{k-1})\right\|^2,$$

where in the last step we used the third identity from Eq. (D.4). Further, for any $\theta > 0$

$$\left\|F(\mathbf{u}_{k+1}) - \widetilde{F}(\mathbf{v}_k)\right\|^2 + \theta L^2\|\mathbf{u}_{k+1} - \mathbf{v}_k\|^2$$
$$\le 2\|F(\mathbf{u}_{k+1}) - F(\mathbf{v}_k)\|^2 + 2\left\|F(\mathbf{v}_k) - \widetilde{F}(\mathbf{v}_k)\right\|^2 + \theta L^2\|\mathbf{u}_{k+1} - \mathbf{v}_k\|^2 \tag{D.6}$$
$$\le \eta_k^2 L^2(2 + \theta)\left\|\widetilde{F}(\mathbf{v}_k) - \widetilde{F}(\mathbf{v}_{k-1})\right\|^2 + 2\left\|F(\mathbf{v}_k) - \widetilde{F}(\mathbf{v}_k)\right\|^2,$$

where again in the last step we used the third identity from Eq. (D.4). Notice that

$$\left\|\widetilde{F}(\mathbf{v}_k) - \widetilde{F}(\mathbf{v}_{k-1})\right\|^2$$
$$= \left\|\widetilde{F}(\mathbf{v}_k) - F(\mathbf{u}_k) + F(\mathbf{u}_k) - F(\mathbf{v}_{k-1}) + F(\mathbf{v}_{k-1}) - \widetilde{F}(\mathbf{v}_{k-1})\right\|^2$$
$$\le 3\left\|\widetilde{F}(\mathbf{v}_k) - F(\mathbf{u}_k)\right\|^2 + 3\|F(\mathbf{u}_k) - F(\mathbf{v}_{k-1})\|^2 + 3\left\|F(\mathbf{v}_{k-1}) - \widetilde{F}(\mathbf{v}_{k-1})\right\|^2$$
$$\le 3\left(\|F(\mathbf{u}_k)\|^2 - 2\left\langle F(\mathbf{u}_k), \widetilde{F}(\mathbf{v}_k)\right\rangle + \left\|\widetilde{F}(\mathbf{v}_k)\right\|^2\right) + 3L^2\|\mathbf{u}_k - \mathbf{v}_{k-1}\|^2$$
$$+ 3\left\|F(\mathbf{v}_{k-1}) - \widetilde{F}(\mathbf{v}_{k-1})\right\|^2.$$

Let $M := 3L^2(2 + \theta)$. Expanding the term $\|F(\mathbf{u}_{k+1}) - \widetilde{F}(\mathbf{v}_k)\|^2$ on the LHS in Inequality (D.6) and combining with the inequality above, we have

$$\|F(\mathbf{u}_{k+1})\|^2 + \left\|\widetilde{F}(\mathbf{v}_k)\right\|^2 - 2\left\langle F(\mathbf{u}_{k+1}), \widetilde{F}(\mathbf{v}_k)\right\rangle + \theta L^2 \|\mathbf{u}_{k+1} - \mathbf{v}_k\|^2$$

$$\leq M\eta_k^2 \left(\|F(\mathbf{u}_k)\|^2 - 2\left\langle F(\mathbf{u}_k), \widetilde{F}(\mathbf{v}_k)\right\rangle + \left\|\widetilde{F}(\mathbf{v}_k)\right\|^2\right) + ML^2\eta_k^2 \|\mathbf{u}_k - \mathbf{v}_{k-1}\|^2$$

$$+ M\eta_k^2 \left\|F(\mathbf{v}_{k-1}) - \widetilde{F}(\mathbf{v}_{k-1})\right\|^2 + 2\left\|F(\mathbf{v}_k) - \widetilde{F}(\mathbf{v}_k)\right\|^2.$$

Multiplying both sides by $\frac{A_k}{M\eta_k^2}$, rearranging this inequality and subtracting $A_{k+1}\|F(\mathbf{u}_{k+1})\|^2$ on both sides, we obtain

$$\mathcal{T}_{[1]} = A_k \|F(\mathbf{u}_k)\|^2 - A_{k+1} \|F(\mathbf{u}_{k+1})\|^2$$

$$\geq \left(\frac{A_k}{M\eta_k^2} - A_{k+1}\right) \|F(\mathbf{u}_{k+1})\|^2 + \frac{A_k(1 - M\eta_k^2)}{M\eta_k^2} \left\|\widetilde{F}(\mathbf{v}_k)\right\|^2$$

$$- \frac{2A_k(1 - M\eta_k^2)}{M\eta_k^2} \left\langle F(\mathbf{u}_{k+1}), \widetilde{F}(\mathbf{v}_k)\right\rangle - 2A_k \left\langle F(\mathbf{u}_{k+1}) - F(\mathbf{u}_k), \widetilde{F}(\mathbf{v}_k)\right\rangle \quad \text{(D.7)}$$

$$+ \frac{A_k\theta L^2}{M\eta_k^2} \|\mathbf{u}_{k+1} - \mathbf{v}_k\|^2 - A_k L^2 \|\mathbf{u}_k - \mathbf{v}_{k-1}\|^2$$

$$- \frac{2A_k}{M\eta_k^2} \left\|F(\mathbf{v}_k) - \widetilde{F}(\mathbf{v}_k)\right\|^2 - A_k \left\|F(\mathbf{v}_{k-1}) - \widetilde{F}(\mathbf{v}_{k-1})\right\|^2.$$

To bound $\mathcal{T}_{[2]}$, notice that $F$ is monotone, so we have

$$\langle F(\mathbf{u}_{k+1}), \mathbf{u}_{k+1} - \mathbf{u}_k\rangle \geq \langle F(\mathbf{u}_k), \mathbf{u}_{k+1} - \mathbf{u}_k\rangle.$$

Using the first line in Eq. (D.4) for the RHS and the second line for the LHS, we can obtain

$$\frac{\lambda_k}{1 - \lambda_k} \langle F(\mathbf{u}_{k+1}), \mathbf{u}_0 - \mathbf{u}_{k+1}\rangle \geq \lambda_k \langle F(\mathbf{u}_k), \mathbf{u}_0 - \mathbf{u}_k\rangle - \eta_k \left\langle F(\mathbf{u}_k), \widetilde{F}(\mathbf{v}_k)\right\rangle$$

$$+ \frac{\eta_k}{1 - \lambda_k} \left\langle F(\mathbf{u}_{k+1}), \widetilde{F}(\mathbf{v}_k)\right\rangle.$$

Multiplying both sides by $\frac{B_k}{\lambda_k}$ and using that $B_{k+1} = \frac{B_k}{1-\lambda_k}$ by Assumption 5, we have

$$\mathcal{T}_{[2]} \geq \frac{B_k\eta_k}{\lambda_k(1 - \lambda_k)} \left\langle F(\mathbf{u}_{k+1}), \widetilde{F}(\mathbf{v}_k)\right\rangle - \frac{B_k\eta_k}{\lambda_k} \left\langle F(\mathbf{u}_k), \widetilde{F}(\mathbf{v}_k)\right\rangle$$

$$= B_{k+1}\eta_k \left\langle F(\mathbf{u}_{k+1}), \widetilde{F}(\mathbf{v}_k)\right\rangle + \frac{B_k\eta_k}{\lambda_k} \left\langle F(\mathbf{u}_{k+1}) - F(\mathbf{u}_k), \widetilde{F}(\mathbf{v}_k)\right\rangle. \quad \text{(D.8)}$$

Combining Inequalities (D.7) and (D.8) and plugging the bounds into Eq. (D.5), we obtain

$$\mathcal{V}_k - \mathcal{V}_{k+1} \geq \left(\frac{A_k}{M\eta_k^2} - A_{k+1}\right) \|F(\mathbf{u}_{k+1})\|^2 + \frac{A_k(1 - M\eta_k^2)}{M\eta_k^2} \left\|\widetilde{F}(\mathbf{v}_k)\right\|^2$$

$$- 2\left(\frac{A_k(1 - M\eta_k^2)}{M\eta_k^2} - \frac{B_{k+1}\eta_k}{2}\right) \left\langle F(\mathbf{u}_{k+1}), \widetilde{F}(\mathbf{v}_k)\right\rangle$$

$$+ \left(\frac{B_k\eta_k}{\lambda_k} - 2A_k\right) \left\langle F(\mathbf{u}_{k+1}) - F(\mathbf{u}_k), \widetilde{F}(\mathbf{v}_k)\right\rangle$$

$$+ L^2 \left(\frac{A_k\theta}{M\eta_k^2} - c_{k+1}\right) \|\mathbf{u}_{k+1} - \mathbf{v}_k\|^2 + L^2(c_k - A_k) \|\mathbf{u}_k - \mathbf{v}_{k-1}\|^2$$

$$- \frac{2A_k}{M\eta_k^2} \left\|F(\mathbf{v}_k) - \widetilde{F}(\mathbf{v}_k)\right\|^2 - A_k \left\|F(\mathbf{v}_{k-1}) - \widetilde{F}(\mathbf{v}_{k-1})\right\|^2.$$

By Assumption 5, we choose $A_k = \frac{B_k \eta_k}{2\lambda_k}$. Define:

$$
\begin{cases}
S_k^{11} := \dfrac{A_k}{M\eta_k^2} - A_{k+1} = \dfrac{B_k}{2M\lambda_k\eta_k} - \dfrac{B_k\eta_{k+1}}{2\left(1-\lambda_k\right)\lambda_{k+1}} \\[2mm]
S_k^{22} := \dfrac{A_k\left(1-M\eta_k^2\right)}{M\eta_k^2} = \dfrac{B_k\left(1-M\eta_k^2\right)}{2M\eta_k\lambda_k} \\[2mm]
S_k^{12} := \dfrac{A_k\left(1-M\eta_k^2\right)}{M\eta_k^2} - \dfrac{B_{k+1}\eta_k}{2} = \dfrac{\left(1-\lambda_k-M\eta_k^2\right)B_k}{2M\left(1-\lambda_k\right)\lambda_k\eta_k}.
\end{cases}
$$

Then, we obtain

$$
\mathcal{V}_k - \mathcal{V}_{k+1} \geq S_k^{11} \left\| F\left(\mathbf{u}_{k+1}\right)\right\|^2 + S_k^{22} \left\| \widetilde{F}\left(\mathbf{v}_k\right)\right\|^2 - 2S_k^{12} \left\langle F\left(\mathbf{u}_{k+1}\right), \widetilde{F}\left(\mathbf{v}_k\right)\right\rangle
$$
$$
+ L^2\left(\frac{A_k\theta}{M\eta_k^2} - c_{k+1}\right)\left\|\mathbf{u}_{k+1} - \mathbf{v}_k\right\|^2 + L^2\left(c_k - A_k\right)\left\|\mathbf{u}_k - \mathbf{v}_{k-1}\right\|^2
$$
$$
- \frac{2A_k}{M\eta_k^2}\left\| F(\mathbf{v}_k) - \widetilde{F}(\mathbf{v}_k)\right\|^2 - A_k \left\| F(\mathbf{v}_{k-1}) - \widetilde{F}(\mathbf{v}_{k-1})\right\|^2.
$$

Suppose that $S_k^{11} \geq 0$, $S_k^{22} \geq 0$ and $\sqrt{S_k^{11} S_k^{22}} = S_k^{12}$. Then, we can conclude

$$
\mathcal{V}_k - \mathcal{V}_{k+1} = \left\| \sqrt{S_k^{11}} F\left(\mathbf{u}_{k+1}\right) - \sqrt{S_k^{22}}\widetilde{F}\left(\mathbf{v}_k\right)\right\|^2
$$
$$
+ L^2\left(\frac{A_k\theta}{M\eta_k^2} - c_{k+1}\right)\left\|\mathbf{u}_{k+1} - \mathbf{v}_k\right\|^2 + L^2\left(c_k - A_k\right)\left\|\mathbf{u}_k - \mathbf{v}_{k-1}\right\|^2
$$
$$
- \frac{2A_k}{M\eta_k^2}\left\| F(\mathbf{v}_k) - \widetilde{F}(\mathbf{v}_k)\right\|^2 - A_k \left\| F(\mathbf{v}_{k-1}) - \widetilde{F}(\mathbf{v}_{k-1})\right\|^2
$$
$$
\geq L^2\left(\frac{A_k\theta}{M\eta_k^2} - c_{k+1}\right)\left\|\mathbf{u}_{k+1} - \mathbf{v}_k\right\|^2 + L^2\left(c_k - A_k\right)\left\|\mathbf{u}_k - \mathbf{v}_{k-1}\right\|^2
$$
$$
- \frac{2A_k}{M\eta_k^2}\left\| F(\mathbf{v}_k) - \widetilde{F}(\mathbf{v}_k)\right\|^2 - A_k \left\| F(\mathbf{v}_{k-1}) - \widetilde{F}(\mathbf{v}_{k-1})\right\|^2.
$$

To complete the proof, let us argue that the assumptions that $S_k^{11} \geq 0$, $S_k^{22} \geq 0$ and $\sqrt{S_k^{11} S_k^{22}} = S_k^{12}$ we made above are valid. First, notice that $S_k^{11} \geq 0$ is equivalent to $\eta_{k+1} \leq \frac{\lambda_{k+1}(1-\lambda_k)}{M\lambda_k\eta_k}$, and $S_k^{22} \geq 0$ is equivalent to $M\eta_k^2 \leq 1$, which are both included in Assumption 5. Moreover, since $B_k > 0$, $\sqrt{S_k^{11} S_k^{22}} = S_k^{12}$ is equivalent to

$$
\frac{\left(1-M\eta_k^2\right)}{M\eta_k} \cdot \left(\frac{1}{M\eta_k} - \frac{\lambda_k\eta_{k+1}}{(1-\lambda_k)\lambda_{k+1}}\right) = \left(\frac{1-\lambda_k-M\eta_k^2}{M(1-\lambda_k)\eta_k}\right)^2,
$$

which is further equivalent to $\eta_{k+1} = \frac{\lambda_{k+1}\left(1-M\eta_k^2-\lambda_k^2\right)}{\lambda_k(1-\lambda_k)\left(1-M\eta_k^2\right)} \cdot \eta_k$, provided that $M\eta_k^2 + \lambda_k^2 \leq 1$. Both these inequalities hold by Assumption 5, thus completing the proof. $\qquad\square$

Motivated by Assumption 5 and Lemma D.1, we make the choice of $\lambda_k$ and $\eta_k$ as

$$
\lambda_k := \frac{1}{k+2} \quad \text{and} \quad \eta_{k+1} := \frac{\left(1-\lambda_k^2-M\eta_k^2\right)\lambda_{k+1}\eta_k}{\left(1-M\eta_k^2\right)(1-\lambda_k)\lambda_k}, \tag{D.9}
$$

where $M = 3L^2(2+\theta)$ and $0 < \eta_0 < \frac{1}{\sqrt{2M}}$. The sequence $\{\eta_k\}_{k\geq 1}$ given by Eq. (D.9) is actually non-increasing and has a positive limit. We summarize this result in the following lemma for completeness, and the proof can be found in [51].

**Lemma D.2.** *Given $M > 0$, the sequence $\{\eta_k\}$ generated by Eq. (D.9) is non-increasing, i.e. $\eta_{k+1} \leq \eta_k \leq \eta_0 < \frac{\sqrt{3}}{2\sqrt{M}}$. Moreover, if $0 < \eta_0 < \frac{1}{\sqrt{2M}}$, we have that $\eta_* := \lim_{k\to\infty}\eta_k$ exists and*

$$
\eta_* \geq \underline{\eta} := \frac{\eta_0\left(1-2M\eta_0^2\right)}{1-M\eta_0^2} > 0. \tag{D.10}
$$

We now prove the results for the iteration complexity and the corresponding oracle complexity for Algorithm 2.

**Theorem 4.1.** *Given an arbitrary initial point $\mathbf{u}_0 \in \mathbb{R}^d$ and target error $\epsilon > 0$, assume that the iterates $\mathbf{u}_k$ evolve according to Algorithm 2 for $k \geq 1$. Then, for all $k \geq 2$,*

$$\mathbb{E}\left[\|F(\mathbf{u}_k)\|^2 + 2L^2 \|\mathbf{u}_k - \mathbf{v}_{k-1}\|^2\right] \leq \frac{\Lambda_0}{(k+1)(k+2)} + \Lambda_1 \epsilon^2, \tag{4.2}$$

*where $\Lambda_0 = \frac{4(L^2 \eta_0 \underline{\eta}+1)\|\mathbf{u}_0 - \mathbf{u}^*\|^2}{\underline{\eta}^2}$ and $\Lambda_1 = \frac{5(1+M\eta\eta_0)}{M\underline{\eta}^2}$. In particular, $\mathbb{E}\left[\|F(\mathbf{u}_N)\|^2 + 2L^2 \|\mathbf{u}_N - \mathbf{v}_{N-1}\|^2\right] \leq 2\Lambda_1 \epsilon^2 = \mathcal{O}(\epsilon^2)$ after at most $N = \lceil \frac{\sqrt{\Lambda_0}}{\sqrt{\Lambda_1}\epsilon} \rceil = \mathcal{O}(\frac{L\|\mathbf{u}_0 - \mathbf{u}^*\|}{\epsilon})$ iterations. The total number of oracle queries to $\widehat{F}$ is $\mathcal{O}(\frac{\sigma^2 L\|\mathbf{u}_0 - \mathbf{u}^*\| + L^3\|\mathbf{u}_0 - \mathbf{u}^*\|^3}{\epsilon^3})$ in expectation.*

*Proof.* We start with verifying that the conditions in Eq. (D.2) of Lemma D.1 are all satisfied. By Eq. (D.9) and Lemma D.2, we know that $\{\eta_k\}$ is non-increasing and $\eta_* = \lim_{k\to\infty} \eta_k > 0$, so the first condition in Eq. (D.2) is satisfied. Also, as $0 < \eta_k \leq \eta_0 \leq \frac{1}{\sqrt{2M}}$, we have $M\eta_k^2 \leq M\eta_0^2 \leq \frac{1}{2} < 1 - \frac{1}{(k+2)^2}$. So the second condition in Eq. (D.2) holds. Moreover, since $\eta_{k+1} \leq \eta_k$, the third condition holds if $\eta_k^2 \leq \frac{\lambda_{k+1}(1-\lambda_k)}{M\lambda_k} = \frac{k+1}{M(k+3)}$. Due to the fact that $\frac{k+1}{M(k+3)} \geq \frac{1}{3M}$ and $\eta_k \leq \eta_0$ for all $k \geq 1$, we can have this condition hold if $\eta_0 \leq \frac{1}{\sqrt{3M}}$. Hence all the conditions hold with our parameter update and letting $\eta_0 \leq \frac{1}{\sqrt{3M}}$.

Let $c_k = A_k$, then we obtain

$$L^2\left(\frac{A_k\theta}{M\eta_k^2} - c_{k+1}\right)\|\mathbf{u}_{k+1} - \mathbf{v}_k\|^2 + L^2(c_k - A_k)\|\mathbf{u}_k - \mathbf{v}_{k-1}\|^2$$

$$= L^2\left(\frac{A_k\theta}{M\eta_k^2} - A_{k+1}\right)\|\mathbf{u}_{k+1} - \mathbf{v}_k\|^2$$

$$= \frac{L^2 B_k}{2}\left(\frac{\theta}{M\lambda_k\eta_k} - \frac{\eta_{k+1}}{\lambda_{k+1}(1-\lambda_k)}\right)\|\mathbf{u}_{k+1} - \mathbf{v}_k\|^2.$$

Here we choose the parameters such that $\eta_k\eta_{k+1} \leq \frac{\theta\lambda_{k+1}(1-\lambda_k)}{M\lambda_k} = \frac{\theta(k+1)}{M(k+3)}$ to ensure $\left(\frac{\theta}{M\lambda_k\eta_k} - \frac{\eta_{k+1}}{\lambda_{k+1}(1-\lambda_k)}\right) \geq 0$. Since $\eta_{k+1} \leq \eta_k$, the required inequality holds if $\eta_k \leq \sqrt{\frac{\theta(k+1)}{M(k+3)}}$, which is satisfied if we let $\eta_0 \leq \sqrt{\frac{\theta}{3M}}$ as $\eta_k \leq \eta_0$ for all $k \geq 1$.

Combining the two conditions on $\eta_0$, and choosing $\theta = 1$, we have

$$\eta_0 \leq \frac{1}{\sqrt{3M}} = \frac{1}{3L\sqrt{2+\theta}} = \frac{1}{3L\sqrt{3}},$$

which is required by Algorithm 2 and thus satisfied.

Hence, with $0 < \eta_0 \leq \frac{1}{3L\sqrt{3}}$, we have

$$\mathcal{V}_k - \mathcal{V}_{k+1} \geq L^2\left(\frac{A_k\theta}{M\eta_k^2} - c_{k+1}\right)\|\mathbf{u}_{k+1} - \mathbf{v}_k\|^2 + L^2(c_k - A_k)\|\mathbf{u}_k - \mathbf{v}_{k-1}\|^2$$

$$\quad - \frac{2A_k}{M\eta_k^2}\left\|F(\mathbf{v}_k) - \widetilde{F}(\mathbf{v}_k)\right\|^2 - A_k\left\|F(\mathbf{v}_{k-1}) - \widetilde{F}(\mathbf{v}_{k-1})\right\|^2 \tag{D.11}$$

$$\geq -\frac{2A_k}{M\eta_k^2}\left\|F(\mathbf{v}_k) - \widetilde{F}(\mathbf{v}_k)\right\|^2 - A_k\left\|F(\mathbf{v}_{k-1}) - \widetilde{F}(\mathbf{v}_{k-1})\right\|^2.$$

Consider $\mathcal{C}_k = A_k\|F(\mathbf{u}_k)\|^2 + B_k\langle F(\mathbf{u}_k), \mathbf{u}_k - \mathbf{u}_0\rangle$. Then:

$$\mathcal{C}_k \overset{(i)}{\geq} A_k\|F(\mathbf{u}_k)\|^2 + B_k\langle F(\mathbf{u}_k) - F(\mathbf{u}^*), \mathbf{u}_k - \mathbf{u}^*\rangle + B_k\langle F(\mathbf{u}_k), \mathbf{u}^* - \mathbf{u}_0\rangle$$

$$\overset{(ii)}{\geq} A_k\|F(\mathbf{u}_k)\|^2 - \frac{A_k}{2}\|F(\mathbf{u}_k)\|^2 - \frac{B_k^2}{2A_k}\|\mathbf{u}_0 - \mathbf{u}^*\|^2$$

$$= \frac{A_k}{2}\|F(\mathbf{u}_k)\|^2 - \frac{B_k^2}{2A_k}\|\mathbf{u}_0 - \mathbf{u}^*\|^2,$$

where $(i)$ is due to $\mathbf{u}^*$ being the solution to the monotone inclusion problem so an (SVI) solution as well, and we use monotonicity and Young's Inequality for $(ii)$. So we obtain

$$\frac{A_k}{2} \|F(\mathbf{u}_k)\|^2 + A_k L^2 \|\mathbf{u}_k - \mathbf{v}_{k-1}\|^2 \leq \mathcal{C}_k + A_k L^2 \|\mathbf{u}_k - \mathbf{v}_{k-1}\|^2 + \frac{B_k^2}{2A_k} \|\mathbf{u}_0 - \mathbf{u}^*\|^2$$
$$= \mathcal{V}_k + \frac{B_k^2}{2A_k} \|\mathbf{u}_0 - \mathbf{u}^*\|^2.$$

Since $B_{k+1} = \frac{B_k}{1-\lambda_k}$ and $\lambda_k = \frac{1}{k+2}$, we have $B_k = (k+1)B_0$ for any $B_0 > 0$. Then we obtain $A_k = c_k = \frac{B_k \eta_k}{2\lambda_k} = \frac{B_0(k+1)(k+2)\eta_k}{2}$. By Lemma D.2, we know that $0 < \underline{\eta} \leq \eta_* \leq \eta_k \leq \eta_0$, so $\frac{B_0(k+1)(k+2)\underline{\eta}}{2} \leq A_k = c_k \leq \frac{B_0(k+1)(k+2)\eta_0}{2}$. By Inequality (D.11) and noticing $\mathbf{v}_{-1} = \mathbf{u}_0$, we have

$$\frac{B_0 \underline{\eta}(k+1)(k+2)}{4} \|F(\mathbf{u}_k)\|^2 + \frac{B_0 \underline{\eta} L^2 (k+1)(k+2)}{2} \|\mathbf{u}_k - \mathbf{v}_{k-1}\|^2$$
$$\leq \frac{A_k}{2} \|F(\mathbf{u}_k)\|^2 + A_k L^2 \|\mathbf{u}_k - \mathbf{v}_{k-1}\|^2$$
$$\leq \mathcal{V}_k + \frac{B_k^2}{2A_k} \|\mathbf{u}_0 - \mathbf{u}^*\|^2$$
$$\leq \mathcal{V}_{k-1} + \frac{B_k^2}{2A_k} \|\mathbf{u}_0 - \mathbf{u}^*\|^2 + \frac{2A_{k-1}}{M\eta_{k-1}^2} \left\| F(\mathbf{v}_{k-1}) - \widetilde{F}(\mathbf{v}_{k-1}) \right\|^2 + A_{k-1} \left\| F(\mathbf{v}_{k-2}) - \widetilde{F}(\mathbf{v}_{k-2}) \right\|^2.$$

Unrolling this recursive bound down to $\mathcal{V}_0$, we obtain

$$\frac{B_0 \underline{\eta}(k+1)(k+2)}{4} \|F(\mathbf{u}_k)\|^2 + \frac{B_0 \underline{\eta} L^2 (k+1)(k+2)}{2} \|\mathbf{u}_k - \mathbf{v}_{k-1}\|^2$$
$$\leq \mathcal{V}_0 + \frac{B_k^2}{2A_k} \|\mathbf{u}_0 - \mathbf{u}^*\|^2 + \sum_{i=0}^{k-1} \frac{2A_i}{M\eta_i^2} \left\| F(\mathbf{v}_i) - \widetilde{F}(\mathbf{v}_i) \right\|^2 + \sum_{i=0}^{k-1} A_i \left\| F(\mathbf{v}_{i-1}) - \widetilde{F}(\mathbf{v}_{i-1}) \right\|^2$$
$$\overset{(i)}{\leq} B_0 \eta_0 \|F(\mathbf{u}_0)\|^2 + \frac{B_0(k+1)^2}{\underline{\eta}(k+1)(k+2)} \|\mathbf{u}_0 - \mathbf{u}^*\|^2$$
$$+ \sum_{i=0}^{k-1} \frac{B_0(i+1)(i+2)}{M\underline{\eta}} \left\| F(\mathbf{v}_i) - \widetilde{F}(\mathbf{v}_i) \right\|^2$$
$$+ \sum_{i=0}^{k-1} \frac{B_0(i+1)(i+2)\eta_0}{2} \left\| F(\mathbf{v}_{i-1}) - \widetilde{F}(\mathbf{v}_{i-1}) \right\|^2,$$

where we plug in the bound for $A_i$ and $\eta_i$ in $(i)$. Taking expectation with respect to all randomness on both sides and using the variance bound from Corollary 2.2, we obtain that

$$
\mathbb{E}\left[\|F(\mathbf{u}_k)\|^2 + 2L^2 \|\mathbf{u}_k - \mathbf{v}_{k-1}\|^2\right]
$$

$$
\leq \frac{4}{B_0 \underline{\eta}(k+1)(k+2)}\left[B_0\eta_0 \|F(\mathbf{u}_0)\|^2 + \frac{B_0(k+1)^2}{\underline{\eta}(k+1)(k+2)} \|\mathbf{u}_0 - \mathbf{u}^*\|^2\right.
$$

$$
+ \sum_{i=0}^{k-1} \frac{B_0(i+1)(i+2)}{M\underline{\eta}}\mathbb{E}\left[\left\|F(\mathbf{v}_i) - \widetilde{F}(\mathbf{v}_i)\right\|^2\right]
$$

$$
\left.+ \sum_{i=0}^{k-1} \frac{B_0(i+1)(i+2)\eta_0}{2}\mathbb{E}\left[\left\|F(\mathbf{v}_{i-1}) - \widetilde{F}(\mathbf{v}_{i-1})\right\|^2\right]\right]
$$

$$
\overset{(i)}{\leq} \frac{4}{\underline{\eta}(k+1)(k+2)}\left[\left(L^2\eta_0 + \frac{1}{\underline{\eta}}\right)\|\mathbf{u}_0 - \mathbf{u}^*\|^2 + \left(\frac{2}{M\underline{\eta}} + \eta_0 + 3\eta_0\right)\frac{\epsilon^2}{8}\right.
$$

$$
\left.+ \sum_{i=1}^{k-1} \frac{(i+1)(i+2)}{M\underline{\eta}}\frac{\epsilon^2}{i} + \sum_{i=2}^{k-1} \frac{(i+1)(i+2)\eta_0}{2}\frac{\epsilon^2}{i-1}\right]
$$

$$
\overset{(ii)}{\leq} \frac{4\left(L^2\eta_0\underline{\eta} + 1\right)\|\mathbf{u}_0 - \mathbf{u}^*\|^2}{\underline{\eta}^2(k+1)(k+2)} + \frac{1 + 2M\underline{\eta}\eta_0}{M\underline{\eta}(k+1)(k+2)}\epsilon^2
$$

$$
+ \frac{4(k-1)(k+4)}{M\underline{\eta}^2(k+1)(k+2)}\epsilon^2 + \frac{4\eta_0(k-2)(k+3)}{\underline{\eta}(k+1)(k+2)}\epsilon^2
$$

$$
\overset{(iii)}{\leq} \frac{4\left(L^2\eta_0\underline{\eta} + 1\right)\|\mathbf{u}_0 - \mathbf{u}^*\|^2}{\underline{\eta}^2(k+1)(k+2)} + \frac{\left(1 + 2M\underline{\eta}\eta_0\right)}{M\underline{\eta}^2(k+1)(k+2)}\epsilon^2 + \frac{4\left(1 + M\eta_0\underline{\eta}\right)}{M\underline{\eta}^2}\epsilon^2,
$$

where we use Lipschitz property and variance bounds by variance reduction for $(i)$. For $(ii)$, we use the fact that $\frac{i+1}{i} \leq 2$ and $\frac{i+2}{i-1} \leq 4$ and sum over $2(i+1)$, respectively. Moreover, $(iii)$ is due to $\frac{(k-1)(k+4)}{(k+1)(k+3)} \leq 1$ and $\frac{(k-2)(k+3)}{(k+1)(k+2)} \leq 1$ and by combining the last two terms.

When $k \geq \frac{\sqrt{\Lambda_0}}{\sqrt{\Lambda_1}\epsilon} = \mathcal{O}\left(\frac{L\|\mathbf{u}_0 - \mathbf{u}^*\|}{\epsilon}\right)$, where $\Lambda_0 = \frac{4\left(L^2\eta_0\underline{\eta}+1\right)\|\mathbf{u}_0 - \mathbf{u}^*\|^2}{\underline{\eta}^2}$ and $\Lambda_1 = \frac{5\left(1+M\underline{\eta}\eta_0\right)}{M\underline{\eta}^2}$, we have

$$
\mathbb{E}\left[\|F(\mathbf{u}_k)\|^2 + 2L^2 \|\mathbf{u}_k - \mathbf{v}_{k-1}\|^2\right] \leq \frac{\Lambda_0}{(k+1)(k+2)} + \Lambda_1\epsilon^2 \leq 2\Lambda_1\epsilon^2.
$$

Claimed stochastic oracle complexity follows from Lemma D.3 below. $\qquad\square$

**Lemma D.3.** *Let $\mathbf{u}_0 \in \mathbb{R}^d$ be an arbitrary initial point and assume that iterates $\mathbf{u}_k$ evolve according to Algorithm 2. Then, Algorithm 2 returns a point $\mathbf{u}_N$ such that $\mathbb{E}\left[\|F(\mathbf{u}_N)\|^2\right] \leq 2\Lambda_1\epsilon^2$ after at most $\mathcal{O}\left(\frac{\sigma^2 L\|\mathbf{u}_0 - \mathbf{u}^*\| + L^3\|\mathbf{u}_0 - \mathbf{u}^*\|^3}{\epsilon^3}\right)$ stochastic queries to $F$.*

*Proof.* Let $m_k$ be the number of stochastic queries made by the variance reduction method at iteration $k$ for $k \geq 1$. Conditional on $\mathcal{F}_{k-1}$, we have

$$
\mathbb{E}\left[m_{k+1}|\mathcal{F}_{k-1}\right] = \mathbb{E}\left[p_k S_1^{(k)} + 2(1 - p_k)S_2^{(k)}\Big|\mathcal{F}_{k-1}\right]
$$

$$
= p_k\left\lceil\frac{8\sigma^2}{p_k\epsilon^2}\right\rceil + 2(1 - p_k)\left\lceil\frac{8L^2 \|\mathbf{v}_k - \mathbf{v}_{k-1}\|^2}{p_k^2\epsilon^2}\right\rceil
$$

$$
\overset{(i)}{\leq} p_k\left(\frac{8\sigma^2}{p_k\epsilon^2} + 1\right) + 2(1 - p_k)\left(\frac{8L^2 \|\mathbf{v}_k - \mathbf{v}_{k-1}\|^2}{p_k^2\epsilon^2} + 1\right),
$$

where $(i)$ is due to the fact that $\lceil x\rceil \leq x + 1$ for any $x \in \mathbb{R}$. Taking expectation with respect to all randomness on both sides, and rearranging the terms, we obtain

$$
\mathbb{E}[m_{k+1}] \leq \frac{8\sigma^2}{\epsilon^2} + \frac{16(1 - p_k)L^2\mathbb{E}\left[\|\mathbf{v}_k - \mathbf{v}_{k-1}\|^2\right]}{p_k^2\epsilon^2} + 2.
$$

With $m_0 = m_1 = S_1^{(0)} = \lceil \frac{8\sigma^2}{\epsilon^2} \rceil$, let $M$ be the total number of stochastic queries up to iteration $N$ such that $\mathbb{E}[\|F(\mathbf{u}_k)\|] \leq 2\Lambda_1\epsilon$ for all $k \geq N$, we have

$$\mathbb{E}[M] = \mathbb{E}\Big[\sum_{k=0}^{N} m_k\Big] = 2\Big\lceil \frac{8\sigma^2}{\epsilon^2}\Big\rceil + \mathbb{E}\Big[\sum_{k=2}^{N} m_k\Big]$$

$$\leq \frac{16\sigma^2}{\epsilon^2} + 2 + \sum_{k=1}^{N}\Big(\frac{8\sigma^2}{\epsilon^2} + \frac{16(1-p_k)L^2\mathbb{E}\big[\|\mathbf{v}_k - \mathbf{v}_{k-1}\|^2\big]}{p_k^2\epsilon^2} + 2\Big)$$

$$\overset{(i)}{\leq} \frac{8\sigma^2\Delta}{\epsilon^3} + \frac{16\sigma^2}{\epsilon^2} + \frac{2\Delta}{\epsilon} + \frac{16L^2}{\epsilon^2}\sum_{k=1}^{N}\frac{(1-p_k)\mathbb{E}\big[\|\mathbf{v}_k - \mathbf{v}_{k-1}\|^2\big]}{p_k^2}, \quad \text{(D.12)}$$

where $(i)$ follows from $N \leq \frac{\Delta}{\epsilon}$ with $\Delta = \sqrt{\frac{\Lambda_0}{\Lambda_1}} + \epsilon$, and $1 - p_k \leq 1$.

Then we come to bound $\mathbb{E}\big[\|\mathbf{v}_k - \mathbf{v}_{k-1}\|^2\big]$. Notice that for $k \geq 1$

$$\mathbf{v}_k - \mathbf{v}_{k-1} = \mathbf{v}_k - \mathbf{u}_{k+1} + \mathbf{u}_{k+1} - \mathbf{u}_k + \mathbf{u}_k - \mathbf{v}_{k-1}$$

$$\overset{(i)}{=} \eta_k\big(\widetilde{F}(\mathbf{v}_k) - \widetilde{F}(\mathbf{v}_{k-1})\big) - \eta_{k-1}\big(\widetilde{F}(\mathbf{v}_{k-1}) - \widetilde{F}(\mathbf{v}_{k-2})\big) + \mathbf{u}_{k+1} - \mathbf{u}_k$$

$$= \eta_k\widetilde{F}(\mathbf{v}_k) - \big(\eta_k + \eta_{k-1}\big)\widetilde{F}(\mathbf{v}_{k-1}) + \eta_{k-1}\widetilde{F}(\mathbf{v}_{k-2}) + \mathbf{u}_{k+1} - \mathbf{u}_k,$$

where $(i)$ is based on the third line in Eq. (D.4). To estimate $\mathbf{u}_{k+1} - \mathbf{u}_k$, we recursively use the first line in Eq. (D.4), and obtain for $k \geq 2$,

$$\mathbf{u}_{k+1} - \mathbf{u}_k = \lambda_k(\mathbf{u}_0 - \mathbf{u}_k) - \eta_k\widetilde{F}(\mathbf{v}_k)$$

$$= \lambda_k(1 - \lambda_{k-1})(\mathbf{u}_0 - \mathbf{u}_{k-1}) + \lambda_k\eta_{k-1}\widetilde{F}(\mathbf{v}_{k-1}) - \eta_k\widetilde{F}(\mathbf{v}_k)$$

$$= \lambda_k\sum_{i=0}^{k-2}\Big(\prod_{j=i+1}^{k-1}(1-\lambda_j)\Big)\eta_i\widetilde{F}(\mathbf{v}_i) + \lambda_k\eta_{k-1}\widetilde{F}(\mathbf{v}_{k-1}) - \eta_k\widetilde{F}(\mathbf{v}_k)$$

$$= \sum_{i=0}^{k-2}\frac{i+2}{(k+2)(k+1)}\eta_i\widetilde{F}(\mathbf{v}_i) + \frac{\eta_{k-1}}{k+2}\widetilde{F}(\mathbf{v}_{k-1}) - \eta_k\widetilde{F}(\mathbf{v}_k).$$

So we have

$$\mathbf{u}_{k+1} - \mathbf{u}_k = \begin{cases} -\eta_0\widetilde{F}(\mathbf{v}_0) & \text{if } k = 0, \\[2mm] \lambda_1\eta_0\widetilde{F}(\mathbf{v}_0) - \eta_1\widetilde{F}(\mathbf{v}_1) & \text{if } k = 1, \\[2mm] \sum_{i=0}^{k-2}\frac{i+2}{(k+2)(k+1)}\eta_i\widetilde{F}(\mathbf{v}_i) + \frac{\eta_{k-1}}{k+2}\widetilde{F}(\mathbf{v}_{k-1}) - \eta_k\widetilde{F}(\mathbf{v}_k) & \text{if } k \geq 2. \end{cases} \quad \text{(D.13)}$$

Then we obtain for $k \geq 3$

$$\|\mathbf{v}_k - \mathbf{v}_{k-1}\|^2$$

$$= \left\|\Big(\frac{\eta_{k-1}}{k+2} - \eta_k - \eta_{k-1}\Big)\widetilde{F}(\mathbf{v}_{k-1}) + \eta_{k-1}\widetilde{F}(\mathbf{v}_{k-2}) + \sum_{i=0}^{k-2}\frac{i+2}{(k+2)(k+1)}\eta_i\widetilde{F}(\mathbf{v}_i)\right\|^2$$

$$\leq 3\Big(\frac{\eta_{k-1}}{k+2} - \eta_k - \eta_{k-1}\Big)^2\left\|\widetilde{F}(\mathbf{v}_{k-1})\right\|^2 + 3\Big(\eta_{k-1} + \frac{k\eta_{k-2}}{(k+1)(k+2)}\Big)^2\left\|\widetilde{F}(\mathbf{v}_{k-2})\right\|^2$$

$$+ \sum_{i=0}^{k-3}\frac{3(i+2)^2(k-2)}{k^2(k+1)^2}\eta_i^2\left\|\widetilde{F}(\mathbf{v}_i)\right\|^2$$

$$\leq \eta_0^2\Big(12\left\|\widetilde{F}(\mathbf{v}_{k-1})\right\|^2 + 5\left\|\widetilde{F}(\mathbf{v}_{k-2})\right\|^2\Big) + \sum_{i=0}^{k-3}\frac{3(i+2)^2}{k(k+1)^2}\eta_0^2\left\|\widetilde{F}(\mathbf{v}_i)\right\|^2.$$

Taking expectation with respect to all randomness on both sides, we have

$$\mathbb{E}\|\mathbf{v}_k - \mathbf{v}_{k-1}\|^2 \leq \eta_0^2\Big(12\mathbb{E}\left\|\widetilde{F}(\mathbf{v}_{k-1})\right\|^2 + 5\mathbb{E}\left\|\widetilde{F}(\mathbf{v}_{k-2})\right\|^2\Big) + \sum_{i=0}^{k-3}\frac{3(i+2)^2}{k(k+1)^2}\eta_0^2\mathbb{E}\left\|\widetilde{F}(\mathbf{v}_i)\right\|^2.$$

Note that for $k \geq 1$, we have

$$\mathbb{E} \left\| \widetilde{F}(\mathbf{v}_k) \right\|^2 \leq 2\mathbb{E} \left\| F(\mathbf{u}_{k+1}) \right\|^2 + 4\mathbb{E} \left\| F(\mathbf{u}_{k+1}) - F(\mathbf{v}_k) \right\|^2 + 4\mathbb{E} \left\| \widetilde{F}(\mathbf{v}_k) - F(\mathbf{v}_k) \right\|^2$$

$$\overset{(i)}{\leq} 2\mathbb{E} \left( \left\| F(\mathbf{u}_{k+1}) \right\|^2 + 2L^2 \left\| \mathbf{u}_{k+1} - \mathbf{v}_k \right\|^2 \right) + 4\frac{\epsilon^2}{k}$$

$$\overset{(ii)}{\leq} 2 \left( \frac{\Lambda_0}{(k+1)^2} + \Lambda_1 \epsilon^2 \right) + 4\frac{\epsilon^2}{k},$$

where $(i)$ is due to the Lipschitz property and the variance bound, and we use the result of Theorem 4.1 for $(ii)$. Proceeding similarly, we have $\mathbb{E} \left\| \widetilde{F}(\mathbf{v}_0) \right\|^2 \leq 2 \left( \Lambda_0 + \Lambda_1 \epsilon^2 \right) + \frac{\epsilon^2}{2}$, so we obtain for $k \geq 4$,

$$\mathbb{E} \left\| \mathbf{v}_k - \mathbf{v}_{k-1} \right\|^2$$

$$\leq \eta_0^2 \left( 12 \left\| \widetilde{F}(\mathbf{v}_{k-1}) \right\|^2 + 5 \left\| \widetilde{F}(\mathbf{v}_{k-2}) \right\|^2 \right) + \sum_{i=0}^{k-3} \frac{3(i+2)^2}{k(k+1)^2} \eta_0^2 \left\| \widetilde{F}(\mathbf{v}_i) \right\|^2$$

$$\leq \eta_0^2 \left[ 24 \left( \frac{\Lambda_0}{k^2} + \Lambda_1 \epsilon^2 + 2\frac{\epsilon^2}{k-1} \right) + 10 \left( \frac{\Lambda_0}{(k-1)^2} + \Lambda_1 \epsilon^2 + 2\frac{\epsilon^2}{k-2} \right) \right]$$

$$+ \frac{12\eta_0^2}{k(k+1)^2} \left[ 2 \left( \Lambda_0 + \Lambda_1 \epsilon^2 \right) + \frac{\epsilon^2}{2} \right] + \sum_{i=1}^{k-3} \frac{6(i+2)^2}{k(k+1)^2} \eta_0^2 \left( \frac{\Lambda_0}{(i+1)^2} + \Lambda_1 \epsilon^2 + 2\frac{\epsilon^2}{i} \right)$$

$$\leq 40\eta_0^2 \frac{\Lambda_0}{(k-1)^2} + 35\eta_0^2 \left( \Lambda_1 + 1 \right) \epsilon^2 + 6\eta_0^2 \sum_{i=1}^{k-3} \left( \frac{4\Lambda_0}{k(k+1)^2} + \frac{\Lambda_1 \epsilon^2}{k} + \frac{2\epsilon^2}{k} \right)$$

$$\leq 40\eta_0^2 \frac{\Lambda_0}{(k-1)^2} + 35\eta_0^2 \left( \Lambda_1 + 1 \right) \epsilon^2 + 6\eta_0^2 \left( \frac{4\Lambda_0}{(k+1)^2} + \left( \Lambda_1 + 2 \right) \epsilon^2 \right)$$

$$= \eta_0^2 \Lambda_0 \left( \frac{40}{(k-1)^2} + \frac{24}{(k+1)^2} \right) + 41\eta_0^2 \left( \Lambda_1 + 2 \right) \epsilon^2.$$

Since $p_k = \frac{2}{k+1}$, we have for $k \geq 4$

$$\frac{\mathbb{E} \left\| \mathbf{v}_k - \mathbf{v}_{k-1} \right\|^2}{p_k^2} \leq \frac{(k+1)^2}{4} \left[ \eta_0^2 \Lambda_0 \left( \frac{40}{(k-1)^2} + \frac{24}{(k+1)^2} \right) + 41\eta_0^2 \left( \Lambda_1 + 2 \right) \epsilon^2 \right]$$

$$\leq 30\eta_0^2 \Lambda_0 + 11\eta_0^2 \left( \Lambda_1 + 2 \right) \epsilon^2 (k+1)^2$$

$$\overset{(i)}{\leq} 30\eta_0^2 \Lambda_0 + 11\eta_0^2 \left( \Lambda_1 + 2 \right) \epsilon^2 \frac{\Delta^2}{\epsilon^2}$$

$$= 30\eta_0^2 \Lambda_0 + 11\eta_0^2 \left( \Lambda_1 + 2 \right) \Delta^2,$$

where $(i)$ is due to $k \leq N < \frac{\Delta}{\epsilon}$. So we obtain

$$\sum_{k=4}^{N} \frac{(1-p_k)\mathbb{E} \left[ \left\| \mathbf{v}_k - \mathbf{v}_{k-1} \right\|^2 \right]}{p_k^2} \leq \sum_{k=1}^{N} \frac{\mathbb{E} \left[ \left\| \mathbf{v}_k - \mathbf{v}_{k-1} \right\|^2 \right]}{p_k^2}$$

$$\leq \sum_{k=1}^{N} \left[ 30\eta_0^2 \Lambda_0 + 11\eta_0^2 \left( \Lambda_1 + 2 \right) \Delta^2 \right]$$

$$\leq \frac{\Delta \left( 30\eta_0^2 \Lambda_0 + 11\eta_0^2 \left( \Lambda_1 + 2 \right) \Delta^2 \right)}{\epsilon}.$$

For $k = 3$, we have

$$\frac{(1-p_3)\mathbb{E} \left\| \mathbf{v}_3 - \mathbf{v}_2 \right\|^2}{p_3^2} = 2\mathbb{E} \left\| \mathbf{v}_3 - \mathbf{v}_2 \right\|^2 \leq 2 \left( 10\eta_0^2 \Lambda_0 + 35\eta_0^2 \left( \Lambda_1 + 2 \right) \epsilon^2 \right).$$

Moreover, we have

$$
\begin{aligned}
\frac{(1-p_2)\mathbb{E}\|\mathbf{v}_2 - \mathbf{v}_1\|^2}{p_2^2} &= \frac{3}{4}\mathbb{E}\|\mathbf{v}_2 - \mathbf{v}_1\|^2 \\
&\leq \frac{3}{2}\mathbb{E}\left[\left(\frac{\eta_0}{6} + \eta_1\right)^2 \left\|\widetilde{F}(\mathbf{u}_0)\right\|^2 + \left(\frac{3\eta_1}{4} + \eta_2\right)^2 \left\|\widetilde{F}(\mathbf{u}_1)\right\|^2\right] \\
&\leq \frac{3}{2}\eta_0^2\left[\frac{49}{36}\left(2\Lambda_0 + 2\Lambda_1\epsilon^2 + \epsilon^2/2\right) + \frac{49}{16}\left(\frac{\Lambda_0}{2} + 2\Lambda_1\epsilon^2 + 4\epsilon^2\right)\right] \\
&\leq 7\eta_0^2\Lambda_0 + 14\left(\Lambda_1 + 2\right)\epsilon^2.
\end{aligned}
$$

Note that $p_1 = 1$, so we have

$$
\begin{aligned}
\mathbb{E}\left[M\right] &\leq \frac{8\sigma^2\Delta}{\epsilon^3} + \frac{16\sigma^2}{\epsilon^2} + \frac{2\Delta}{\epsilon} + \frac{16L^2}{\epsilon^2}\sum_{k=1}^{N}\frac{(1-p_k)\mathbb{E}\left[\|\mathbf{v}_k - \mathbf{v}_{k-1}\|^2\right]}{p_k^2} \\
&\leq \frac{8\sigma^2\Delta}{\epsilon^3} + \frac{16\sigma^2}{\epsilon^2} + \frac{2\Delta}{\epsilon} + \frac{16L^2\Delta\left(24\eta_0^2\Lambda_0 + 5\eta_0^2\left(\Lambda_1 + 2\right)\Delta^2\right)}{\epsilon^3} \\
&\quad + \frac{32L^2\left(10\eta_0^2\Lambda_0 + 35\eta_0^2\left(\Lambda_1 + 2\right)\epsilon^2\right)}{\epsilon^2} + \frac{16L^2\left(7\eta_0^2\Lambda_0 + 14\left(\Lambda_1 + 2\right)\epsilon^2\right)}{\epsilon^2} \\
&= \mathcal{O}\left(\frac{\sigma^2 L\|\mathbf{u}_0 - \mathbf{u}^*\| + L^3\|\mathbf{u}_0 - \mathbf{u}^*\|^3}{\epsilon^3}\right),
\end{aligned}
$$

where we assume without loss of generality that $L\|\mathbf{u}_0 - \mathbf{u}^*\| \geq 1$, thus completing the proof. $\square$

# E   Omitted proofs from Section 5

**Theorem 5.1.** *Given $L$-Lipschitz and $\mu$-sharp $F$ and the precision parameter $\epsilon$, Algorithm 3 outputs $\mathbf{u}_N$ with $\mathbb{E}[\|\mathbf{u}_N - \mathbf{u}^*\|^2] \leq \epsilon^2$ as well as $\mathbb{E}\left[\|F(\mathbf{u}_N)\|^2\right] \leq L^2\epsilon^2$ after $N = \mathcal{O}\left(\frac{L}{\mu}\log\frac{\|\mathbf{u}_0 - \mathbf{u}^*\|}{\epsilon}\right)$ iterations with at most $\mathcal{O}\left(\frac{\sigma^2(\mu + L)\log(\|\mathbf{u}_0 - \mathbf{u}^*\|/\epsilon) + L^3\|\mathbf{u}_0 - \mathbf{u}^*\|^2}{\mu^3\epsilon^2}\right)$ queries to $\widehat{F}$ in expectation.*

*Proof.* Let $\mathcal{G}_{k-1}$ be the natural filtration of all the random variables used up to (and including) the $(k-1)^{\text{th}}$ outer loop. By Theorem 4.1, we have

$$
\mathbb{E}\left[\|F(\mathbf{u}_k)\|^2 \,|\, \mathcal{G}_{k-1}\right] \leq \frac{\Lambda_0^{(k)}}{(K+1)(K+2)} + \Lambda_1^{(k)}\epsilon_k^2, \tag{E.1}
$$

where $\Lambda_0^{(k)} = \frac{4(L^2\eta_0\underline{\eta} + 1)\|\mathbf{u}_{k-1} - \mathbf{u}^*\|^2}{\underline{\eta}^2}$ and $\Lambda_1^{(k)} = \frac{5(1 + M\underline{\eta}\eta_0)}{M\underline{\eta}^2}$.

By the sharpness condition, we have

$$
\begin{aligned}
\|\mathbf{u}_k - \mathbf{u}^*\|^2 &\leq \frac{1}{\mu}\langle F(\mathbf{u}_k) - F(\mathbf{u}^*), \mathbf{u}_k - \mathbf{u}^*\rangle \\
&\overset{(i)}{\leq} \frac{1}{\mu}\langle F(\mathbf{u}_k), \mathbf{u}_k - \mathbf{u}^*\rangle \\
&\overset{(ii)}{\leq} \frac{1}{\mu}\|F(\mathbf{u}_k)\|\|\mathbf{u}_k - \mathbf{u}^*\|,
\end{aligned}
$$

where $(i)$ is because $\mathbf{u}^*$ is a solution to (SVI), and we use Cauchy-Schwarz inequality for $(ii)$. Taking expectation conditional on $\mathcal{G}_{k-1}$ on both sides, we have

$$
\mathbb{E}\left[\|F(\mathbf{u}_k)\|^2 \,|\, \mathcal{G}_{k-1}\right] \geq \mathbb{E}\left[\mu^2\|\mathbf{u}_k - \mathbf{u}^*\|^2 \,|\, \mathcal{G}_{k-1}\right],
$$

which leads to

$$
\mathbb{E}\left[\|\mathbf{u}_k - \mathbf{u}^*\|^2 \,|\, \mathcal{G}_{k-1}\right] \leq \frac{1}{\mu^2}\left[\frac{\Lambda_0^{(k)}}{(K+1)(K+2)} + \Lambda_1^{(k)}\epsilon_k^2\right].
$$

If we choose $K \geq \frac{4\sqrt{L^2\eta_0\underline{\eta}+1}}{\mu\underline{\eta}}$, we have $\frac{\Lambda_0^{(k)}}{(K+1)(K+2)} \leq \frac{\mu^2\|\mathbf{u}_{k-1}-\mathbf{u}^*\|^2}{4}$. On the other hand, by our choice of $\epsilon_k$ in Algorithm 3, we obtain

$$\Lambda_1^{(k)}\epsilon_k^2 \leq \frac{5\left(1+M\underline{\eta}\eta_0\right)}{M\underline{\eta}^2}\frac{\mu^2\epsilon^2 M\underline{\eta}^2}{20\left(1+M\underline{\eta}\eta_0\right)} \leq \frac{\mu^2\epsilon^2}{4}.$$

So we have

$$\mathbb{E}\left[\|\mathbf{u}_k-\mathbf{u}^*\|^2\,\middle|\,\mathcal{G}_{k-1}\right] \leq \frac{\|\mathbf{u}_{k-1}-\mathbf{u}^*\|^2}{4} + \frac{\epsilon^2}{4}.$$

Taking expectation with respect to all the randomness on both sides, we obtain

$$\mathbb{E}\left[\|\mathbf{u}_k-\mathbf{u}^*\|^2\right] \leq \frac{\mathbb{E}\left[\|\mathbf{u}_{k-1}-\mathbf{u}^*\|^2\right]}{4} + \frac{\epsilon^2}{4}. \tag{E.2}$$

Recursively using Inequality (E.2) till $k=0$, we have

$$\mathbb{E}\left[\|\mathbf{u}_k-\mathbf{u}^*\|^2\right] \leq \frac{1}{4^k}\|\mathbf{u}_0-\mathbf{u}^*\|^2 + \sum_{i=1}^{k}\frac{\epsilon^2}{4^i} \leq \frac{1}{4^k}\|\mathbf{u}_0-\mathbf{u}^*\|^2 + \sum_{i=1}^{\infty}\frac{\epsilon^2}{4^i} \leq \frac{1}{4^k}\|\mathbf{u}_0-\mathbf{u}^*\|^2 + \frac{\epsilon^2}{3}.$$

Hence, after $\left\lceil \log\left(\frac{\sqrt{6}\|\mathbf{u}_0-\mathbf{u}^*\|}{2\epsilon}\right)\right\rceil$ outer loops, the Algorithm 3 can output a point $\mathbf{u}_k$ such that $\mathbb{E}\left[\|\mathbf{u}_k-\mathbf{u}^*\|^2\right] \leq \epsilon^2$, as well as $\mathbb{E}\left[\|F(\mathbf{u}_k)\|^2\right] \leq L^2\epsilon^2$. More specifically, the total number of iterations such that the algorithm can return a point $\mathbf{u}_k$ such that $\mathbb{E}\|\mathbf{u}_k-\mathbf{u}^*\|^2 \leq \epsilon^2$ will be

$$\left\lceil \log\left(\frac{\sqrt{6}\|\mathbf{u}_0-\mathbf{u}^*\|}{2\epsilon}\right)\right\rceil\left\lceil\frac{4\sqrt{L^2\eta_0\underline{\eta}+1}}{\mu\underline{\eta}}\right\rceil = \mathcal{O}\left(\frac{L}{\mu}\log\frac{\|\mathbf{u}_0-\mathbf{u}^*\|}{\epsilon}\right).$$

Next we come to bound the expected number of the stochastic oracle queries for each call to Algorithm $\mathcal{A}$. Denote $i$-th iterate in $k$-th call as $\mathbf{u}_i^{(k)}$ and $\mathbf{v}_i^{(k)}$, and let $K = \left\lceil\frac{4\sqrt{L^2\eta_0\underline{\eta}+1}}{\mu\underline{\eta}}\right\rceil$, then proceeding as in the proof of Corollary D.3, we obtain

$$\mathbb{E}\left[M_k|\mathcal{G}_{k-1}\right] = \mathbb{E}\left[\sum_{i=0}^{K}m_i^{(k)}\,\middle|\,\mathcal{G}_{k-1}\right]$$

$$\leq \frac{16\sigma^2}{\epsilon_k^2} + 2 + \sum_{i=1}^{K}\left(\frac{8\sigma^2}{\epsilon_k^2} + \frac{16(1-p_i)L^2\mathbb{E}\left[\left\|\mathbf{v}_i^{(k)}-\mathbf{v}_{i-1}^{(k)}\right\|^2\,\middle|\,\mathcal{G}_{k-1}\right]}{p_i^2\epsilon_k^2} + 2\right)$$

$$= \frac{16\sigma^2}{\epsilon_k^2} + 2(K+1) + \frac{8\sigma^2 K}{\epsilon_k^2} + \sum_{i=1}^{K}\frac{16(1-p_i)L^2\mathbb{E}\left[\left\|\mathbf{v}_i^{(k)}-\mathbf{v}_{i-1}^{(k)}\right\|^2\,\middle|\,\mathcal{G}_{k-1}\right]}{p_i^2\epsilon_k^2}, \tag{E.3}$$

where $M_k$ is the total number of queries at the $k^{\text{th}}$ call. Notice that $K = \left\lceil\frac{4\sqrt{L^2\eta_0\underline{\eta}+1}}{\mu\underline{\eta}}\right\rceil = \mathcal{O}\left(\frac{L}{\mu}\right)$ and $\epsilon_k^2 = \frac{\mu^2\epsilon^2 M\underline{\eta}^2}{20\left(1+M\underline{\eta}\eta_0\right)} = \mathcal{O}\left(\mu^2\epsilon^2\right)$, then it remains to bound $\frac{\mathbb{E}\left[\left\|\mathbf{v}_i^{(k)}-\mathbf{v}_{i-1}^{(k)}\right\|^2\right]}{p_i^2}$ for $1 \leq i \leq K$. The proof of Lemma D.3 shows that for $i \geq 4$

$$\frac{\mathbb{E}\left[\left\|\mathbf{v}_i^{(k)}-\mathbf{v}_{i-1}^{(k)}\right\|^2\,\middle|\,\mathcal{G}_{k-1}\right]}{p_i^2} \leq 30\eta_0^2\Lambda_0^{(k)} + 11\eta_0^2\left(\Lambda_1^{(k)}+2\right)\epsilon_k^2(i+1)^2.$$

On the other hand, for $1 \leq i \leq 3$, we have $\sum_{i=1}^{3}\frac{\mathbb{E}\left[\left\|\mathbf{v}_i^{(k)}-\mathbf{v}_{i-1}^{(k)}\right\|^2\right]}{p_i^2} = o\left(\frac{L^2\|\mathbf{u}_{k-1}-\mathbf{u}^*\|^2}{\epsilon}\right)$, we obtain

$$\sum_{i=1}^{K}\frac{16(1-p_i)L^2\mathbb{E}\left[\left\|\mathbf{v}_i^{(k)}-\mathbf{v}_{i-1}^{(k)}\right\|^2\,\middle|\,\mathcal{G}_{k-1}\right]}{p_i^2\epsilon_k^2} = \mathcal{O}\left(\frac{L^3\|\mathbf{u}_{k-1}-\mathbf{u}^*\|^2}{\mu^3\epsilon^2}\right).$$

Combining last inequality with Inequality (E.3) and taking expectations on both sides, we obtain

$$\mathbb{E}[M_k] = \mathcal{O}\Big(\frac{\sigma^2(\mu + L) + L^3 \mathbb{E}\big[\|\mathbf{u}_{k-1} - \mathbf{u}^*\|^2\big]}{\mu^3 \epsilon^2}\Big).$$

Telescoping from $k = 1$ to $N = \Big\lceil \log\Big(\frac{\sqrt{6}\|\mathbf{u}_0 - \mathbf{u}^*\|}{2\epsilon}\Big)\Big\rceil$ and noticing that

$$\mathbb{E}\big[\|\mathbf{u}_k - \mathbf{u}^*\|^2\big] \leq \frac{1}{4}\mathbb{E}\big[\|\mathbf{u}_{k-1} - \mathbf{u}^*\|^2\big] + \frac{\epsilon^2}{4} \leq \frac{1}{4^k}\|\mathbf{u}_0 - \mathbf{u}^*\|^2 + \frac{\epsilon^2}{3},$$

we have

$$\sum_{k=1}^{N} \mathbb{E}\|\mathbf{u}_{k-1} - \mathbf{u}^*\|^2 \leq \|\mathbf{u}_0 - \mathbf{u}^*\|^2 \sum_{k=1}^{\infty} \frac{1}{4^k} + \frac{N\epsilon^2}{3} \leq \frac{\|\mathbf{u}_0 - \mathbf{u}^*\|^2}{3} + \frac{N\epsilon^2}{3}.$$

Hence, we finally arrive at

$$\mathbb{E}\Big[\sum_{k=1}^{N} M_k\Big] = \mathcal{O}\Big(\frac{\sigma^2(\mu + L)\log(\|\mathbf{u}_0 - \mathbf{u}^*\|/\epsilon) + L^3 \|\mathbf{u}_0 - \mathbf{u}^*\|^2}{\mu^3 \epsilon^2}\Big),$$

which completes the proof. $\qquad\square$