# OpenReview forum: "Stochastic Halpern Iteration with Variance Reduction for Stochastic Monotone Inclusions"
_NeurIPS.cc/2022/Conference — NeurIPS 2022 Accept_

### Official Review · Reviewer_KtEb · 2022-07-11

**Rating:** 6
**Confidence:** 3
**Soundness:** 3 good
**Presentation:** 3 good
**Contribution:** 3 good

**Summary:**

The work focuses on solving stochastic monotone inclusions problems. where the information on the problem is available through the calls of unbiased stochastic oracle with bounded variance. Moreover, stochastic realizations are assumed to be Lipschitz-continuous in expectation. In particualr, the authors study the convergence of Halpern iterations with stochastic unbiased estimator and PAGE-like variance reduced estimator and derive new complexity bounds for the expected norm of the operator in 3 different cases: 1) when the target operator is cocoercive, 2) monotone and Lipschitz, and 3) $\mu$-sharp and Lipschitz. The derived upper bounds for Halpern iterations with PAGE-like estimator are near-optimal in $\epsilon$ for the second and the third cases.

**Questions:**

1. It might be hard for some non-expert readers to understand Section 1.2 before reading the technical details from the further sections. I suggest to move this discussion further in the paper.
2. Line 247, "... with probability at least $1-\epsilon$, both $\|F(u_k)\|$ and $\| \tilde{F} \|$ are $O(\epsilon)$ in expectation": this sentence is unclear to me. Could you please explain this formally?
3. Line 278: The upper bound for $\mathbb{E}\|F(u_i)\|$ does not give an upper bound for $\mathbb{E}\|F(u_i)\|^2$, since $(\mathbb{E}\|F(u_i)\|)^2 \leq \mathbb{E}\|F(u_i)\|^2$. Nevertheless, the proof can be easily fixed if one assumes a proper upper bound on $\mathbb{E}\|F(u_i)\|^2$ in (i). Moreover, the resulting upper bound from line 278 should have $\epsilon^2$-term as well.
4. Line 282: one should point out that the number of oracle call is given in expectation.
5. Theorem 5.1: the resulting upper bound has a term $O(\frac{L^3}{\mu^3\epsilon^3})$, which is typically extremely large, since $L/\mu$ is large. For such problems, one can simply apply SGDA and get better rate (in terms of the dependence on $\mu$ and $L$), see [28] and also Beznosikov et al. "Stochastic gradient descent-ascent: Unified theory and new efficient methods." arXiv preprint arXiv:2202.07262 (2022). Indeed, these assumptions imply star-cocoercivity, which is sufficient to derive $O(1/\mu^2 \varepsilon^2)$ rate in this settings (see Beznosikov et al. (2022)).
6. What is the main challenge in the analysis? If I am not mistaken, in view of Corollary 2.2 everything reduces to the analysis of deterministic Halpern iterations with bounded additive error in the operator values. Analyzing optimization methods with bounded additive error is usually relatively straightforward.

**Limitations:**

There are no potential negative social impact: the work is purely theoretical.

**Strengths And Weaknesses:**

## Strengths
1. **Clarity.** The paper is clearly written. The discussion of the related work is detailed enough to place the work in the literature properly.
2. **Guarantees for the last iterate.** In contrast to the majority of existing results for solving stochastic variational inequalities/monotone inclusions, the results of this work are derived for the last iterate, which is usually more challenging than obtaining results for the averaged or the best iterate.

## Weaknesses
1. **Large batchsizes.** The derived results require to use $\Theta(1/\epsilon^2)$ batchsizes at each iteration on average. These batchsizes are very large and, therefore, cannot be used in the majority of practical tasks. Moreover, in the analysis, under such batchsizes the stochasticity almost vanishes : as Corollary 2.2 states $\mathbb{E}\|\tilde F(u_k) - F(u_k)\|^2 \leq \frac{\epsilon^2}{k}$ implying that the method is almost deterministic. The analysis of such method reduces to the analysis of deterimistic method with bounded additive error (in expectation). All these facts dramatically limit the significance of the contributions.
2. **No empirical evaluation.** The paper proposes a new method, but does not studies its behavior in numerical experiments. In particular, given the limitations of the analysis, it would be interesting to study the impact of the batchsize on the convergence of Halpern-PAGE and Halpern-SGD at least in the experiments. Moreover, it would be also important to validate the importance of using PAGE estimator in comparison to SGD and the importance of using Halpern iterations in the stochastic settings.

---

> ### Author Response · Authors · 2022-08-01
> **Response to Reviewer KtEb**
>
> Thank you for your feedback and questions. We hope that the answers provided below appropriately address your concerns and that you will consider increasing your score. Due to the space limit, please let us know if any additional information would be helpful.
>
> > Q1
>
> Thank you for the suggestion, we will do this in the final version of the paper, as it requires us more work to meet the page limit.
>
> > Q2
>
> As is dicussed in Lines 245-246 (Lines 251-252 in the revision), we can verify by Chebyshev bound that $\mathbb{P}[||F(\mathbf{u}_k) - \tilde{F}(\mathbf{u}_k)|| \geq \epsilon] \leq \frac{1}{k}$ due to low error for the estimates $\mathbb{E}[||F(\mathbf{u}_k) - \tilde{F}(\mathbf{u}_k)||^2] \leq \frac{\epsilon^2}{k}$. Hence, after $\Omega(\frac{1}{\epsilon})$ iterations, we have $\mathbb{P}[||F(\mathbf{u}_k) - \tilde{F}(\mathbf{u}_k)|| \geq \epsilon] \leq \epsilon$. Combining with results in Theorem 3.1, i.e., $\mathbb{E}[||F(\mathbf{u}_k)||] \leq 4\epsilon$ after $\mathcal{O}(\frac{1}{\epsilon})$ iterations, we can verify by triangle inequality that $\mathbb{E}[||\tilde{F}(\mathbf{u}_k)||] \leq \mathbb{E}[||\tilde{F}(\mathbf{u}_k)|| + ||F(\mathbf{u}_k)||] \leq 5\epsilon$ with probability at least $1 - \epsilon$.
>
> > Q3 & Q4
>
> In the proof for Theorem 3.1 in Appendix C, we proved the upper bound $(\mathbb{E}[||F(\mathbf{u}_k)||^2])^{1/2} \leq \frac{\Lambda_0}{k} + \Lambda_1\epsilon$, which induces the upper bound on $\mathbb{E}[||F(\mathbf{u}_k)||]$ in the Theorem by Jensen's inequality. The upper bound from Line 278 holds for all $i \leq k - 1 \leq \mathcal{O}(\frac{1}{\epsilon})$. The number of stochastic oracle calls in Line 282 is given in expectation, which is standard for estimators like PAGE. We have addressed these writing issues in the revision of the paper.
>
> > Q5
>
> Thanks for pointing out these papers; we will discuss them in the final version of the paper, as it requires us more work to meet the page limit. We would like to also point out that the paper by Beznosikov et al.~is concurrent to ours. In the sharp monotone Lipschitz settings, their dependence on $\mu$ and $L$ is better than ours, as you correctly point out, however, the difference is not as high as it may appear at first once the bound is properly translated to our setting. The mentioned papers assume that the operator is $\ell_D$-cocoercive. Operators that are $L$-Lipschitz and $\mu$-sharp are only $\frac{L^2}{\mu}$-cocoercive. Further, the bounds reported in e.g., Beznosikov et al. (2022), Corollary 3.3 and Loizou et al., (2021) are for the distance to opt. Once translated to a bound on the norm of the operator using Lipschitzness, the required error is $\frac{\epsilon^2}{L^2},$ leading to complexity $O(\frac{L^2}{\mu^2}\log(\frac{1}{\epsilon}) + \frac{\sigma^2 L^2}{\mu^2 \epsilon^2}).$ The bound we obtained in Theorem 5.1 is $O(\frac{\sigma^2 (L + \mu)}{\mu^3 \epsilon^2}\log(\frac{1}{\epsilon}) + \frac{L^3}{\mu^3 \epsilon^2}).$
>
> Our high order dependence on $L/\mu$ comes from applying recursive variance reduction to monotone (convex-concave) settings and bounding the operator norm (as opposed to the primal-dual gap or distance to opt). The analysis in [28] and Beznosikov et al. (2022) is specialized to the quasi-strongly monotone assumption and primal-dual gap guarantees, and still unclear how to extend to the more general monotone Lipschitz setups that are the main focus of our work. Further, Algorithm 3 and Theorem 5.1 in our work are short extensions from monotone Lipschitz settings in our paper that lead to near-optimal dependencies on $\epsilon$ and $\sigma$. Although this part is not the main focus of our paper, we still believe there is value to including results for the sharp case, for completeness.

---

> > ### Author Response · Authors · 2022-08-01
> > **Response to Reviewer KtEb (Cont.)**
> >
> > > Q6 and Weakness 1
> >
> > Thank you for raising this question; the summary is as follows:
> >
> > 1. Standard analysis in stochastic optimization generally is based on deterministic settings and uses either batch samples or step sizes to control the error due to noise variance. In this problem, it is unclear whether we can control this error via carefully chosen step sizes; the same issue appears in all existing work on bounding the operator norm in VIs (monotone inclusion) we are aware of and remains open. Instead, we handle the additive error terms in the analysis using batch samples to reduce the variance, which is an approach taken by a large fraction of the ML community that works on variance reduction techniques. Our quantitative theoretical analysis is non-trivial from both aspects of stochastic Halpern iteration and variance reduction.
> >
> > 2. The variance bound $\mathbb{E}[||F(\mathbf{u}_k) - \tilde{F}(\mathbf{u}_k)||^2] \leq \frac{\epsilon^2}{k}$ in Corollary 2.2 is a must for stochastic Halpern iteration to handle error accumulation, which also appears in a stochastic version of Nesterov's fast gradient method for smooth convex minimization. So the averaged batch size at each iteration needs to be $\Theta(\frac{1}{\epsilon^2})$, which is the first known result for stochastic Halpern iteration and leads to near-optimal total complexity in this problem setting. More details can be found in Lee and Kim (2021) cited in our paper, and we will add more discussions about it in our final revision due to the current page limit.
> >
> > 3. Analyzing stochastic Halpern iteration with additional noise error terms is quite different from the deterministic case. In deterministic settings, we can prove that the potential function is non-increasing, which leads to the upper bound on the operator norm. In the stochastic settings, we are choosing the same potential function for analysis because it is standard for Halpern-type algorithms. However, we get a recursion for such a potential function with additional error terms as in Line 569 (Line 594 in the revision) in the supplemental. This requires us to not only have a bound on the noise variance but also do intricate inductive arguments to resolve the recursion. Note that such a recursion does not appear in the analysis for extrapolated Halpern iteration in monotone Lipschitz settings. But the analysis in both setups in this paper shares the same challenge discussed in the next point.
> >
> > 4. The complexity in stochastic optimization is based on the (expected) total number of stochastic oracle queries needed to attain certain accuracy. If we only use mini-batch estimators to bound the noise variance, it can only give us $\mathcal{O}(1/\epsilon^4)$ complexity, which is much worse than our results. Also, Corollary 2.2 does not provide an upper bound on the number of stochastic oracle queries, as the batch size $S_2^{(k)}$ is adaptive with respect to $||u_k - u_{k - 1}||$. This requires us to do more technical analysis to bound these terms (see, e.g., the proofs of Corollary 3.4 and Lemma D.3), which is not the case in standard PAGE analysis. Further, the complexity results from our analysis are optimal up to log terms in terms of $\epsilon$ and $\sigma$, as is discussed in Appendix A, which are the first and the best-known results for the problem classes considered in this paper.
> >
> > > Weakness 2
> >
> > We would like to reiterate that our contribution is theoretical, with our main focus being on proving near-optimal complexity guarantees for stochastic monotone inclusion settings (minimizing gradient norm in unconstrained min-max problems), where the guarantee is on the last-iterate. However, we have uploaded a revised version of our paper with preliminary numerical results done in limited time, for illustration.
> >
> > We gave a more detailed summary of our added results to a similar question raised by Reviewer qc4S (empirical evaluation) and are not stating them here to avoid repetition, but please let us know if we should clarify further. In the experiment, we consider solving robust least squares with the Lagrangian relaxation, where $u = (x, y)^T$ and $F(u) = (\nabla_{x}L_{\lambda}(x, y), -\nabla_{y}L_{\lambda}(x, y))^T$ for $L_{\lambda}(x, y) = \frac{1}{2n}||Ax - y||^2_2 - \frac{\lambda}{2n}||y - b||_2^2$. We compare vanilla Halpern iteration, extrapolated Halpern iteration (E-Halpern), and Restarted E-Halpern with Alternating Gradient Descent (GDA), Extragradient method (EG), and Popov's method (Popov), in stochastic settings. Additionally, we compare E-Halpern with the PAGE estimator against single-sample and mini-batch estimators. More details are provided in the Numerical Experiment section in the revised version of the paper.

---

> ### Comment · Reviewer_KtEb · 2022-08-08
> **Reply to authors**
>
> I thank the authors for their detailed response. I have read the response in detail, but I still have several questions and concerns, let me clarify.
>
> ### On Question 2
>
> I still do not understand the claim: the authors write that an expectation is smaller than $\mathcal{O}(\epsilon)$ **with some probability**. In the derived results, the expectation is full, i.e., expected (squared) norm of operator is a deterministic quantity, not a random variable. In particular, the authors write in their response that "$\mathbb{E}[\|\tilde{F}(u_k)\|] \leq 5\epsilon$ with probability at least $1-\epsilon$", which is a mathematically inaccurate statement. Moreover, in the derivation, there is a mistake: the authors use **inside the expectation** the upper bound for the norm of the difference that holds **with some probability**.
>
> ### On Question 5
>
> I believe that in the $\mu$-sharp (quasi-strongly monotone) case it is more natural to consider $\mathbb{E}||u^k - u^\ast||^2$ as the main convergence criterion. In terms of this criterion, the result from Beznosikov et al. (2022) has $\frac{L}{\mu}$ times better $\mathcal{O}(1/\epsilon)$ term.
>
> ### On large batchsizes
>
> > *Instead, we handle the additive error terms in the analysis using batch samples to reduce the variance, which is an approach taken by a large fraction of the ML community that works on variance reduction techniques.*
>
> Typically, variance reduced methods require large-batch/full-batch computations only rarely. The method proposed in this work requires large-batch computations at each step.
>
> > *The variance bound $\mathbb{E}\left[ || F(u_k) - \tilde{F}(u_k) ||^2 \right] \leq \frac{\epsilon^2}{k}$ in Corollary 2.2 is a must for stochastic Halpern iteration to handle error accumulation, which also appears in a stochastic version of Nesterov's fast gradient method for smooth convex minimization.*
>
> The authors do not prove that $\mathbb{E}\left[ || F(u_k) - \tilde{F}(u_k) ||^2 \right] \leq \frac{\epsilon^2}{k}$ is a neccessary condition to ensure the convergence of stochastic Halpern iteration, so, it is unclear why it is a must: maybe there exists a proof without large batchsizes? Regarding the stochastic accelerated methods for smooth convex minimization: there exist stochastic accelerated methods for smooth convex minimization that do not require large batchsizes, e.g., AC-SA from [Ghadimi, S., & Lan, G. (2012). Optimal stochastic approximation algorithms for strongly convex stochastic composite optimization i: A generic algorithmic framework. SIAM Journal on Optimization, 22(4), 1469-1492].
>
> > *Analyzing stochastic Halpern iteration with additional noise error terms is quite different from the deterministic case. [...] However, we get a recursion for such a potential function with additional error terms as in Line 569 (Line 594 in the revision) in the supplemental. [...]*
>
> The proof follow exactly the same pattern as the proofs in the deterministic settings. The additional error terms like in Line 594 are estimated via Lemma 2.1, which is almost the same ad the main derivation in the analysis of PAGE, and Lemma C.3, which is relatively straightforward. After that, one just need to take full expectation, telescope the inequality from Line 594, and do some simple arithmetics. So, the analysis essentially follows the deterministic case.
>
> > *The complexity in stochastic optimization is based on the (expected) total number of stochastic oracle queries needed to attain certain accuracy. If we only use mini-batch estimators to bound the noise variance, it can only give us $\mathcal{O}\left(\frac{1}{\epsilon^4}\right)$ complexity, which is much worse than our results.*
>
> I believe it is not fair to compare the results for simple mini-batch estimator and PAGE estimator, since these results are derived for different setups. The result for mini-batch estimator does not require Assumption 3, while the result for PAGE estimator crucially relies on that (in Lemma 2.1). This is quite natural and the same phenomenon is known for non-convex minimization (SGD is optimal under bounded variance, but PAGE is better and optimal under bounded variance + smoothness in expectation).
>
> ### On the numerical experiments
>
> I thank the authors for adding the numerical experiments. However, it is not clear what stepsizes and batchsizes were picked for each method. Could the authors report these parameters? I am also curious how exactly the authors tuned batchsizes for PAGE estimator and ensure that they decrease with $\mathcal{O}(1/k)$ rate. Also, the tuning procedure itself is not clear: the authors write that the parameters were tuned to achieve faster convergence. Could the authors clarify what exactly does this mean? Do the authors use some accuracy threshold or choose the parameters that allow to achieve fastest convergence after certain number of steps? Also, it is not clear how many runs were averaged for each method.

---

> > ### Author Response · Authors · 2022-08-09
> > **Response to Reviewer KtEb (Post-Rebuttal)**
> >
> > Thank you for your further feedback and questions.
> >  ### On Question 2
> > Thank you for bringing up this point and apologies for the poorly worded initial response. Let us clarify now. We can make two main statements here:
> >
> > 1. After $O(\frac{1}{\epsilon})$ iterations, once we have $||\tilde{F}(\mathbf{u}_k)|| \leq \epsilon$, $||F(\mathbf{u}_k)||$ is also $O(\epsilon)$ with probability at least $1 - \epsilon$. This is because our algorithm ensures $\mathbb{E}[||F(\mathbf{u}_k) - \tilde{F}(\mathbf{u}_k)||^2] \leq \frac{\epsilon^2}{k}$, which leads to $\mathbb{P}[||F(\mathbf{u}_k) - \tilde{F}(\mathbf{u}_k)|| \geq \epsilon] \leq \frac{1}{k}$ by Chebyshev bound. So $||\tilde{F}(\mathbf{u}_k)||$ can be a stopping criterion that can also be efficiently computed.
> >
> > 2. If one wanted to make a statement about the expectation of the norm of the estimator $\mathbb{E}[||\tilde{F}(\mathbf{u}_k)||]$ (instead of $\mathbb{E}[||F(\mathbf{u}_k)||]$ in the theorem statement), then we could simply use the variance bound to claim that because $\mathbb{E}[||F(\mathbf{u}_k)||] \leq 4\epsilon$ after $O(\frac{1}{\epsilon})$ iterations, we also have $\mathbb{E}[||\tilde{F}(\mathbf{u}_k)||] \leq \mathbb{E}[||{F}(\mathbf{u}_k)||] + \mathbb{E}[||F(\mathbf{u}_k) - \tilde{F}(\mathbf{u}_k)||] \leq 5 \epsilon,$ where we use Jensen's inequality to bound $\mathbb{E}[||F(\mathbf{u}_k) - \tilde{F}(\mathbf{u}_k)||]$ via $(\mathbb{E}[||F(\mathbf{u}_k) - \tilde{F}(\mathbf{u}_k)||])^2 \leq \mathbb{E}[||F(\mathbf{u}_k) - \tilde{F}(\mathbf{u}_k)||^2] \leq \frac{\epsilon^2}{k} \leq \epsilon^2,$ as $k \geq 1.$
> >
> > We have revised the paper based on this discussion.
> >
> > ### On Question 5
> > We did not disagree in our response that the dependence on $\mu$ and $L$ in the paper by Beznosikov et al. is better than ours. But we believe that operator norm guarantee stands in its own right, as it is a measure of convergence that can be evaluated as a stopping criterion in practice. Further, we would like to reiterate that their work is concurrent to ours, and the $\mu$-sharp case in our paper is one natural and short extension of the monotone Lipschitz case, which we include for completeness.
> >
> > ### On Large Batch Sizes
> > > Typically, variance reduced methods require large-batch/full-batch computations only rarely. The method proposed in this work requires large-batch computations at each step.
> >
> > This is true. However, the setting of convex minimization is not directly comparable to the small gradient norm guarantee in min-max optimization. Whether it is possible to alternate between frequent small batch sizes and occasionally large batch sizes in the setting we consider remains an interesting and important open question: there are no such results in the literature. There are also no other results that obtain stochastic oracle complexity $O(\frac{1}{\epsilon^3})$ as in our work (for which we show it is near-optimal), to the best of our knowledge.
> >
> > > The authors do not prove that $\mathbb{E}[||F(u_k) - \tilde{F}(u_k)||^2] \leq \frac{\epsilon^2}{k}$ is a necessary condition to ensure the convergence of stochastic Halpern iteration, so, it is unclear why it is a must: maybe there exists a proof without large batchsizes? Regarding the stochastic accelerated methods for smooth convex minimization: there exist stochastic accelerated methods for smooth convex minimization that do not require large batchsizes, e.g., AC-SA from [Ghadimi, S., \& Lan, G. (2012). Optimal stochastic approximation algorithms for strongly convex stochastic composite optimization i: A generic algorithmic framework. SIAM Journal on Optimization, 22(4), 1469-1492].
> >
> > This is a fair point and perhaps our initial response was not properly phrased. What we wanted to say is that some variance control appears inevitable for Halpern iteration: notice the iteration can converge only when $\lambda_k\to0$, and under such a choice the stochastic oracle term (without variance reduction) $(1-\lambda_k)\tilde F(\mathbf{u}_k)$ introduces variance $\Omega(\sigma^2)$ to the last iterate; see also a similar discussion in [Lee \& Kim, NeurIPS 2021]. Regarding the results pointed out by the reviewer: small batches are provably convergent in the setting of stochastic convex optimization and stochastic variational inequalities (wrt the gap function), yet we are not aware of any such results for stochastic monotone inclusions. Hence, although we may agree large batches are a downside, our results are the first of their kind.

---

> > > ### Author Response · Authors · 2022-08-09
> > > **Response to Reviewer KtEb (Post-Rebuttal) Cont.**
> > >
> > > ### On Large Batch Sizes
> > > > The proof follow exactly the same pattern as the proofs in the deterministic settings. The additional error terms like in Line 594 are estimated via Lemma 2.1, which is almost the same ad the main derivation in the analysis of PAGE, and Lemma C.3, which is relatively straightforward. After that, one just need to take full expectation, telescope the inequality from Line 594, and do some simple arithmetics. So, the analysis essentially follows the deterministic case.
> > >
> > > If you take a closer look at Lemma C.2, you will notice that, in addition to the variance term, there is an inner product term that is neither zero in expectation nor directly bounded by the variance. Dealing with this term takes quite a bit of technical work and is handled by an intricate inductive argument carried out in the proof of Theorem 3.1. We are not aware of a similar argument in the existing literature.
> > >
> > > Further, using the variance bound from the PAGE paper is insufficient for obtaining the stochastic oracle complexity bounds reported in our work. The reason is that the batch size $S_2^{(k)}$ depends on the distance between the successive iterates, $||u_k - u_{k - 1}||$. In smooth nonconvex optimization considered in the PAGE paper, this is handled using a descent lemma. In our work, it requires not only unrolling the recursive definition of $u_k,$ but actually bounding all the resulting terms using the bounds on $\mathbb{E}[||F(u_k)||]$ obtained via the inductive argument from Theorem 3.1. While our presentation may make these arguments look "simple" in retrospect, coming up with this analysis is by no means trivial and required significant technical work.
> > >
> > > >  I believe it is not fair to compare the results for simple mini-batch estimator and PAGE estimator, since these results are derived for different setups. The result for mini-batch estimator does not require Assumption 3, while the result for PAGE estimator crucially relies on that (in Lemma 2.1). This is quite natural and the same phenomenon is known for non-convex minimization (SGD is optimal under bounded variance, but PAGE is better and optimal under bounded variance + smoothness in expectation).
> > >
> > > Thank you for this comment. The bound for the simple mini-batch estimator is provided for completeness and as a warm-up. Our main focus is on obtaining last-iterate operator norm guarantee with near-optimal complexity in terms of $\epsilon$ for the standard problem classes with monotone Lipschitz operators, and our results obtained in this paper are the first and best-known so far. Assumption 3 is commonly used in recursive variance reduction methods and holds in many practical settings, such as, for example, in standard risk minimization problems/supervised learning. It is unclear at the moment whether such complexity results can be obtained without Assumption 3, which is beyond the scope of this paper and we leave for future work.
> > >
> > > ### On Experiments
> > > Thank you for the clarifying questions regarding our experiment. Although we may not have mentioned this explicitly, we have already included our code in the revision of the supplemental material, so all the numerical details are provided there as well.
> > >
> > > As we mentioned before, our main contribution is theoretical, and the preliminary numerical results are provided solely for the purpose of illustration. Now going back to your questions:
> > >
> > > 1. As we mentioned in the revised paper and the reply to Reviewer qc4S (which we pointed to in the rebuttal), we used constant batch sizes and stepsizes, tuned them via grid search for each method individually, and reported the performance for the best choice. For PAGE estimator, we empirically tuned the parameters $p_k  = \Theta(\frac{1}{k})$, $S_1^{(k)} = \Theta(k)$ and $S_2^{(k)} = \Theta(1)$ based on our theoretical analysis without pre-assuming accuracy $\epsilon$, as we were comparing the methods over a fixed number of stochastic operator evaluations. Note that because $S_1^{(k)}$ grows with $k$ it is possible that this choice leads to values larger than the dataset size if the iteration count is sufficiently large. Thus, in the case that the value of $S_1^{(k)}$ reaches the size of the dataset in some iteration, we keep it constant from then on.
> > >
> > > 2. The parameters are tuned via grid search to achieve the fastest convergence for a fixed number of stochastic operator evaluations.
> > >
> > > 3. For the experiment displayed in the revision, instead of averaging over many runs, we used the same fixed random seed for each method for illustration in the limited rebuttal time. We have run the experiments multiple times and tried different random seeds, and all of them exhibited similar results to the ones shown in the revised paper. We will summarize and report these results in a later revision of the paper.

---

> > > > ### Comment · Reviewer_KtEb · 2022-08-09
> > > > **Thank you for the clarifications, I am raising my score**
> > > >
> > > > I thank the authors for the additional clarifications. After the discussion with authors, I decided to increase the score due to non-triviality of the obtained results given that they are near-optimal. Below I just leave few comments.
> > > >
> > > > ### On Question 2
> > > >
> > > > I think it is worth to mention in the first point that the algorithm does not guarantee that $\|F(\tilde{u}_k)\| \leq \epsilon$ after $O(1/\epsilon)$ iterations. The guarantee holds only in expectation.
> > > >
> > > > ### On large batchsizes
> > > >
> > > > Indeed, the problem of deriving convergence guarantees in terms of the expected squared norm without assuming large batchsizes is open. Even without anchoring there are no such results.
> > > >
> > > > Moreover, although the inductive argument is natural, I agree that similar arguments appear relatively rare in the analysis. Similar arguments appear in the minimization [1], where the authors also used a bound on the norm of the gradient inductively (see their proof of Theorem F.1). However, in the context of monotone inclusions/min-max problems/variational inequalities, I am not aware of similar arguments used in the proofs. Due to this fact and the near-optimality of the derived results I decided to increase the score.
> > > >
> > > > References:
> > > >
> > > > [1] Gorbunov, E., Danilova, M., & Gasnikov, A. (2020). Stochastic optimization with heavy-tailed noise via accelerated gradient clipping. Advances in Neural Information Processing Systems, 33, 15042-15053.

---

> > > > > ### Author Response · Authors · 2022-08-09
> > > > > **Thank you!**
> > > > >
> > > > > We really appreciate your careful consideration of our response and increasing the score based on our discussion.
> > > > >
> > > > > Thank you for the additional comments; we'll add these suggestions to the revised paper.
> > > > >
> > > > > Finally, thank you once again for the constructive feedback!
> > > > >
> > > > > Authors

---

### Official Review · Reviewer_NtRF · 2022-07-11

**Rating:** 7
**Confidence:** 4
**Soundness:** 4 excellent
**Presentation:** 3 good
**Contribution:** 3 good

**Summary:**

The framework of Monotone variational inequalities capture several well-studied optimization problems, including convex minimization, saddle-point problems, and min-max optimization.
This paper studies stochastic versions of monotone variational inequalities, and presents algorithms based on stochastic Halpern iteration. The paper shows that for Lipschitz-monotone *stochastic* variational inequalities, a two step version of Halpern iteration (which can be thought of as extending the extra-gradient method to variational inequalities) can find a solution that is within $\varepsilon$ of the optimal solution within $O(\varepsilon^{-4})$ iterations. Combining with variance reduction, the paper shows that the iteration count can be improved to  $O(\varepsilon^{-3})$.
Under stronger sharpness assumptions on the operator (this can be thought of as the assumption that the operator is "strongly convex" around the solution, not necessarily everywhere else), the paper establishes that a suitably restarted version can further improve the iteration complexity to $\tilde{O}(\varepsilon^{-2})$.
The paper further establishes, via reductions from stochastic convex minimization, that these rates are optimal in $\varepsilon$ up to logarithmic factors.

Edit (post-rebuttal): The authors clarified that it is more accurate to identify their contributions as combining Halpern iteration with stochastic variance reduction methods to achieve near-optimal guarantees in statistical settings.

**Questions:**

The authors mention in the abstract that the $O(\varepsilon^{-4})$ rate achievable by combining stochastic Halpern iteration with mini-batch estimates is "current state of the art", but I don't see that result being directly attributed to a reference. The only mention of this result seems to be in line 148, attributed to [12]
Q1. Was this result known for stochastic monotone-lipschitz VIs for a computable iterate? "Best iterate" results are less interesting since it is not feasible to compute the best iterate.
Q2. Was this result known for the stochastic mini-batch Halpern iteration?

**Limitations:**

In my view, this is a strong work that establishes near-tight bounds for an important problem, and I don't see any serious limitations. It would be interesting to extend these ideas to a practical setting, but that is beyond the scope of this work.

**Strengths And Weaknesses:**

**Strengths**:
+ Several important recent applications in machine learning, e.g. GANs, adversarial learning, require solution concepts that cannot be captured by minimizing a single variable function. A more powerful framework of min-max optimization captures the solution concepts proposed in these settings. Variational inequalities captures min-max optimization and hence algorithms for Variational inequalities have received renewed attention recently.
+ The stochastic version of the VI problem studied in this paper is very natural given that the operator in most machine learning applications are stochastic in nature. This paper seems to establish (to my knowledge), the first upper bounds in the stochastic setting.
+ The reductions from stochastic convex minimization presented in the paper show that the existing lower bounds for stochastic convex minimization imply that the rates achieved in this paper are near-optimal in the most-well studied Lipschitz monotone setting.

**Weakness**:
- The primary weakness of the paper in my view is that the technical novelty in the algorithms is limited. The main algorithm in the Lipschitz-monotone setting is based on combining the 2-step Halpern iteration studied in [45] and the proof is based on the same potential function, combined with the variance reduced estimator PAGE from [26].

[26] Zhize Li, Hongyan Bao, Xiangliang Zhang, and Peter Richtárik. PAGE: A simple and optimal probabilistic gradient estimator for nonconvex optimization. arXiv preprint arXiv:2008.10898, 2020.
[45] Quoc Tran-Dinh and Yang Luo. Halpern-type accelerated and splitting algorithms for monotone inclusions. arXiv preprint arXiv:2110.08150, 2021.

---

> ### Author Response · Authors · 2022-08-01
> **Response to NtRF**
>
> Thank you for the valuable feedback and positive evaluation of our paper!
>
> Regarding your question about the mini-batch $O(\frac{1}{\epsilon^4})$ complexity result: [12] is the first to our knowledge and for the best (not computable iterate), while the same result for the last iterate and a variant of Halpern iteration can be attributed to Lee \& Kim, (2021). We have clarified this in the revised version.
>
> We also wanted to provide a brief clarification about the contribution: our main contribution is not in algorithm design, but in proving that variants of the classical Halpern iteration can be effectively combined with recursive variance reduction to obtain near optimal (in $\epsilon$ and $\sigma$) complexity guarantees in stochastic settings. We gave more detailed responses to similar questions raised by Reviewer qc4S (contributions) and KtEb (novel technical ideas) and are not stating them here to avoid repetition, but please let us know if we should clarify further.

---

> > ### Comment · Reviewer_NtRF · 2022-08-08
> > **Thank you for your clarifications**
> >
> > >Regarding your question about the mini-batch  $O(\frac{1}{\epsilon^4})$ complexity result: [12] is the first to our knowledge and for the best (not computable iterate), while the same result for the last iterate and a variant of Halpern iteration can be attributed to Lee & Kim, (2021). We have clarified this in the revised version.
> >
> > Thank you for this clarification, and reflecting this in the revision.
> >
> > > We also wanted to provide a brief clarification about the contribution: our main contribution is not in algorithm design, but in proving that variants of the classical Halpern iteration can be effectively combined with recursive variance reduction to obtain near optimal (in  and ) complexity guarantees in stochastic settings. We gave more detailed responses to similar questions raised by Reviewer qc4S (contributions) and KtEb (novel technical ideas) and are not stating them here to avoid repetition, but please let us know if we should clarify further.
> >
> > Thank you for this clarification, and pointing to your responses to the other two reviewers. I added a small note at the end of my original review to reflect this clarification.
> >
> > My score stands.

---

> > > ### Author Response · Authors · 2022-08-09
> > > **Thank you!**
> > >
> > > We were much appreciate your constructive feedback and positive evaluation of our work.

---

### Official Review · Reviewer_qc4S · 2022-07-12

**Rating:** 4
**Confidence:** 3
**Soundness:** 3 good
**Presentation:** 2 fair
**Contribution:** 3 good

**Summary:**

This paper proposes several stochastic Halpern iteration algorithms with recursive variance reduction for solving monotone inclusion problems such as robust regression and adversarial learning.

**Questions:**

1. The proposed algorithm can have a better convergence property. However, it is not clear whether the proposed algorithm converges faster than existing methods in practice.

2. The authors should report some numerical results to show the advantage of the proposed algorithm.

3. What’s the advantage of the proposed algorithm against existing Halpern iteration algorithms?

4. There are several variance reduction techniques such as SVRG, SAGA and SARAH. Why the authors choose the variance reduction technique in SARAH?

5. The writing of this paper should be improved, for instance, “stochastic variants of these methods had not been considered prior to our work”.


**Ethics Review Area:**

["I don’t know"]

**Limitations:**

1. The proposed algorithm can have a better convergence property. However, it is not clear whether the proposed algorithm converges faster than existing methods in practice.

2. The authors should report some numerical results to show the advantage of the proposed algorithm.

3. What’s the advantage of the proposed algorithm against existing Halpern iteration algorithms?

4. There are several variance reduction techniques such as SVRG, SAGA and SARAH. Why the authors choose the variance reduction technique in SARAH?

5. The writing of this paper should be improved, for instance, “stochastic variants of these methods had not been considered prior to our work”.

**Strengths And Weaknesses:**

Strengths：
This paper proposes a stochastic Halpern iteration algorithm, which has O(1/\epsilon^{-3}) stochastic oracle complexity.

Under an additional sharpness assumption, the complexity of the proposed algorithm has a better stochastic complexity, O((1/\epsilon^{-2})log(1/\epsilon)).

Weaknesses：

The proposed algorithm uses both stochastic Halpern iteration and recursive variance reduction techniques, which are NOT new. Thus, the novelty of this paper is limited.

By using an additional sharpness assumption, the proposed algorithm can have a better convergence property. However, it is not clear whether the proposed algorithm converges faster than existing methods in practice.

The authors should report some numerical results to show the advantage of the proposed algorithm.

---

> ### Author Response · Authors · 2022-08-01
> **Response to Reviewer qc4S**
>
> Thank you for the feedback and for the questions you raised. We hope that in light of the clarifications provided below you would consider increasing your score. Please let us know if there is any additional information you would find helpful.
>
> > Weakness: The proposed algorithm uses both stochastic Halpern iteration and recursive variance reduction techniques, which are NOT new. Thus, the novelty of this paper is limited.
>
> We would like to clarify the contributions of our work, which are on analyzing the stochastic variants of Halpern iteration with variance reduction. Thus, the contributions are not in coming up with this combination of algorithmic ideas, which would indeed be trivial, but in proving the complexity guarantees. NeurIPS community generally values such contributions, as is evidenced by a large body of literature analyzing other classical algorithms such as the extragradient and Popov's method.
>
> To the best of our knowledge, the only prior literature considering stochastic Halpern iteration for the gradient norm guarantee in min-max problems is by Lee and Kim (NeurIPS, 2021) cited in our paper. However, Lee \& Kim did not provide stochastic oracle complexity results, but only an upper bound on the expected squared gradient norm with pre-assumed noise variance bound requirements. Providing an explicit variance reduction method with technical analysis to obtain the near-optimal complexity results is part of major contributions of this paper. For example, bounding the adaptive batch size $S_2^{(k)}$ in the paper is non-trivial and requires new analysis techniques. Moreover, our paper shows that recursive variance reduction methods can be applied effectively to (convex-concave) min-max problems (not only to min-only problems as in existing work, which typically relies on descent lemmas), which is another conceptual contribution of this paper.
>
> > Question: The proposed algorithm can have a better convergence property. However, it is not clear whether the proposed algorithm converges faster than existing methods in practice. The authors should report some numerical results to show the advantage of the proposed algorithm.
>
> We would like to reiterate that our contribution is theoretical, with our main focus being on proving near-optimal complexity guarantees for stochastic monotone inclusion settings (minimizing gradient norm in unconstrained min-max problems), where the guarantee is on the last-iterate. However, we have uploaded a revised version of our paper with preliminary numerical results done in limited time, for illustration.
>
> We consider solving robust least squares with the Lagrangian relaxation, where $u = (x, y)^T$ and $F(u) = (\nabla_{x}L_{\lambda}(x, y), -\nabla_{y}L_{\lambda}(x, y))^T$ for $L_{\lambda}(x, y) = \frac{1}{2n}||Ax - y||^2_2 - \frac{\lambda}{2n}||y - b||_2^2$. We set $\lambda = 1.5$, and use a superconductivity dataset which is of size $21263 \times 81$. We compare vanilla Halpern iteration, extrapolated Halpern iteration (E-Halpern), and Restarted E-Halpern with Alternating Gradient Descent (GDA), Extragradient method (EG), and Popov's method (Popov), in stochastic settings. All Hapern variants are implemented with PAGE estimator considered in our paper; all other algorithms are implemented using mini-batches, and the batch sizes are determined using a grid search and are kept constant. Additionally, we compare E-Halpern with the PAGE estimator against single-sample and mini-batch estimators. More details are provided in the revised version of the paper.
>
> > Question: What’s the advantage of the proposed algorithm against existing Halpern iteration algorithms?
>
> To the best of our knowledge, there are no theoretical guarantees for Halpern iteration applied directly to stochastic setups. Moreover, when implemented with small batch sizes, the algorithms exhibit oscillatory behavior and do not appear to be convergent. In particular, variance control through either large batch sizes or variance reduction appears crucial. As discussed and proved in our paper, variance reduction leads to lower oracle complexity by a factor of the order $\frac{1}{\epsilon}.$
>
> > Question: There are several variance reduction techniques such as SVRG, SAGA and SARAH. Why the authors choose the variance reduction technique in SARAH?
>
> In this paper, we do not assume the finite-sum structure for the monotone operator for which SVRG-type and SAGA-type variance reduction estimators were originally developed. Recursive variance methods (including SARAH) have been shown to attain the optimal oracle complexity for finding stationary points in non-convex stochastic infinite-sum / online minimization problems. We show such techniques also apply to infinite-sum convex-concave min-max problems for the first time and we further obtain near-optimal complexity results when combined with Halpern iteration. Additionally, PAGE estimator is compatible with our convergence analysis framework.

---

### Official Review · Reviewer_W3Ku · 2022-07-16

**Rating:** 6
**Confidence:** 3
**Soundness:** 3 good
**Presentation:** 3 good
**Contribution:** 2 fair

**Summary:**

The paper considers solving stochastic monotone inclusions with respect to a (continuous) operator $F$ under three setting:
1. $F$ is $\gamma$-cocoercive (implying $\frac{1}{\gamma}$-Lipschitz continuity and monotonicity)
2. $F$ is monotone and Lipschitz
3. $F$ satisfies a sharpness condition, which is weaker than strong monotonicity

The authors propose 3 variants of stochastic Halpern iterations that achieve improved stochastic oracle calls $O(1/\epsilon^3)$ for the first two settings and $O(1/\epsilon^2 \log(1 / \epsilon))$ under additional sharpness condition.


**Questions:**

- Can you remove the need to know the distance to a solution and replace it with some proxy for this quantity?
- Can you comment on the requirement of $\lVert u_0 – u^\star \rVert$, $\epsilon$ and $\sigma$? Would it be possible to remove the dependence on any of them with a reasonably simple algorithm?
- Can the existing VR frameworks for VIs, such as [1, 5, 34], be extended for monotone inclusions in the setting you consider? These methods consider finite-sum-like structures, but could there be a way to extend their results in your opinion?


**Limitations:**

I would suggest the authors to comment on the need for $\lVert u_0 – u^\star \rVert$, $\epsilon$ and $\sigma$ in setting the parameters of the algorithms proposed. They are partially addressed in the main text, but I believe a more thorough discussion would be preferable.

To my knowledge, I do not suspect any potential negative societal impact of this work.

**Strengths And Weaknesses:**

The paper is written in a clear manner for the most part. The related work seems cover the relevant literature sufficiently. The algorithm and the settings are studied and the proof techniques are highlighted with possible challenging components. I would suggest to talk about the techniques in the high level in section 1.2 after formalizing the setting and set of assumptions. Moreover, Theorem 4.1 and 5.1 were presented in a somewhat hasty manner without sufficient discussions.

The authors highlight the connections and differences between monotone inclusions (MI) and monotone variational inequalities (VI), role of unbounded/bounded set for approximately solving each case, the classical Halpern iterations with its properties in relation to the problem setting considered. The authors also lay out the challenges pertaining to the stochastic setting and their algorithmic framework.

While doing so, they motivate the algorithmic choices in conjunction with the main challenges in solving stochastic monotone inclusions with Halpern iteration. I find the high level explanations in Section 1.2 to be informative and helpful for following the main claims of the manuscript. The authors also build upon their main results by explaining the key properties of their variance reduction (VR) framework, and the respective variance bounds. Then, they consider a simple iterative scheme that uses a ``conceptual’’ estimator with target properties, which is later expanded to mini-batch setting and the final VR scheme. It helps the reader understand the main components of the convergence bound and the main difficulties of the analysis.

Given the results in the literature and the discussion about tightness of the bounds in Appendix A (which could be mathematically more rigorous for more concrete deductions), the VR framework proposed in Algorithms 1,2 and 3 provide interesting convergence results in terms of the dependence on target accuracy and variance. I find the results of Section 4 and 5 (Lipschitz monotone and Lipschitz monotone with sharpness condition, respectively) to be more interesting in general, but, to my knowledge, the results exhibit novelty to some extend in the literature. Moreover, Corollary 3.2 and Lemma 3.3 presents nice complementary results. Especially, Lemma 3.3 proves bounds on the norm of the difference of consecutive iterates, which might be of use for future work.

My main criticism is with respect to the parameter choices of the algorithms. Specifically, the requirement to know the initial distance to a solution, which is a rather restrictive assumption to make, and the need to fix the target accuracy ahead of time. The algorithms proposed need to know the variance of stochastic oracle, target accuracy and distance to the solution to set the number of iterations and the (adaptive) batch-size. Although the sample complexities achieved in the paper are important and propose good contributions, these algorithmic requirements, to my understanding, limit the impact.

---

> ### Author Response · Authors · 2022-08-01
> **Response to Reviewer W3Ku**
>
> Thank you for your thoughtful feedback. We hope that our response below clarifies the contributions of our work; please also see detailed responses to other reviews. Please let us know if there is any additional information we can provide and that could potentially convince you of the strengths of our contributions.
>
> > Q1: Can you remove the need to know the distance to a solution and replace it with some proxy for this quantity?
>
> Yes, because the distance to a solution only appears in the total iteration number $N$ for termination to have $\mathbb{E}[||F(\mathbf{u}_k)||] \leq \epsilon$, where $\epsilon$ is the target error. In practical settings, we can terminate the algorithm once observing the norm of stochastic estimator $\tilde{F}(\mathbf{u}_k)$ is smaller than $\epsilon$, i.e., when $||\tilde{F}(\mathbf{u}_k)|| \leq \epsilon$, as is discussed in Lines 245-249 in our paper (Lines 251-255 in the revision). Because our algorithm ensures that  $\mathbb{E}[||F(\mathbf{u}_k) - \tilde{F}(\mathbf{u}_k)||^2] \leq \frac{\epsilon^2}{k}$, we also have $\mathbb{P}[||F(\mathbf{u}_k) - \tilde{F}(\mathbf{u}_k)|| \geq \epsilon] \leq \frac{1}{k}$, by Chebyshev bound. Hence, after $\Omega(\frac{1}{\epsilon})$ iterations, once we have $||\tilde{F}(\mathbf{u}_k)|| \leq \epsilon$, $||F(\mathbf{u}_k)||$ is also $O(\epsilon)$ with probability at least $1 - \epsilon$. So $||\tilde{F}(\mathbf{u}_k)||$ can be a proxy for a stopping criterion that can also be efficiently evaluated.
>
> > Q2: Can you comment on the requirement of $||\mathbf{u}_0 - \mathbf{u}^*||$, $\epsilon$ and $\sigma$? Would it be possible to remove the dependence on any of them with a reasonably simple algorithm?
>
> $||\mathbf{u}_0 - \mathbf{u}^*||$: As we discussed in **Q1**, this term only appears in the total iteration number $N$ needed to reach the target error, and $||\tilde{F}(\mathbf{u}_k)||$ can be a good proxy that is easy to evaluate for termination.
>
> $\epsilon, \sigma$: These two terms only appear in the batch sizes needed for PAGE estimator. Here $\epsilon$ is the chosen target error, and the variance $\sigma^2$ is known in certain risk minimization problems. Further, in practical implementations, the batch sizes can be constant hyperparameters to tune when $\sigma$ is unknown and we do not want to fix $\epsilon$ ahead of time. One generic way to find the proper batch sizes is by a grid search. By our preliminary numerical results, this strategy gives good empirical performance, at least on a proof-of-concept example (included in the revised version).
>
> > Q3: Can the existing VR frameworks for VIs, such as [1, 5, 34], be extended for monotone inclusions in the setting you consider? These methods consider finite-sum-like structures, but could there be a way to extend their results in your opinion?
>
> We believe it is possible with alternative convergence arguments, but not completely clear at this point. The convergence analyses for VIs usually rely on gap/merit function-type analysis that appears quite different than the potential function-based arguments used in the analysis of variants of Halpern iteration. It is worth mentioning again that our results are already near-optimal in terms of the dependence on $\sigma$ and $\epsilon,$ as discussed in Appendix A. Although we do not have proof for this, it appears to us (based on working on this problem in the meantime) that the possible suboptimality in terms of other problem parameters would need to be dealt with using a new variance reduction method in place of existing ones.

---

### Meta-Review · Area_Chair_7k3f · 2022-08-26

**Recommendation:** Accept
**Confidence:** Certain

**Metareview:**

Reviewers appreciated the technical difficulties and novel theoretical results established in this work and recommend acceptance.

**Award:**

No

---

### Decision · Program_Chairs · 2022-09-14

Accept